# High-density sampling reveals volume growth in human tumours

Arman Angaji[1], Michel Owusu[2], Christoph Velling[1†], Nicola Dick[2,3], Donate Weghorn[2,3]*, Johannes Berg[1]*

[1]Institute for Biological Physics, University of Cologne, Cologne, Germany; [2]Centre for Genomic Regulation, Barcelona, Spain; [3]Universitat Pompeu Fabra, Barcelona, Spain

## eLife Assessment

The article uses a cell-based model to investigate how mutations and cells spread throughout a tumour. The paper uses published data and the proposed model to understand how growth and death mechanisms lead to the observed data. This work provides an **important** insight into the early stages of tumour development. From the work provided here, the results are **convincing**, using a thorough analysis.

**\*For correspondence:**
dweghorn@crg.eu (DW);
bergj@uni-koeln.de (JB)

**Present address:** †Kishony lab, Faculty of Biology, Technion, Haifa, Israel

**Competing interest:** The authors declare that no competing interests exist.

**Abstract** In growing cell populations such as tumours, mutations can serve as markers that allow tracking the past evolution from current samples. The genomic analyses of bulk samples and samples from multiple regions have shed light on the evolutionary forces acting on tumours. However, little is known empirically on the spatio-temporal dynamics of tumour evolution. Here, we leverage published data from resected hepatocellular carcinomas, each with several hundred samples taken in two and three dimensions. Using spatial metrics of evolution, we find that tumour cells grow predominantly uniformly within the tumour volume instead of at the surface. We determine how mutations and cells are dispersed throughout the tumour and how cell death contributes to the overall tumour growth. Our methods shed light on the early evolution of tumours in vivo and can be applied to high-resolution data in the emerging field of spatial biology.

## Introduction

The evolution of a solid tumour is governed by the division, motion, and death of cancer cells. Genetic mutations arising during cell divisions can serve as cell markers to track this dynamics. From the observed spatial distribution of mutations, it should in principle be possible to infer the spatio-temporal principles of tumour evolution: Is the growth rate uniform across the tumour, or does growth predominantly take place near the edge of the tumour? What is the interplay between the tissue dynamics of the tumour and its genetic evolution? These broad modes of tumour evolution affect for instance the signature of neutral evolution, the response to selection, or the number of low-frequency mutants which can confer therapy resistance (*Waclaw et al., 2015*; *Chkhaidze et al., 2019*).

However, to answer such questions on the basis of genetic tumour data is challenging because only partial information is available: (i) The sequencing depth (average number of reads covering a nucleotide in NGS sequencing) is finite. This means that only high-frequency mutations are observed. (ii) Usually only a small number of samples are taken from different parts of a solid tumour, which limits the information on mutations present in the full tumour (*Gerlinger et al., 2012*; *McGranahan and Swanton, 2015*). (iii) Longitudinal data from ctDNA measurements are highly limited in the observable range of mutations and provide noisy frequency estimates.

**eLife digest** Our bodies are made up of organs and tissues, which, in turn, are made up of individual cells. Normally, different types of cells in a tissue perform distinct roles, working together to keep the tissue healthy and functioning properly.

However, a genetic mutation that makes them divide and grow uncontrollably can cause tumors to from. As the tumor grows, its cells can undergo further mutations. Each new mutation can serve as a 'marker', allowing scientists to track how a tumor has grown, by looking at which cells have specific mutations.

Angaji et al. wanted to know if tumors grow through cells on the surface dividing more quickly and invading surrounding tissue; or if all cells in and across the tumor divide at the same rate.

To answer this question, the researchers used high-resolution data looking at where in a tumor mutations accumulate. The experiments examined the early evolution of a tumor because only early mutations resulted in enough detectable cells through sequencing. The researchers then compared these 'tumor maps' to simulations of tumors growing in different ways, to see which growth mode fit the maps better.

Angaji et al. found that the tumors they looked at grew uniformly across the tumor volume. They also established that the overall growth of the tumor was slow compared to the rate of growth predicted by the speed of the cells dividing. This means that the development of a tumor is finely balanced between net growth and shrinkage, and a small change in the external conditions could potentially kill a tumor.

Angaji et al. have developed methods that will allow us to better track tumor growth, and provide further insights into cancer biology. These high-resolution tumor maps may provide clues about how to treat different types of tumors depending on how they grow.

Over the next years, some of these restrictions will be lifted by the advent of spatial genomics (*Takei et al., 2021*; *Zhao et al., 2022*; *Lomakin et al., 2021*). These techniques allow assaying the genomic information almost at single-cell level in intact tissue sections. Currently, the attainable sequencing depth is too low to identify point mutations across different parts of the tumour. However, it is clear that the coming-of-age of spatial genomics will bring new opportunities to understand the past evolution of a population of tumour cells from a late-stage snapshot. This implies a need for new tools to analyse spatio-temporal evolution, since standard tools of population genetics, like the site-frequency spectrum, are designed for spatially mixed populations and disregard spatial information.

One particular question concerns two different modes in which a tumour can grow; surface growth and volume growth. Under surface growth, the cancer cells divide predominantly at the border with healthy tissue. The potential reasons for this spatial dependence include higher nutrient levels near normal tissue, higher levels of metabolic waste products in the tumour bulk, or mechanical stress in the tumour centre (*Shraiman, 2005*; *Montel et al., 2011*). A faster growth rate at the edge of the tumour leads to a radially outward growth of the cell population. The surface growth mode is well-known from bacterial growth (*Hallatschek et al., 2007*). In tumours, some evidence for surface growth comes from histological stainings, which show an enhanced level of the Ki-67 protein (a cellular marker for proliferation) near a tumour surface (*Brú et al., 2003*; *Waclaw et al., 2015*; *Hoefflin et al., 2016*). However, the reverse situation has been found as well (*Zhao et al., 2022*), with elevated Ki-67 levels near the centre of a renal carcinoma. The surface growth mode also has a long history in the modelling of tumour evolution (*Greenspan, 1972*; *Ward and King, 1997*; *Baish and Jain, 2000*; *González-García et al., 2002*; *Komarova, 2006*; *Sottoriva et al., 2010*; *Waclaw et al., 2015*; *Sottoriva et al., 2015*; *Iwasaki and Innan, 2017*; *Sun et al., 2017*; *van der Heijden et al., 2019*) and has been used to analyze multi-region tumour sequencing data (*Sottoriva et al., 2015*; *Sun et al., 2017*; *Chkhaidze et al., 2019*; *Sinha et al., 2022*; *Li et al., 2022*; *Noble et al., 2022*; *Fu et al., 2022*).

In volume growth, on the other hand, cancer cells grow irrespective of their location in the tumour: although each cell has a physical location, and upon division its offspring is in a similar location, location does not affect cell division or death. Under volume growth, subclones can originate from any location in the tumour (*Zhao et al., 2022*). As a result, under volume growth mutation frequencies evolve exactly in the same way they would do in a well-mixed population, even though the resulting

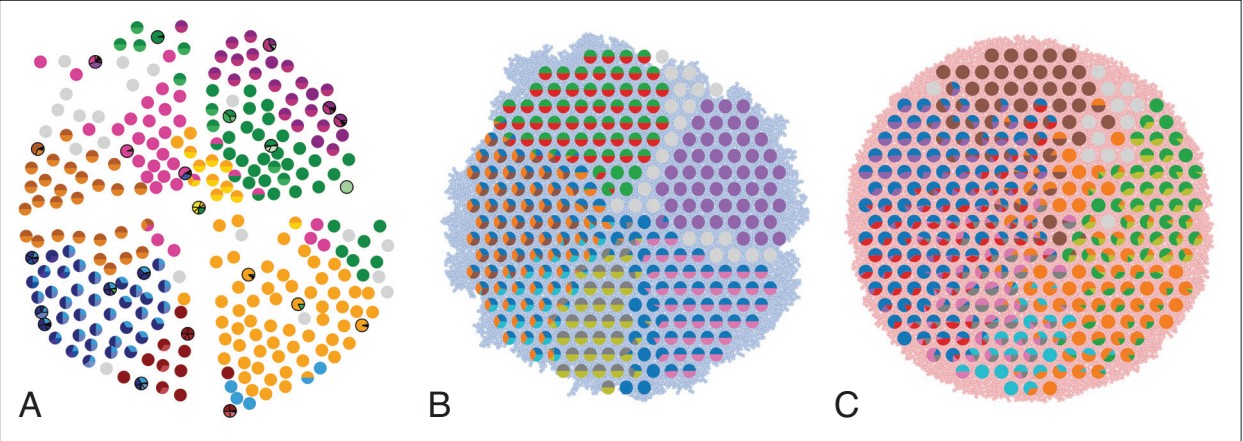

**Figure 1.** Multi-region sampling of a hepatocellular tumour and cell-based simulations. **A** shows the spatially resolved sequencing data of 285 samples of a hepatocellular carcinoma analyzed by *Ling et al., 2015*. Each sample is indicated by a small pie chart in which colors indicate specific mutations, and slice sizes indicate the mutation frequencies within each sample. The 23 samples highlighted by a black outline were also subjected to whole-exome sequencing. The samples form a honey-comb structure, because the tumour slice had been cut into four quadrants, see Fig S1 in *Ling et al., 2015*. (**B and C**) Results of a cell-based simulation in the surface growth mode (**B**) and the volume growth mode (**C**). In each case, 280 evenly spaced samples were taken from the population of 10000 cells of a 2D simulation, see text. The most frequent mutations are shown as in (**A**), superimposed on the structure of the simulated tumour.

tumour will generally be spatially heterogeneous (see below). Mixed population models have been used extensively to model all aspects of tumor evolution, from tumorigenesis to the formation of metastases and the response to therapy (*Luebeck and Moolgavkar, 2002*; *Michor et al., 2004*; *Durrett, 2013*; *Bozic et al., 2013*; *Foo and Michor, 2014*; *Heyde et al., 2019*). They also form the basis of almost all population-genetic approaches to analyzing tumour data (*Attolini et al., 2010*; *Williams et al., 2016*; *Williams et al., 2018*).

To tell between these two different evolutionary dynamics, we use high-resolution data on the spatial distribution of mutations found in solid tumours. We use whole-exome mutation data obtained from hepatocellular carcinomas and published previously (*Ling et al., 2015*; *Li et al., 2022*). From these snapshots of late-stage tumours we infer how the tumours grew in earlier stages. To this end, we develop metrics of intra-tumour heterogeneity which leverage the information from the spatial position of all samples.

## Methods
### Spatially resolved data
We analyze two datasets where large numbers of samples (> 100) were taken from different hepatocellular tumours, once from a two-dimensional section (*Ling et al., 2015*), and once from a three-dimensional microsampling (*Li et al., 2022*).

In *Ling et al., 2015*, 285 samples of a planar section of a hepatocellular carcinoma of diameter 35mm were taken and analyzed. Each sample consisted of about 20,000 cells (*Ling et al., 2015*) determined by cell counting. 23 of these samples were subjected to whole-exome sequencing (average read depth of 74 reads per nucleotide), and loci with mutations found in these 23 samples were probed by genotyping (by Sequenom and Sanger sequencing) in all 285 samples (*Ling et al., 2015*). *Figure 1A* shows the positions of the samples and the mutations found at each position. The data thus effectively consist of a set of 285 samples at high spatial resolution, but with limited genomic information (genotyping), and a subset of 23 samples at high genomic resolution from whole-exome genomic sequencing, but limited spatial resolution. We reanalyzed the whole-exome data as described in Appendix 4.2. The genotyped data consists of 35 mutated loci. Calling variants jointly on all 23 whole-exome samples and filtering as described in Appendix 4.2 returned 217 mutated loci.

In the second data set, *Li et al., 2022*, samples were taken from multiple planar sections of two different hepatocellular tumours from the same patient. This gives a three-dimensional picture of the genetic heterogeneity of these tumours. 169 samples were taken from 11 sections of tumour T1,

which was approximately of 20mm diameter, and 160 samples from 6 sections of tumour T2, which was of approximately of 15mm diameter. Each sample consisted of ca. 3200 cells. 16 (9) samples from T1 (T2) were whole-genome sequenced and the remainder genotyped using the Ampliseq method. Li et al. report 906 (564) mutations used for genotyping of T1 (T2), of which 563 (259) pass our filters.

## Computational model of volume and surface growth

We use an off-lattice cell-based model, well-known from simulations of tissue mechanics (*Drasdo and Höhme, 2005*; *Van Liedekerke et al., 2019*; *Malmi-Kakkada et al., 2018*; *Metzcar et al., 2019*), which we endow with mutations entering the population at a constant rate per cell division.

In this model, each tumour cell is described by a sphere centered on a particular point in 3D space or 2D space, as well as a genome. Each cell division introduces a new point adjacent to its parent in a random (uniformly distributed) direction, and each cell death removes a point. At division both mother and daughter cell can acquire mutations, the number of mutations is Poisson distributed with mean $\mu$. The spheres are not allowed to overlap and can push one another out of the way to achieve this. This off-lattice approach differs from cellular automaton models of tumour growth, where cells occupy discrete lattice sites (*Waclaw et al., 2015*; *Chkhaidze et al., 2019*). Specifically, in off-lattice models like the one we use, cells can gradually push one another out of the way during growth, see below. Cells cannot move by themselves but as they divide overlaps are introduced and resolved by pushing the neighboring cells.

We implement the resolution of overlaps introduced by a cell division using a 'width-first' pushing-algorithm, where the immediate neighborhood of the newly placed cell is resolved first before proceeding to the new overlaps this step introduces. A fast search first identifies overlaps with the current cell in its close neighborhood. The cell then pushes its neighbor along their connecting axis only by the overlap plus a small margin which acts as a buffer and avoids many tiny recurring overlaps. The neighboring cell is then added to a queue for subsequent iterations. Neighborhoods are shuffled such that there is no particular order in which cells are pushed. After each overlapping cell in the neighborhood is pushed and added to the queue, the search for overlaps is repeated on the first cell in the queue, adding new pairs to the queue, which are in turn iteratively resolved until the queue is empty. Thereby each cell division can set off a cascade of rearrangements of cell positions. This level of microscopic detail of course comes at a computational cost. Using off-lattice cell-based modelling, population sizes of several ten thousand cells can be simulated, compared to billions of cells with a cellular automaton (*Waclaw et al., 2015*).

In regions of high cancer cell density, the division rate may be reduced (for instance due to lack of nutrients, toxic metabolic products, or mechanical stress). We effectively encode such potential spatial effects in the rate at which individual cells divide: the cell division rate $b = b(\rho)$ depends on the local density of cells $\rho = \rho(\mathbf{r})$. For simplicity, we consider a division rate $b(\rho)$ which decreases in a straight line from $b(\rho = 0) = b_0$ to $b(\rho = \rho_c) = 0$, see Fig. S1. Setting $\rho_c = \infty$ leads to a constant division rate for all cells, irrespective of their local density. This growth mode is termed volume growth. Lower values of $\rho_c$ lead to an increased growth rate in parts of the tumour where the local density of cancer cell is low, and hence enhance the growth rate at the surface of the tumour relative to the bulk of the tumour. Surface growth is characterized by a $\rho_c$ equal to the density defined by the minimal cell distance. By changing $\rho_c$ one can thus tune the dynamics of this model continuously from volume to surface growth. To describe the genetic changes in the tumour, we use an infinite-sites model and a constant mutation rate per cell division. *Figure 1B and C* show the resulting tumor sections for the surface and volume growth mode, respectively.

For a detailed description of the model, its implementation and performance in comparison with variants of the kinetic Monte Carlo and pushing algorithms, see Appendix 1. The code is available as a Julia package at https://github.com/aangaji/TumorGrowth (copy archived at *Angaji, 2024*).

## Results

### Volume versus surface growth

Under surface growth, cells divide predominantly near the edge of a tumour. Under volume growth on the other hand, cells divide uniformly across the entire tumour mass. To distinguish these two modes in empirical data, we look at the angle between a line from the tumour centre to a parent clone and

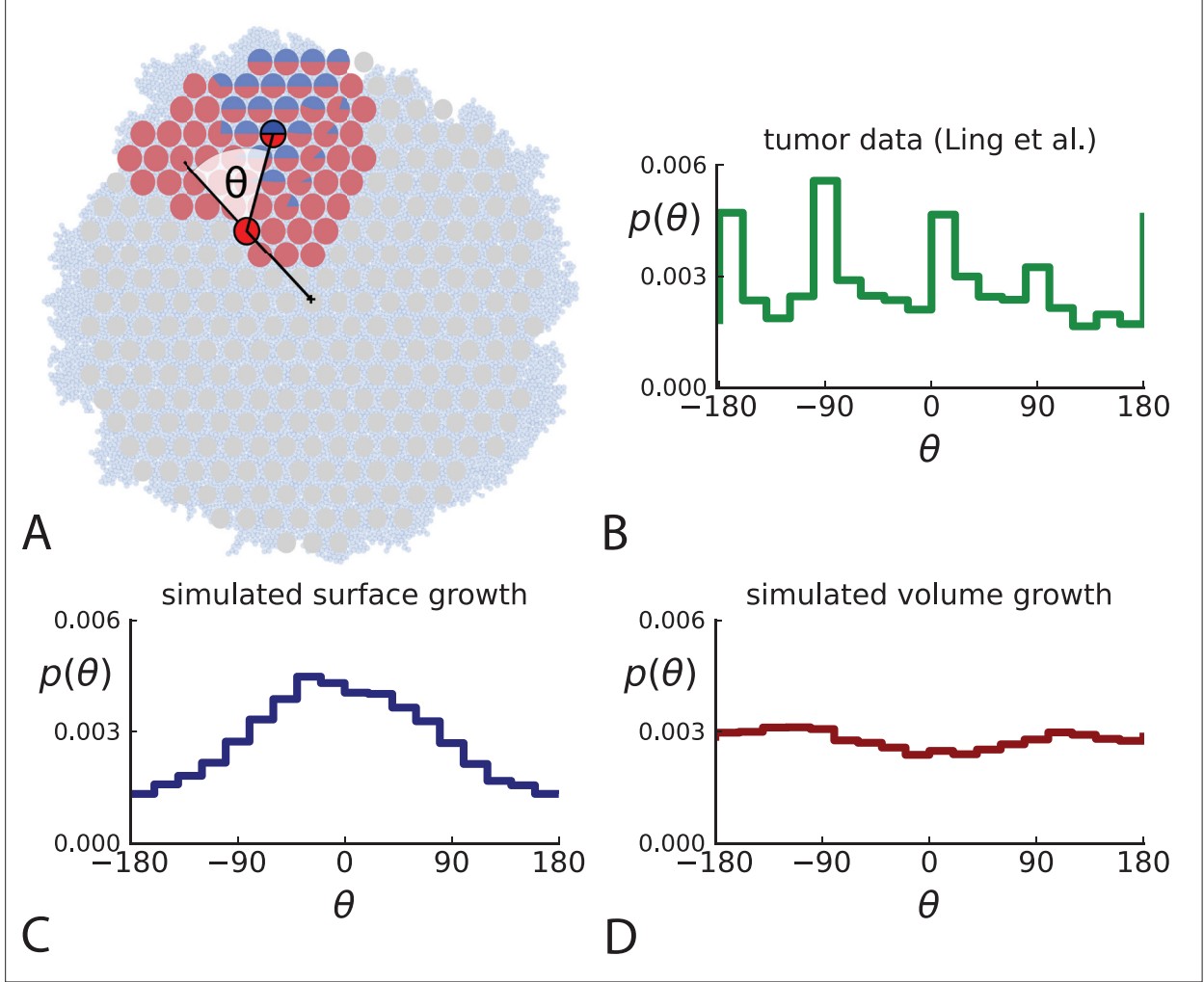

**Figure 2.** Relative position of mutants under different growth modes. (**A**) The direction angle $\theta$ quantifies the direction of a new mutant clone relative to its parent clone. In this illustration, a new mutant clone indicated in blue appears and grows radially outward on a red parental background, resulting in an angle $\theta$ near zero. Each pair of cells indicated in red and blue contributes to the distribution of $\theta$, with the statistical weight of each mutant clone adding to one (see Appendix 2). (**B**) The distribution of angles $\theta$ for different mutant clones found in the spatially-resolved data of *Ling et al., 2015*. The peaks at $\theta \approx -180^{\text{deg}}, -90^{\text{deg}}, 0^{\text{deg}}$ come from individual clones. The contributions from different clones to the distribution of angles is shown in *Appendix 2—figure 1*. (**C and D**) show the corresponding distribution of angles for numerical simulations. Subfigure **C** shows simulations of surface growth, resulting in a distribution of direction angles with a pronounced maximum near zero. Under volume growth (**D**) a nearly flat distribution is seen. For **C and D**, simulations were run in three dimensions with a maximum population size of 40,000 cells grown at division rate $b = 1$, a rate of cell death $d = 0.4$ and $d = 0.8$ for surface and volume growth respectively, and a whole-exome mutation rate $\mu = 0.3$ before taking a two-dimensional cross-section of 280 samples mimicking the sampling procedure in *Ling et al., 2015*. (The different death rates were chosen to make the extinction probabilities of the populations comparable for the two cases. Changing these rates did not affect the distributions of angles.).

from the parent to their offspring, see *Figure 2A* and Appendix 2. Under surface growth, offspring tend to lie radially outward from their parents, leading to a distribution of these direction angles centered around zero. Under volume growth, cells divide isotropically, leading to a uniform distribution of angles. The distribution of direction angles is a model-independent metric, it does not rely on a particular model of cell dynamics, population dynamics or sampling. It is also robust against the presence of selected subclones: if a subclone grows isotropically from some point, but at an elevated rate, it will contribute to a flat distribution of angles like all other clones.

In *Figure 2B* we show the distribution of direction angles found in the spatially resolved data of *Ling et al., 2015*. The empirical data clearly show a uniform distribution of the direction angle, and thus no evidence for radially outward growth caused by faster growth near the edge of the tumour. For comparison, we look at the distribution of direction angles in simulations of different growth

modes. For surface growth, the histogram of direction angles shows a pronounced maximum at $\theta = 0$ produced by the radial outgrowth of clones (*Figure 2C*), in a simulation of volume growth, we find a flat distribution (*Figure 2D*). These simulations were done with samples placed in the same positions as in the real tumour to eliminate effects from uneven sampling.

This result is corroborated by the average number of mutations of a clone as a function of its distance from the tumour centre. Under surface growth and neutral evolution, the number of cell divisions since tumorigenesis, and hence the number of mutations found in a cell, increases with distance from the tumour centre. However, we do not find this signature of surface growth in the tumour data. Instead, we find that the number of mutations at different distances from the centre is compatible with volume growth (see *Appendix 3—figure 4*).

Similarly, the dataset of *Li et al., 2022* with three-dimensional sampling also shows the signature of volume growth, both in the distribution of direction angles (Appendix 2.2) and in the number of mutations as a function of distance from the tumour centre, see Appendix 3.3.

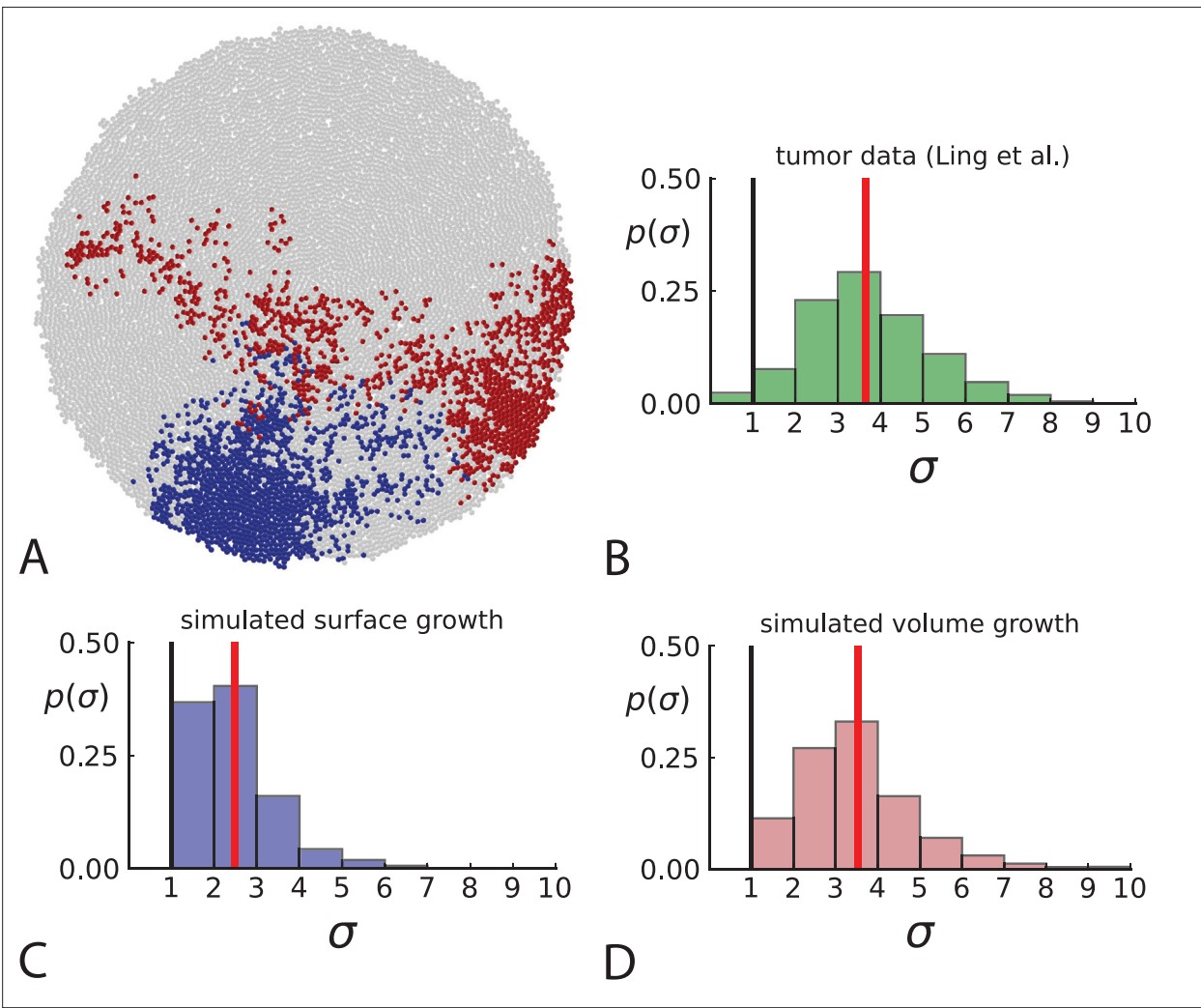

**Figure 3.** Dispersion of mutations within the tumour. (**A**) Cells with a particular mutation can form tight spatial clusters within a tumour (simulated example here: mutant shown in blue), or they can be more widely dispersed (mutant shown in red). We quantify the dispersion of a mutation using the dispersion parameter σ, see text and Appendix 4. In this illustrative example, the blue mutation has a small dispersion parameter $\sigma = 1.3$, the red one has $\sigma = 2.5$. (**B**) Histogram of the dispersion parameters σ across 217 mutations in the whole-exome data of *Ling et al., 2015*. The red line indicates the mean of the histogram. (**C**) and (**D**) show the corresponding histograms for simulations of surface growth and volume growth, respectively. The simulations were run in 3D with populations grown up to 40,000 cells before taking 23 evenly spaced samples from a 2D cross-section. Only mutations with a whole-tumour frequency larger than 1/40 were considered, mimicking the limited sequencing resolution in the Ling et al. data. Simulation parameters were division rate $b = 1$, mutation rate $\mu = 0.3$, and death rates $d = 0.4$ and $d = 0.8$ for surface growth and bulk growth, respectively.

This result is at variance with a recent spatio-phylogenetic analysis performed by *Lewinsohn et al., 2023*, which finds a signal of surface growth on the same data. In Appendix 3, we show that the findings of *Lewinsohn et al., 2023* are compatible with volume growth; on artificially generated volume-growth data sampled like in Li et al., the algorithm used in *Lewinsohn et al., 2023* also returns a spurious signal of surface growth.

## Spatial dispersion of cells

Another aspect where surface and volume growth lead to qualitatively different behaviour is the spatial dispersion of mutations. Cells carrying a particular mutation can in principle form a tightly spaced colony or be dispersed throughout the tumour mass, see *Figure 3A*. It turns out this dispersion of mutants differs between surface and volume growth.

To quantify the spatial dispersion of a mutation, we define the dispersion parameter σ: For a given mutation, we compute the average distance of all pairs of samples carrying that mutation and divide by the average distance these cells would have if they were arranged next to one another in a spherical shape (the tightest configuration possible). Values of σ much larger than one can arise when a mutation is scattered throughout the tumour, with many cells that do not carry that mutation located between cells that do. The dispersion σ can be estimated from mutation frequencies in the whole-exome sequencing data of *Ling et al., 2015*, see Appendix 4 for details.

*Figure 3B* shows a histogram of the dispersion parameter σ for 217 mutations in the whole-exome sequencing data of *Ling et al., 2015*. We find an average value of the dispersion of approximately 3.5, but the values for specific mutations can be much higher than that (up to 9). *Figure 3C and D* show the corresponding distributions for simulations of volume and surface growth, respectively. The distribution of dispersion parameters found under volume growth agrees with that in the empirical data (Kolmogorov-Smirnov statistics 1.13), whereas surface growth generates lower values of the dispersion parameter (Kolmogorov-Smirnov statistics 5.74 indicating a poorer match with the empirical data, see Appendix 4.3 for details). Also in the three-dimensional dataset of *Li et al., 2022*, we find higher values of the dispersion parameter than expected under surface growth (Appendix 4.4). However, the larger distances between samples in three dimensions, compared to the samples in a single plane, turn out to limit the accuracy with which the dispersion parameters can be computed. A higher number of samples would be needed to make the distances between samples comparable between the two- and three-dimensionally sampled data.

Volume growth offers a simple mechanism causing the dispersion of mutations: Tumour cells displace one another when new cells are born and grow. In this way, two cells which are initially close to one another and share a mutation can be pushed apart by the birth and growth of adjacent cells that do not share the mutation. The further apart a pair of cells becomes, the higher the probability that they will be moved even further apart due to the motion of the increasing number of cells between them. This amplification of initially small distances leads to an instability that can generate the large values of the dispersion σ found both in the data of *Ling et al., 2015* and in numerical simulations of the volume growth model (*Figure 3B and D*, respectively). On the other hand, under surface growth, cells do not push each other apart, leading to lower values of the dispersion parameter. Also, high values of the dispersion parameter cannot arise spuriously due to positive selection, as selected clones would enter the population later than neutral ones (given the same final frequency), and thus do not have as much time to be moved within the tumour.

The dispersion of cells in a growing population via this pushing effect is well known in tissue mechanics (*Ranft et al., 2010*; *Malmi-Kakkada et al., 2018*). In the context of cancer, it provides a mechanism for the clonal mixing found by ultra-deep sequencing of multiple samples from a solid tumour (*Sottoriva et al., 2015*; *Suzuki et al., 2017*; *Sun et al., 2017*), which leads to mutations that occur at high frequencies in some samples being present at low (but non-zero) frequencies in other samples. Dispersion and clonal mixing thus arise as a by-product of volume growth. This is in contrast to surface growth, where a specific dispersal mechanism has been postulated to account for the observed spatial distribution of mutations (*Waclaw et al., 2015*).

To probe how this migration of cells between different regions of a tumour affects multi-region sampling, we look at the number of mutations detectable in a limited number of samples. Specifically, we consider the fraction of mutations present in a certain subset of samples (relative to the mutations appearing in all the samples taken together) and ask how this fraction is affected by migration.

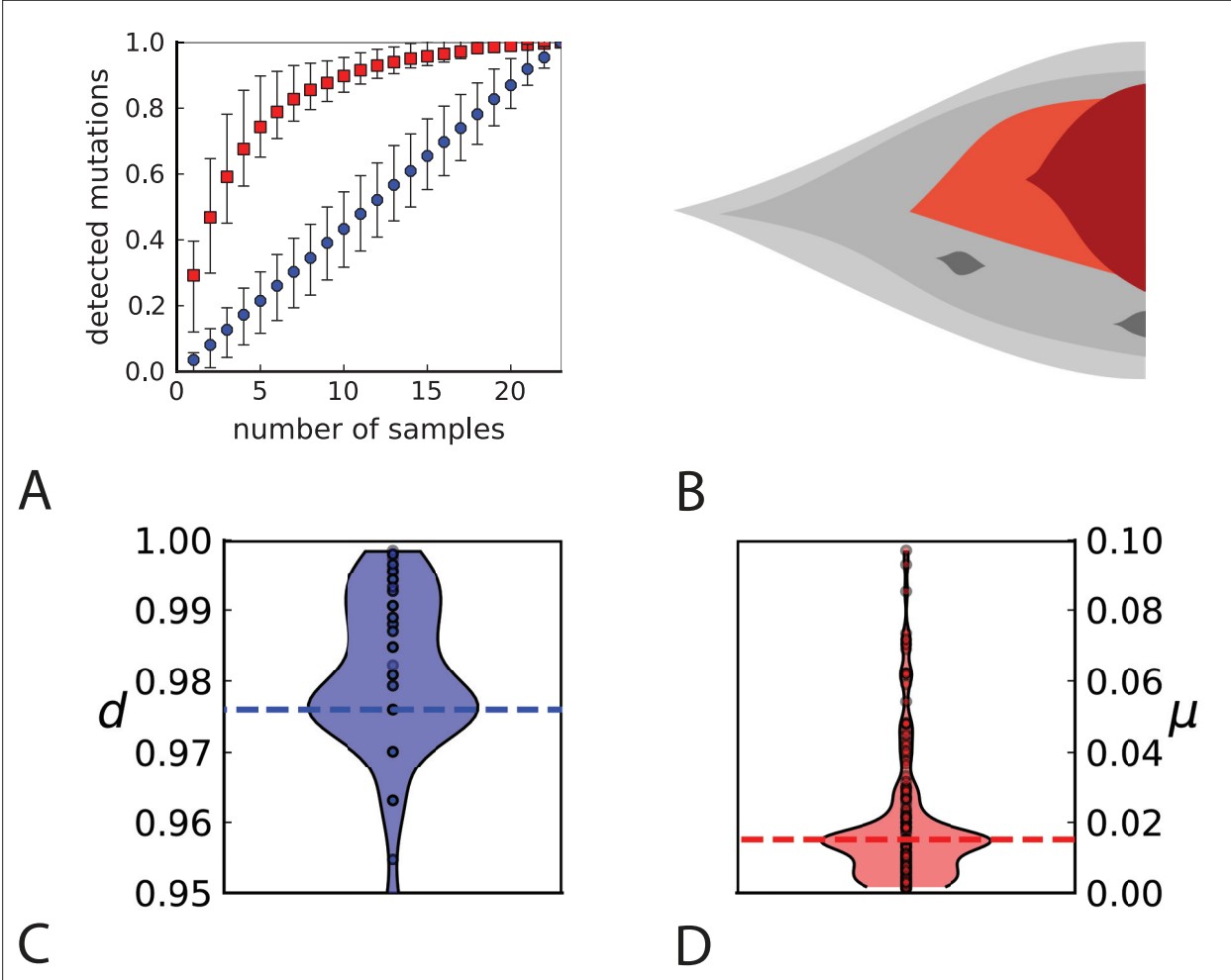

**Figure 4.** Rate of cell death and the mutation rate. (**A**) We ask how a limited number of samples identify mutations. We pick a subset of the whole-exome sequenced samples of *Ling et al., 2015* and plot the fraction of mutations present at least in one of these samples against the number of samples in the subset (red symbols, fractions are relative to the number of mutations present in at least one of the 23 samples. Mutations must be supported by at least 5 reads at a coverage of at least 150). The procedure is repeated (blue symbols) with those mutations removed that occur in some other sample with a higher frequency than the frequencies with which the mutation occurs in the subset of samples. Error bars indicate the range of the 95-percentile. (**B**) The schematic Muller plot shows how cell death leads to the loss of clones, some of which have extant offspring. Time runs on the horizontal axis, the vertical shows a number of different clones indicated by different colours. In the example shown here, the clone shown in light red becomes extinct, leaving behind its darker-shaded offspring clone with no parental clone. The rate of this loss of parental clones depends on the rate of cell death, and can be used for inference, see text. (**C**) shows the inferred rate of cell death and (**D**) the inferred rate of mutation per generation. The violin plots show how the inferred values vary when subsampling different fractions of all mutations and scaling the inferred mutation rate correspondingly, with the dashed lines indicating the mean inferred values. (The fraction of mutations sampled ranges from 0.5 to 0.9, with the results shown separately in Appendix 9).

*Figure 4A* shows the fraction of mutations detectable in a given number of samples randomly picked from the 23 whole-exome samples of *Ling et al., 2015*. One finds that typically about 75% of all the mutations can be detected with only five samples. We repeat this analysis but do not count mutations in a sample that occur with a higher frequency in some other sample. The rationale is that these mutations have arisen elsewhere (where they are present at higher frequency) and have migrated into the sample, where they are now present at a lower frequency. With such mutations removed, five samples typically only contain about one-fifth of the mutations. This is in line with 5 samples representing only about one-fifth of the 23 whole-exome samples. Migration of cells thus makes a single sample capture a far larger share of the genetic variability than expected from its share of the sampled tumour volume and provides an a-posteriori rationale for assessing the genetic variation found in a tumour on the basis of only a few samples.

## Cell turnover

At low rates of cell death, tumours grow nearly at the rate at which tumour cells divide. Higher rates of cell death slow down the net growth of the population and lead to a constant turnover of cells. The relative rates of cell birth and death have been studied in cell cultures since the classic work of *Steel, 1967* using thymidine labelling. Also in cell cultures, live cell imaging has been used to track individual cell divisions (*Lorenzo et al., 2011*). In the following, we use the genetic record in spatially resolved genomic data to probe the role of cell death in vivo during early tumour evolution.

Cell death removes cells, some of which have had offspring prior to death. This can lead to clones without extant parents (*Figure 4B*), and the rate at which such 'orphan clones' arise can be used to infer the rate of cell death. We consider two metrics which quantify how frequently parental cells are removed from the population by cell death. The clone turnover gives the fraction of genotypes whose parental genotype is no longer extant in the tumour. The clade turnover gives the fraction of clades that coincide with their ancestral clade. (A clade is defined by a given mutation. Without backmutations, it comprises all individuals carrying this mutation.) These two metrics depend on the rate of cell death (relative to the birth rate) and on the mutation rate per cell division. Increasing the rate of cell death increases both clade and clone turnover, whereas increasing the mutation rate leads to a larger clone turnover only. These relationships have been determined analytically for a simple model of a growing population (*Angaji et al., 2021*). Inverting these relationships, the relative rate of cell death and the mutation rate can be determined from these metrics, see Appendix 5 and *Angaji et al., 2021* for details and extensive tests.

*Figure 4C and D* show the violin plots for the relative death rate and the mutation rate per cell division inferred in this way from the genotyped data of *Ling et al., 2015*. We find a rate of cell death nearly as high as the rate of birth (relative death rate 0.975 with 90% confidence interval [0.83, 0.995]). Hence during its early stage, the tumour was balanced nearly perfectly between growth and extinction. The inferred exome-wide mutation rate $\mu$ per cell division is 0.015 with 90% confidence interval [0.002, 0.1]. This result is at variance with a recent analysis based on the distribution of mutational distances between samples. *Werner et al., 2020* find low rates of cell death across many cancer data sets. In Appendix 6, we show that this discrepancy arises because widely different rates of cell deaths are compatible with a given distribution of pairwise distances between samples.

To obtain the mutation rate per nucleotide, we need to divide $\mu$ by the effective genome size, that is the number of sites in the genome that could have acquired detectable mutations. For many sites only few or no reads are retrieved during sequencing, so dividing $\mu$ by the size of the whole exome would underestimate the mutation rate. Ling et al. apply a set of criteria to filter out sites with poor coverage (*Ling et al., 2015*, SI): A site must have (1) a total coverage of more than 150 reads when summed over all samples, (2) at least one sample with a coverage of 10 reads or more, and (3) a coverage of 6 reads or more in the normal tissue sample. We find that $4.8 \times 10^5$ sites in the exome fulfill these criteria, and the resulting mutation rate estimate per nucleotide is $1.6 \times 10^{-8}$ with 90% confidence interval $[2e-9, 1e-7]$. This coincides with the result of $2.9 \times 10^{-8}$ SNV per basepair and per generation found in a colon cancer cell line by lineage sequencing (*Brody et al., 2018*). Comparing this result to a mutation rate estimate from a healthy human fibroblasts cell line of $2.7 \times 10^{-9}$ (*Milholland et al., 2017*), this would mean the tumour mutation rate was enhanced by a factor of 5. We note that the relatively small number of mutations in our data likely affects these estimates, and that analyses of larger datasets in the future may lead to more accurate results.

## Mutational signatures and temporal heterogeneity

Somatic mutations in cancer tumours are caused by different mutational processes, which can be identified by their mutational signatures (*Alexandrov et al., 2020*). To investigate which processes were active in the tumours analysed here, we decomposed the mutational profiles of all three tumours into mutational signature components. *Figure 5* shows that the strongest signature in all tumours is the single-base-substitution signature 22 (SBS22). This signature has been associated with exposure to the exogenous mutagen aristolochic acid (*Hoang et al., 2013*), a herbal component used in traditional Chinese medicine. (The cancer patients of the Ling et al. and Li et al. lived in China.) To ask if there is a pseudo-temporal heterogeneity in the mutational processes, we divided the mutations from each tumour into clonal (early) and subclonal (late) mutations. While the Li et al. tumours maintain similar mutational signatures over pseudo-time, we find that the exposure to aristolochic acid of

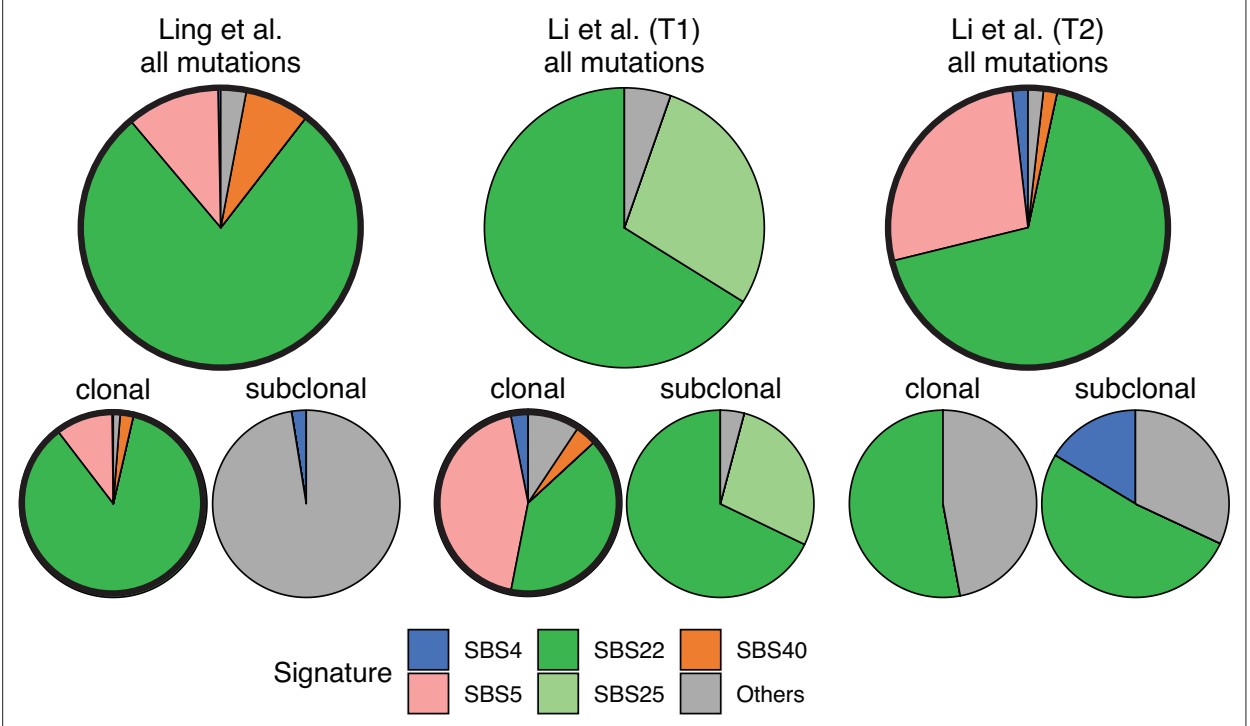

**Figure 5.** Mutational signature decomposition. Relative weights of single-base substitution (SBS) mutational signatures *Brody et al., 2018* in all three tumours were derived with SigNet Refitter where possible (highlighted pie charts), otherwise with non-negative least squares (*Alexandrov et al., 2020*) (Methods). (left) Ling et al., (centre) tumour T1 of Li et al., (right) tumour T2 of Li et al. Top: all mutations, bottom: mutations stratified by their clonality. Signature SBS22 is associated with exposure to aristolochic acid. In the Ling et al. data, this signature is prominent among clonal mutations, but absent in subclonal mutations. Also shown are signatures with relative weight larger than 1% and attributed to endogenous mutational processes (SBS5 and SBS40), tobacco smoking (SBS4) and a signature that resembles SBS22 (SBS25). All other signatures were combined into a single category ('Others").

the patient from Ling et al. appears to have ceased over the course of tumour evolution: while the mutational signature SBS22 dominates the clonal mutations, it is absent in the subclonal mutations, see *Figure 5*. The probability of not observing subclonal SBS22 due to sampling noise (p-value) was estimated to be less than $10^{-4}$, see Appendix 9.

## Population size

The results on the mode of evolution, migration, and cell turnover characterize the early stages of the tumours analysed here. The reason is that in an exponentially growing population, late-stage mutations have frequencies too low to be detected: Under neutral evolution and neglecting fluctuations in clone sizes, the final frequency (in the entire population) of a mutation that arose when a tumour consisted of $N$ cells is given by $f = 1/N$ (*Durrett, 2013*). A lower limit on the frequency $f$ thus corresponds to an upper limit on the population size. In order to determine the population size the tumour had when the mutations we analysed here arose, we generalize this result to include the effects of spatial sampling and stochastic population dynamics, see Appendix 7. The higher the spatial resolution, the lower the whole-tumour frequency of a detectable mutation can be. For this reason, a high spatial resolution means that late-coming (low frequency) mutations can be detected. We find that the mutations detected in the samples with high spatial resolution (surface growth/volume growth analysis and genetic turnover) arose when the tumour consisted of about 80,000 cells. The mutations in the whole-exome samples used to measure the dispersion of mutations arose when the tumour size was around 7000 cells.

## Discussion

### Modes of tumour evolution

We have used genetic data taken from hepatocellular tumours at high spatial resolution (*Ling et al., 2015*; *Li et al., 2022*) to analyse the early evolution of a tumour in vivo. We are not the first to link spatial tumour sampling with evolution: on a larger spatial scale, phylogenetic comparison of samples from primary and metastatic tumours has elucidated the dynamics of metastasis (*Jones et al., 2008*; *Zhao et al., 2016*; *Alves et al., 2019*; *Hu et al., 2020*). Analyses of multiple samples from the same tumour have established intra-tumour heterogeneity and found neutral evolution in many cases (*Gerlinger et al., 2012*; *Sottoriva et al., 2015*; *Ling et al., 2015*; *Williams et al., 2016*; *Sun et al., 2017*; see *Tarabichi et al., 2018* for a critique), as well as instances of selection (*Williams et al., 2018*; *Caravagna et al., 2018*). The mode of growth of a tumour affects the resulting genetic diversity of a tumour (*Noble et al., 2022*; *Fu et al., 2022*).

Here, we have used the spatial information contained in data based on hundreds of samples from single tumours. We found that the tumour evolved under uniform volume growth; there was no evidence of spatial constraints leading to radially outward growth. We also found a substantial turn-over of cells with a rate of cell death comparable to the rate of cell division.

The rate of cell death being comparable to that of cell division implies that the tumour grew much more slowly than suggested by the rate of cell division. Instead, in its early stage, the tumour was balanced precariously between continued growth to macroscopic size and shrinkage: A small increase in the rate of cell death, or a small decrease in the growth rate would have caused the small net growth rate to become negative, possibly even leading to the extinction of the tumour.

The high spatial resolution also allows to track changes in the mutational processes over evolutionary time far more easily than in bulk data. Based on the high-resolution data of *Ling et al., 2015*, we found a drastic difference between the mutational signature of early and late mutations. In this data, the dominant mutational signature in early mutations is associated with an exogenous mutagen (aristolochic acid) found in herbal traditional Chinese medicine and thought to cause liver cancer (*Ng et al., 2017*). This signature is absent in late mutations, compatible with a discontinued dose of the mutagen. In the data set of *Li et al., 2022*, the same signature is dominant both in the early and late mutations, compatible with a continuous dose of the mutagen.

### Mutation dispersion and sampling

We have found mutations in the tumour that are widely dispersed throughout the tumour; as a result they can have low frequencies in one particular sample but high frequencies in other parts of the tumour. We find the same level of dispersal of mutations in off-lattice simulations of growing tissues, where it is due to cells pushing each other out of the way as a part of the growth process. Hence, cells that were initially adjacent to each other and share a mutation can get pushed apart during the course of tumour growth and end up with a large distance between them. We found that this effect can be highly advantageous when assessing the mutations in a tumour from a limited number of samples: cells carrying a particular mutation can migrate into a spatial region that is later sampled. Mutations might thus be present at low frequency in a particular sample, because the cells carrying it migrated from afar, rather than because the mutation either arose late during the growth of the tumour (*Williams et al., 2016*) or was under negative selection (*Weghorn and Sunyaev, 2017*).

Our results are based on three well-encapsulated hepatocellular tumours, resected and sampled in two and three dimensions (*Ling et al., 2015*; *Li et al., 2022*). Potentially, different types of tumours or tumours at different stages may exhibit other growth modes. For instance, tumours that are not encapsulated might have a more heterogeneous environment, and evolve under surface growth. Also, the spatial and genomic resolution of any data set imposes limits on how late stage events can be observed. We estimate that the data analysed here capture the evolutionary dynamics up to 80,000 cells. Increasing the genomic and spatial resolution will allow to identify mutations of lower frequency and thus characterize the in-vivo evolution beyond the early stages. Future techniques to sequence resected tumours at higher spatial and genomic resolution than currently possible will allow to trace key later-stage evolutionary changes like angiogenesis, effects of the tumour microenvironment, or the development of genetic instability.

## Acknowledgements

This work was funded by the Deutsche Forschungsgemeinschaft (DFG, German Research Foundation) grant SFB1310/2 - 325931972. We acknowledge support of the Spanish Ministry of Science and Innovation through the Centro de Excelencia Severo Ochoa (CEX2020-001049-S, MCIN/AEI /10.13039/501100011033), and the Generalitat de Catalunya through the CERCA programme. We also acknowledge funding by the Spanish Ministry of Science and Innovation through grants PGC2018-100941-A-I00 and PID2021-128976NB-I00. This work was also partially funded by the FWF Austrian Science Fund (Erwin-Schrödinger postdoctoral fellowship, J4366). We thank Xuemei Lu and Chung-I Wu for discussions on the data sets (*Li et al., 2022*) and (*Ling et al., 2015*), respectively, and Martin Peifer and Michael Lässig for discussions. Many thanks to Alison Feder and Nicola Müller for discussions on their SDevo algorithm.

## Additional information

### Funding

| Funder | Grant reference number | Author |
| --- | --- | --- |
| Deutsche Forschungsgemeinschaft | SFB1310/2 - 325931972 | Johannes Berg |
| Spanish Ministry of Science and Innovation | CEX2020-001049-S | Donate Weghorn |
| FWF Austrian Science Fund | 10.55776/J4366 | Michel Owusu |
| Generalitat de Catalunya | CERCA programme | Donate Weghorn |
| Spanish Ministry of Science and Innovation | MCIN/AEI /10.13039/50110001 | Donate Weghorn |
| Spanish Ministry of Science and Innovation | PGC2018-100941-A-I00 | Donate Weghorn |
| Spanish Ministry of Science and Innovation | PID2021-128976NB-I001033 | Donate Weghorn |

The funders had no role in study design, data collection and interpretation, or the decision to submit the work for publication.

### Author contributions

Arman Angaji, Conceptualization, Data curation, Software, Formal analysis, Validation, Investigation, Visualization, Methodology; Michel Owusu, Data curation, Software, Formal analysis, Validation, Investigation; Christoph Velling, Data curation, Investigation; Nicola Dick, Software, Investigation; Donate Weghorn, Conceptualization, Data curation, Formal analysis, Supervision, Funding acquisition, Investigation, Visualization, Methodology, Writing – original draft, Project administration, Writing – review and editing; Johannes Berg, Conceptualization, Formal analysis, Supervision, Funding acquisition, Investigation, Visualization, Methodology, Writing – original draft, Project administration, Writing – review and editing

### Author ORCIDs

Donate Weghorn https://orcid.org/0000-0001-7722-8618
Johannes Berg https://orcid.org/0000-0001-6569-3061

Reviewer #1 (Public review): https://doi.org/10.7554/eLife.95338.3.sa1
Reviewer #2 (Public review): https://doi.org/10.7554/eLife.95338.3.sa2
Author response https://doi.org/10.7554/eLife.95338.3.sa3

# Additional files

## Supplementary files
• MDAR checklist

## Data availability

No new data was generated as part of this project. All data used is available from the original publications under the links provided.

The following previously published datasets were used:

| Author(s) | Year | Dataset title | Dataset URL | Database and Identifier |
|---|---|---|---|---|
| Ling et al. | 2015 | Evolutionary genomics in Hepatocellular carcinoma | https://ngdc.cncb.ac.cn/bioproject/browse/PRJCA000091 | NGDC BioProject, PRJCA000091 |
| Li et al. | 2022 | HRA000188 Evolution under spatially-heterogeneous selection in solid tumors | https://ngdc.cncb.ac.cn/gsa-human/browse/HRA000188 | NGDC Genome Sequence Archive, HRA000188 |

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

## Appendix 1

### Computational modelling of volume and surface growth

We use a cell-based and off-lattice computational model of neutral spatial tumour growth in three dimensions. The model is based on a continuous-time many-type branching process well known from the population genetics of asexual reproduction (*Durrett, 2015*), coupled to a simple model of spatial dynamics. Under this model, an individual cell can die, divide, and acquire neutral mutations at division. Cells inherit the spatial position from their parents, and a simple pushing algorithm to avoid overlapping cells is used to simulate the tissue mechanics. The model is implemented by a rejection-kinetic Monte Carlo algorithm as outlined below. In this section, we introduce the dynamics of cell division and spatial displacement, specify the spatial sampling in Appendix 1.2, and in Appendix 1.1 we discuss the performance of this algorithm compared to rejection-free kinetic Monte Carlo algorithms such as the Gillespie algorithm. The code to the model is available as a Julia package at https://github.com/aangaji/TumorGrowth, (copy archived at *Angaji, 2024*).

In our model, the growth dynamics of a cell can differ depending on its local environment. Under volume growth, the birth and death rates of cells are independent of their spatial position, under surface growth we make the rate of cell birth depend on the local cell density. Specifically, the rate $b$ at which a particular cell divides depends on the local cell density $\rho$ via a function $b(\rho)$. The local density $\rho$ at a position $\mathbf{x}$ is computed as a weighted sum over cells $i$ in proximity of $\mathbf{x}$, each contributing a Gaussian weight by their distance

$$\rho(x) = \sum_{\{i \,|\, |\mathbf{x}-\mathbf{x_i}| < 7\,[\text{cell radii}]\}} \frac{1}{\sigma\sqrt{2\pi}} e^{-(\mathbf{x}-\mathbf{x}_i)^2/(2\sigma^2)} . \tag{1}$$

We set the width of the Gaussian σ to 4 cell radii and evaluate the sum over $i$ over cells within 7 cell radii. This dependency of the cell birth rate on the local neighborhood is the reason why updating the rates of all cells at each time step (as required by a rejection-free method) would be prohibitively expensive.

The rejection-kinetic Monte-Carlo algorithm for a density-dependent off-lattice spatial population dynamics is as follows:

1. Pick a cell $i$ uniformly
2. Update cell birth rate $b_i = b(\rho_{nn})$
3. Draw $r$ uniformly from the interval $(0, b_{\max} + d)$
   - $r < d$ : cell dies
   - $d < r < b_i + d$: cell divides
     - → each cell draws new mutations $m \sim \text{Poisson}(\mu)$
     - → resolve overlaps by pushing
   - $b_i + d < r < b_{\max} + d$ : cell skips its turn
4. Increment time by $\Delta t \sim \text{Exp}(\frac{1}{(b_{\max}+d)\cdot N})$

The function $b(\rho)$ is chosen such that rate of birth is maximal at $\rho = 0$ and zero at a density threshold $\rho_c$, and (for simplicity) decreases in a straight line with $\rho$ , see *Appendix 1—figure 1A*. This dependence encodes potential effects neighbouring cancer cells may have on the division rate of a cell, for instance due to the buildup of toxic metabolites or mechanical stress. At high values of $\rho_c$ (relative to the maximum density achieved without overlapping cells, see below), the birth rate is independent of the local cell density, at low values of the threshold the birth rate depends on the local density $\rho$ and becomes zero at $\rho_c$. Thus, by setting a high threshold $\rho_c \to \infty$ the model produces volume growth, and a low value of $\rho_c$ leads to surface growth. Under surface growth, the growth rate is zero or nearly zero throughout the tumour, except at the surface, where the local cancer cell density is smaller than in the tumour bulk. By tuning the parameter $\rho_c$, we can continuously vary the mode of growth between surface growth and volume growth.

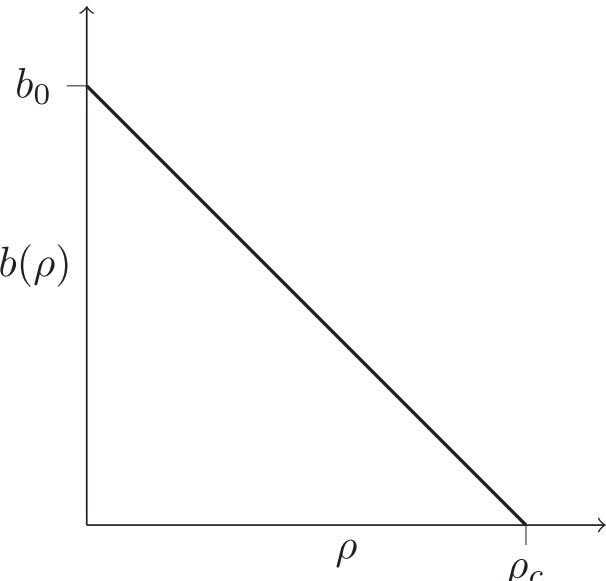

**Appendix 1—figure 1.** Density-dependent birth rate. A schematic plot of the birth rate $b(\rho)$ decreasing with the local density of cells $\rho$, see text.

The cell death rate and the mutation probability at division are the same across all cells. As a result, under volume growth cells divide and die independently of their spatial position and the position of other cells. The population thus evolves like a mixed population would, although every cell has a distinct position in space and this position is inherited from one generation to the next.

During the simulation, cells are uniformly drawn from the population for birth and death. Importantly, this approach allows us to update a cell's birth rate only when it is selected for division as the total rate of events - comprising birth, death and null-event - is uniform across cells and constant $b_{\max} + d$. On the other hand, this means that a cell with a low birth rate might also be rejected (null-event), that is neither die nor divide.

After each draw of a cell, time is incremented. Time steps are exponentially distributed random variables with the mean total rate given by $\tau \sim \mathrm{Exp}\left(\frac{1}{(b_{\max}+d)\cdot N}\right)$, where $b_{\max} = b(\rho = 0)$ and $N$ is the current population size.

At division, a new cell is placed randomly on the surface of the dividing cell. The resulting overlaps between cells are resolved by pushing overlapping cells away from each other. This step of the algorithm is computationally expensive, so keeping track of immediate neighborhoods is essential, see Appendix 1.1 on the computational performance.

The details of the pushing algorithm are as follows: First, the close neighbourhood of the newly placed cell is searched for overlaps. Any overlapping cell is pushed away from the new cell by the overlap plus a small margin which acts as a buffer and avoids many tiny overlaps. The cell is then added to a queue for subsequent iterations. Neighborhoods are shuffled such that there is no particular order in which cells are pushed. After each overlapping cell in the neighborhood is pushed and added to the queue the search for overlaps is repeated on the first cell in the queue, adding new pairs to the queue, which are in turn iteratively resolved until the queue is empty. In this 'width-first' approach, overlaps are first resolved within a given neighborhood and new neighborhoods are appended to the queue, as opposed to a 'depth-first' approach which follows a sequence of pushes by immediately checking a pushed cell's neighborhood for new overlaps and adding these to the front of the queue (which then operates like a stack). This can be efficiently implemented by recursion but has no noticeable performance advantage and results in the same growth patterns.

A commonly used procedure to increase the computational performance in such collision models is to rasterize continuous space. Each cell is assigned to a position on a grid which has a lattice constant of two cell radii. When searching for neighbours - both when pushing and updating the density dependent birth rate $b(\rho)$ - a quick selection of the close (Moore) neighbourhood on the grid returns candidates for which to measure the precise distance. After a cell is either pushed or dies, its position on the grid is updated.

At cell division, each of the two cells - parent and offspring - acquires new mutations whose number is drawn from a Poisson distribution with mean $\mu/2$. These mutations are taken to be neutral and do not affect the birth or death rates.

Our model combines a simple computational off-lattice framework combining genetic mutations in a growing population with a simple tissue dynamics. Additional features, such as selection or different growth laws (for instance Gompertzian growth) can be implemented easily. The model avoids the well-recognized problems that arise in lattice-based models: volume growth on a fully occupied lattice necessitates artificial steps like expanding all distances by some factor while keeping the lattice spacing constant (*van der Heijden et al., 2019*), or moving a column of cells one step in one particular direction to generate empty sites (*Chkhaidze et al., 2019*). However, the computational cost of the off-lattice model means that we can only simulate populations of a few ten thousand cells, compared to billions of cell that can be simulated on a lattice (*Waclaw et al., 2015*).

## 1.1 Algorithm performance and CPU runtimes

Kinetic Monte Carlo (KMC) is a stochastic simulation of a discrete Markov process under continuous time. Under KMC, cells are picked to potentially replicate or die with a certain probability, and after each event time is incremented by a stochastic waiting time. In a rejection or null-event KMC (rKMC) scheme, the cell that has been drawn for update might not change its state (e.g. replicate or die) after a given time step, in contrast to rejection-free KMC (rfKMC) where all events lead to a change of state.

The advantage of rKMC compared to a rfKMC algorithm is that one does not need to know the rates of all cells at each step. Instead, the drawn cell's rates are updated just in time and normalized by a constant, maximum rate, which is larger than each cell's total rate $b_i + d \leq b_{\max} + d$. The maximum rate should be as small as possible to minimize the number of null-events (rejections) and can be conveniently set to $b_{\max} + d$ using the maximum birthrate at $\rho = 0$.

Rejection-free methods, on the other hand, require global updates of the relevant rates at every step. In our model, this would involve computing the local density at every cell. Such an update scales with population size and would be computationally prohibitive. Rejection KMC, on the other hand, allows for local updates and only computes the birth rate $b$ of a single cell per step, at the cost of null-events. However, these null-events do not involve the computationally expensive pushing step (cell movements) and add little to CPU runtime even at low density thresholds $\rho_c$ and high death rates $d$. For a detailed review on the two classes of KMC algorithms, see *Chatterjee and Vlachos, 2007*. The rKMC approach has already been used in non-spatial simulations of populations of cancer cells (*Williams et al., 2018*).

A 3D simulation to 40,000 cells at turnover rate $d/b = 0.2$ takes 5 min under surface growth but 16 min under volume growth on an AMD Ryzen 5 3600 CPU at 3.6 GHz. At high turnover rate $d/b = 0.975$ a simulation under volume growth takes from 19 up to 28 min. The time required for a simulation is almost fully spent on moving cells to resolve overlaps. Drawing birth and death events on the other hand is very fast even at high death rates $d$. High death rates mainly prolong run times by increasing the number of births necessary to reach the final tumour size which entails more pushing steps. This is particularly noticeable for volume growth simulations, where cells predominantly divide and push from within the tumour volume, whereas overlaps under surface growth are more quickly resolved because they affect fewer cells.

## 1.2 Spatial sampling in simulations

The tumour data used here stems from a large number of samples taken from histopathological sections of resected tumours (*Ling et al., 2015*; *Li et al., 2022*). To compare the results of our simulations with this data, we take samples from the simulated populations in a way that mimicks the empirical sampling.

Ling et al. take samples from a single planar section through the tumour (*Ling et al., 2015*). To mimick their sampling scheme, we first cut a two-dimensional plane out of a three-dimensional simulated cell population, and then take small, circular 'punch' samples on a triangular lattice with adjustable number of samples, sample size, and spacing. Given the desired number of samples $n$, we consider a triangular lattice on the plane with a suitable lattice spacing and place samples on the lattice points. The triangular lattice has the highest packing factor $\frac{\pi}{2\sqrt{3}}$ compared to other lattices and fits the most samples into a given area. Using the packing factor we estimate the

lattice constant such that $n$ primitive cells of the lattice cover the area of the plane. The number of samples can fluctuate a little from run to run, depending on the precise shape of the tumour and the orientation of the lattice relative to the tumour. The number of cells in the plane scales poorly with total population size, and at 40,000 total cells, the plane contains about 2600 cells given a narrow plane width of 3 cell radii. To ensure that there is free space between samples (like in *Ling et al., 2015*), we set the number of cells per sample to 5 when taking 285 samples, but have 20 cells or more per sample when 23 samples are taken. The numbers of samples are picked to be the same as in Ling et al. for the spatially resolved data obtained by genotyping and the genomically resolved date obtained by whole-exome sequencing (WES; see main text and Methods: Spatially resolved data). We always compare the results obtained from this spatial sampling in the cross section of 3D simulations to sampling on 2D simulations of the plane, where for a population size of 10,000 cells we take larger samples of 20 cells for 285 samples. The results, however, do not depend much on this size differences of samples and cell populations but rather on the frequency resolution of mutant frequencies (only mutations with frequency larger than a cutoff set by the sequencing depth are retained) and the number of samples taken.

Data from samples taken in 3D from two hepatocellular carcinomas (tumours T1 and T2) from a single patient is discussed in a recent publication by *Li et al., 2022*. The data is available from https://ngdc.cncb.ac.cn/search/specific?db=hra&q=+HRA000188. Similar to Ling et al., hundreds of samples were taken from these tumours (169 from tumour T1 and 160 from T2), but different slices out of the upper hemispheres of the two tumours were taken before taking samples within each slice. Again some of these samples were then analysed by whole-genome sequencing (16 in T1 and 9 in T2), and the remaining 153 and 151 samples were genotyped based on the mutations found by WGS. The multi-region sampling scheme of Li et al., is different from Ling et al., in that it takes samples from several planes of a tumour hemisphere as opposed to sampling from only a single cross section. To imitate the sampling scheme of Li et al., we consider slices in the upper hemisphere of our 3D simulated tumours at distances specified by Li et al.,. From these slices we then take small 'punch' samples at the coordinates specific to the tumours T1 and T2, both for WGS and genotyped samples. Samples in the two tumours are placed rather unevenly, in contrast to the more uniform sampling by Ling et al. For this reason, we place the samples in simulations close to the empirical positions of the samples given in Li et al.

## Appendix 2

### Surface and volume growth: direction of mutants

We construct a simple metric to distinguish surface growth and volume growth. It is based on the directions of newly occurring mutations with respect to their parental background. If the tumour predominantly grows on the surface, then new mutants appear radially outwards relative to their parental clone. All pairs of parental and offspring cells are assigned a direction angle and a statistical weight such that every mutation $m$ contributes equally to the metric.

### 2.1 Algorithmic description

The metric is based on lineages of clones. Clone A is an ancestor of clone B if A's mutations are a true subset of B's mutations. We therefore need to determine the set of clones. This is straightforward for single cells where clones are genotypes. Samples containing many cells, however, may display a mixture of different clones. We use a simple clustering scheme where mutations that coincide across samples are clustered into clones, Appendix 8. We compared the results of this clustering scheme also to those of the LICHeE algorithm for the inference of multi-sample lineages (*Popic et al., 2015*), which produces very similar sets of clones and virtually identical final results. The direction angles of new mutations are computed as follows:

1. Infer clones and their lineage (Appendix 8)
2. for each sample $s_k$ that has cells of clone $\{m_i\}$
   for each ancestral clone $\{m_j\} \subset \{m_i\}$ and each sample $s_l$ that has cells of the ancestral clone
   a. determine distances $\vec{\Delta}_{k,l} = \vec{p}_k - \vec{p}_l$ between $s_k$ and $s_l$
   and $\vec{\Delta}_{l,\mathrm{cm}} = \vec{p}_l - \vec{p}_\mathrm{cm}$ between $s_l$ and the tumour centre of mass
   b. skip if $|\vec{\Delta}_{k,l}|$ is larger than the distance between $s_l$ and the surface
   c. determine the direction angle $\theta_{k,l}^{m_i,m_j} = \angle(\vec{\Delta}_{l,\mathrm{cm}}, \vec{\Delta}_{k,l}) \cdot \mathrm{sign}(\Delta_{l,\mathrm{cm}}^x \Delta_{k,l}^y - \Delta_{k,l}^x \Delta_{l,\mathrm{cm}}^y)$
   (zero if vectors are aligned i.e. offspring lies radially outward relative to the ancestor)
   d. assign weight $w_{k,l}^{m_i,m_j} = \left( \sum_{m_j} \sum_{s_k \in \{m_i\}} \sum_{s_l \in \{m_j\}} 1 \right)^{-1}$ (one over the number of valid pairs)
3. plot the histogram of $\theta_{k,l}^{m_i,m_j}$ with weights $w_{k,l}^{m_i,m_j}$
4. from $c = \left\langle w_{k,l}^{m_i,m_j} \exp(i\theta_{k,l}^{m_i,m_j}) \right\rangle$ compute mean angle as $\arg(c)$ and radius as $|c|$

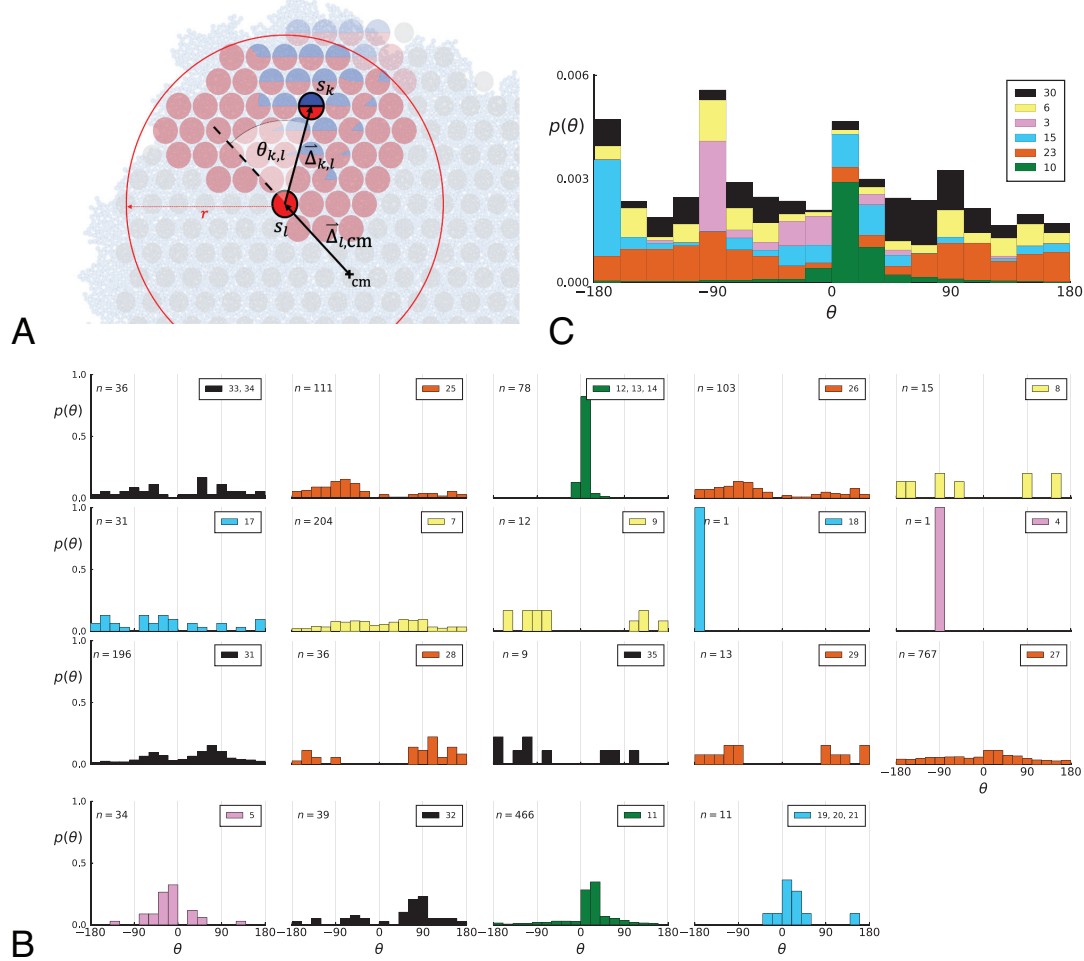

**Appendix 2—figure 1.** Parent-offspring direction angle algorithm illustration and contributions of different clones. (**A**) This cutout of *Figure 2A* showcases the different variables used in step 2 of the algorithm to calculate the angles $\theta$. A new mutant clone with mutations $\{m_i\}$ indicated in blue appears and grows radially outward on a red parental background with mutations $\{m_j\} \subset \{m_i\}$. For a given parental sample $s_l$ belonging to the parental background, we consider every sample $s_l$ belonging to the offspring clone within a distance $r$ from $s_l$, where $r$ is the distance between $s_l$ and the tumour surface. $\theta_{k,l}$ for the pair $s_k$ and $s_l$ is the angle between the arrow $\vec{\Delta}_{l,\mathrm{cm}}$ pointing from the center of mass cm to $s_l$ and $\vec{\Delta}_{k,l}$ connecting $s_l$ to $s_k$ as calculated in step 2c. A weight is assigned to $\theta_{k,l}$ such that the total weight of all pairs where the blue clone is the offspring is 1; this way each clone contributes equally to the distribution of $\theta$. (**B**) The distribution of angles $\theta$ in the Ling et al. data shown *Figure 2B* of the main text comprises contributions from different clones. These contributions are shown here separately for each clone. On each subplot, the top-right label gives the private mutations of the clone and on the top-left label gives the number of parent-offspring pairs contributing to the histogram. The color indicates which of 6 distinct clades the clone belongs to. Crucially, there is no bias for a particular value of $\theta$ across clones. An artefact of giving equal weights to clones are the sharp peaks when there are few parent-offspring pairs. In particular clones 4 and 18 (panel 9 and 10) each have only 1 parent-offspring angle. (**C**) This figure shows the contribution of the 6 distinct clades (clade colours as in Subfigure B) to the total angle distribution in *Figure 2B* of the main text. Histograms of subfigure B of clones belonging to the same clade are added and the resulting clade histograms are stacked (weights of the stacked histogram are divided by the total number of clones for normalisation).

## Step 1

In words, when a mutation $m_i$ enters the population, it defines a new clone $\{m_i\}$ whose genotype consists of the new mutation $m_i$ and all its ancestral mutations. The clone $\{m_i\}$ is the most recent common ancestor of all clones containing mutation $m_i$. We ask how the radial position of this clone relates to its ancestral clones. (2) Given an ancestral clone $\{m_j\}$ we take one sample that has cells of

the offspring clone $\{m_i\}$ and one that has cells of the ancestral clone, (2a) draw a line between the two and (2c) measure the direction angle θ between this line and the line connecting the sample with the ancestral clone to the tumour centre of mass. If tumour growth occurs on its surface (interface with normal tissue), the offspring grows outward, and we expect this angle to be biased towards zero. The direction angle is symmetric with respect to the line between parent and centre of mass, so to recover the whole −180° to 180° range we multiply it by a term which distinguishes clockwise from counterclockwise. We repeat this measurement for all pairs from the two clones and again with all other ancestral clones following the lineage backwards in time. (2d) The weight associated with each direction angle θ is one over the total number of ancestor-offspring angles for clone $\{m_i\}$, such that, after summing over all samples with cells of the ancestral clone, every mutation contributes with equal weight.

In the case where frequencies of clones are known in each sample, one can assign a weight $w_{k,l}^{m_i,m_j} = f_k^{m_i} \cdot f_l^{m_j}$ to the sample pair $k \in \{m_i\}, l \in \{m_j\}$, where $f_x^m$ is the frequency of clone $\{m\}$ in sample $x$. The product of frequencies $f_k^{m_i} \cdot f_l^{m_j}$ is proportional to the number of offspring-ancestor pairs of cells between the two samples. Dividing each weight $w_{k,l}^{m_i,m_j}$ by the sum $\sum_{k,l,m_j} w_{k,l}^{m_i,m_j}$ over all weights involving clone $\{m_i\}$ as offspring again imposes a normalization where each mutation contributes equally to the distribution of direction angles.

## Step 2

Part (b) accounts for a simple geometric effect: If a sample containing offspring is placed randomly around a sample containing ancestors, the direction angle θ is uniformly distributed, however, more cells lie inward than outward from any point except the centre of the disk resulting in a purely geometric bias towards ±180°. The restriction to only consider offspring samples within the distance between ancestor sample and tumour surface mitigates this bias by only considering as many samples inward from the ancestor sample as lie outward from it.

## Step 3

Directly plotting the weighted histogram of direction angles θ gives a picture of the radial direction of growth across all mutations, (4) but one can also compute a mean angle and radius as summary statistics of the distribution. The expression for $c$ is the weighted sum over all unit-length 2D vectors pointing in the directions $\theta_{k,l}^{m_i,m_j}$, resulting in an average vector pointing towards the direction of growth and whose length quantifies the strength of this bias, that is a radius $|c| = 1$ means that all offspring have a direction angle $\bar{\theta} = \arg(c)$ relative to the ancestor, while $|c| = 0$ corresponds to a uniform placement of the offspring relative to the ancestor (in this case $\bar{\theta}$ is indeterminate).

We compare 3D simulations of 40,000 cells under volume growth ($\rho = \infty$) and surface growth ($\rho = 6$) both for sampling single cells and with a sampling scheme that imitates the experimental data of Ling et al,. For each simulated tumour, we take a thin planar cross-section through the centre of mass. (The case of 3D sampling is treated in Appendix 2.2).

For single cells, the distribution of θ can be measured directly for the single cells in the plane, resulting in many offspring-ancestor pairs. To study the effect of sampling, we employ the spatial sampling scheme described in Appendix 1 and take 285 samples of 5 cells each from 2D cross-sections of the 3D simulated tumour. The same sampling is repeated on 2D simulations without taking a cross section allowing for larger sample sizes of 20 cells. Finally, we compute the direction angles $\theta_{ij}^{\{m_i\}}$ and weights $w_{ij}^{\{m_i\}}$ as described above. The resulting distributions are shown in **Appendix 2—figure 2** for 2D slices of 3D simulations and in **Appendix 2—figure 3** for 2D simulations (where larger numbers of cells can be achieved). In both cases, we consider both the uniform sampling of single cells from the tumour and the sampling scheme mimicking (**Ling et al., 2015**) described in Appendix 1.

Both 2D and 3D simulations of surface growth clearly display the expected bias in the distribution of direction angles, compared to the flat distribution under volume growth.

The distribution of direction angles is very similar in 2D and 3D (comparing the left plots for single cells in **Appendix 2—figures 2 and 3** and similarly the right plots for samples). On the other hand, between single cells and samples in each figure, showing a small loss in signal strength (surface growth peak against the volume growth background) due to sampling noise and errors in the reconstruction of parent-offspring clone relations from samples.

**Appendix 2—figures 2 and 3** show that the signal for surface growth can also be detected in a regime of high cell turnover (high rate of cell death $d$ compared to the birth rate). The distributions

of direction angles θ in the low turnover regime are not shown here, they have the same shape as for high $d$ but show less sampling noise.

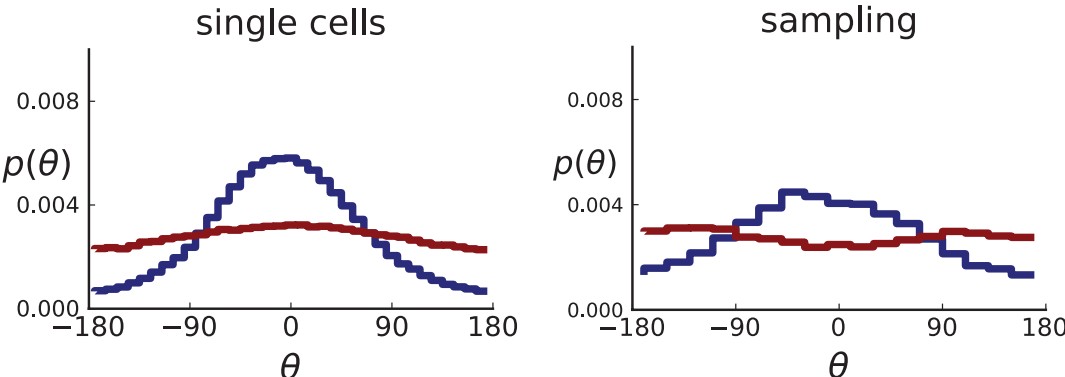

**Appendix 2—figure 2.** Distributions of parent-offspring direction angles $\theta$ for 3D simulations under volume and surface growth. Direction angles are taken from single cells (left) and the 2D spatial sampling scheme (right, Appendix 1). Distributions of parent-offspring direction angles $\theta$ for cross-sections of 3D simulations of 40,000 cells at constant division rate $b = 1$, mutation rate $\mu = 0.3$, as well as cell death rate $d = 0.8$ and $d = 0.4$ under volume growth (red, $\rho = \infty$) and surface growth (blue, $\rho = 6$), respectively. On the left, direction angles are determined for single cells (these curves are shown in **Figure 2** of the main text), whereas on the right, we additionally apply a spatial sampling with 285 samples. The cell death rate $d$ is set lower in surface growth simulations because also the rate at which cell divisions take place is much lower under surface growth due to the reduced growth in the tumour bulk. This leads to frequent population extinctions at higher rates of cell death $d$.

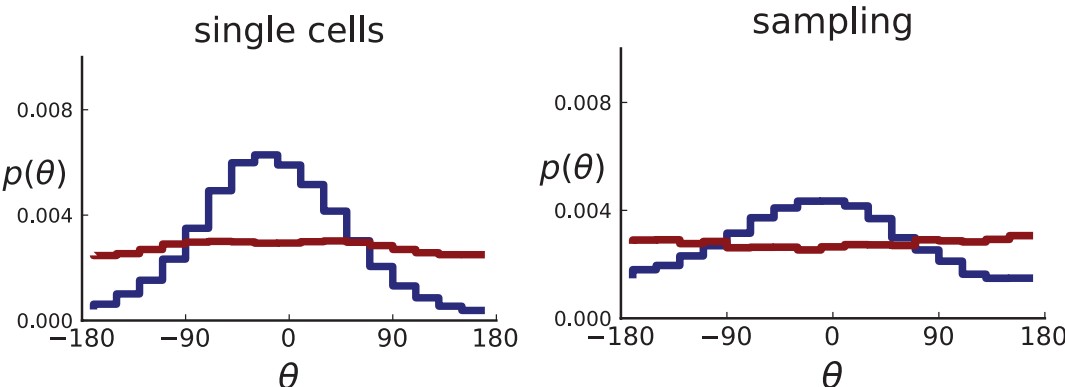

**Appendix 2—figure 3.** Distributions of parent-offspring direction angles $\theta$ for 2D simulations under volume growth (red) and surface growth (blue). Direction angles are taken from single cells (left) and the spatial sampling scheme (right). Parameters are as in **Appendix 2—figure 2** except that populations are grown to 10,000 cells, no cross section is taken, and individual samples consist of 20 cells.

## 2.2 Direction of mutants in the 3D sphere

So far, we have restricted ourselves to a planar cut through the center of a 3D tumour, motivated by the data from Ling et al. which was sampled from such a cross section. However, the definition of the direction of mutants can be readily extended to single cells and samples taken in 3D. An example such data is *Li et al., 2022*.

The algorithm to determine ancestor-offspring direction angles as presented before is the same in 3D, with the exception that angles are not given a sign in step (2b) of the algorithm. The sign was computed relative to the $z$ axis orthogonal to the plane which defined an orientation to the direction angles and distinguished $0 - 180°$ clockwise from counterclockwise. This notion of orientation does not exist in the sphere.

More importantly, the most useful metric to quantify direction of mutants and distinguish surface from volume growth in 3D is not simply θ as on the 2D plane. To find the corresponding metric we ask what function of θ is uniformly distributed on the 3D sphere. To this end, we consider θ the polar angle in spherical coordinates. For isotropic growth of new cells the distribution of θ is specified by the volume element in spherical coordinates, and hence proportional to $\sin\theta$. Correspondingly, its indefinite integral $-\cos\theta$ and hence also $\cos\theta$ is uniformly distributed on the interval $[-1, 1]$. (Since we are looking for a function $f(\theta)$ that is uniformly distributed, the transformation of probabilities gives $\sin(\theta)d\theta = df$ and hence $\sin(\theta)d\theta = \frac{df}{d\theta}$)

*Appendix 2—figure 4* showcases these results using 3D simulations and sampling of single cells. Under volume growth, $\cos\theta$ shows a uniform distribution (red line). Surface growth, on the other hand leads to a bias towards the direction of the pole, $\theta = 0$, and thus a bias towards $\cos\theta = 1$ (blue line).

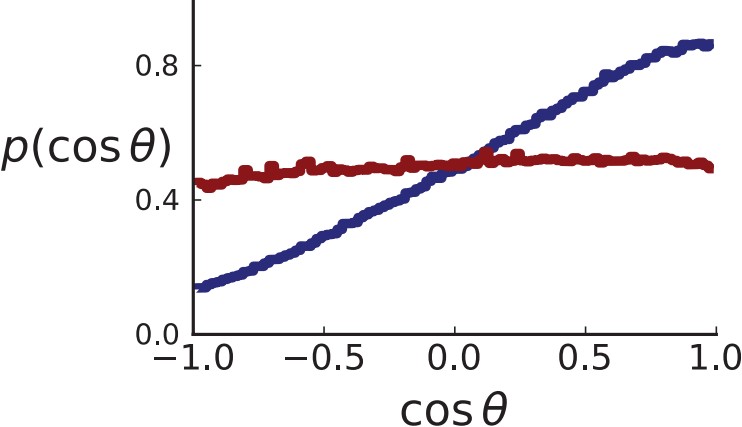

**Appendix 2—figure 4.** Distributions of parent-offspring direction cosines $\cos(\theta)$ for single cells in 3D simulations under volume (red) and surface (blue) growth. Direction angles $\theta$ are taken from single cells in the spherical tumour. Simulation parameters are as in *Appendix 2—figure 2* except that no cross section is taken.

The set of samples taken in 3D by Li et al., see Appendix 1.2, comprises samples that were whole-genome (WG) sequenced as well as samples that were genotyped. The sampling takes places at a lower spatial density than (*Ling et al., 2015*) and read counts are not reported for individual SNVs. In both T1 and T2, the spacing between neighboring samples is 8% ± 2% of the tumour diameter whereas in the considerably larger tumour of Ling et al. (3.5cm compared to the .5cm of T1 and .5cm of T2) samples were taken at median distance of 4% ± 1% of the tumour diameter. Measured in tumour diameters, the sampling density in Li et al. is about half that of Ling et al., mainly due to sampling in 3D where the same number of samples is more spread out than in the plane. While 3D sampling may give a global picture of the intra tumour heterogeneity it comes at the cost of reduced spatial resolution.

However, Li et al. genotype a larger number of mutations than Ling et al. — 906 for T1 and 565 for T2 — because WGS targets a larger part of the genome than WES and they apply less stringent filtering criteria for SNVs compared to Ling et al. (see our discussion of the WES data in Appendix 4).

As explained in Appendix 8 on clonal inference, given the larger genotyping set of mutations (compared to the Ling et. al data), we use the LICHeE tool to determine clones, before identifying the inferred clones within the larger set of genotyped samples, which provide a higher spatial resolution. We then measure the distribution of ancestor-offspring direction cosine $\cos\theta$; the results are shown in *Appendix 2—figure 5*. For comparison with simulations, we perform a spatial sampling of planar cuts through simulated tumours in 3D under surface and volume growth. We use the coordinates of planes and samples from tumours T1 and T2 to mimic their sampling (and potential spatial inhomogeneities therein) in our simulations. First, we extract samples for sequencing at the coordinates of WGS samples before taking samples at the coordinates specified for genotyped samples (*Li et al., 2022*) to confirm the presence or absence of mutations found in the first set of

samples, see Appendix 1.2 on simulated sampling. For the mutations found in the first 'sequencing' set we measure the distribution of $\cos\theta$, see *Appendix 2—figure 6*.

The simulation results show that sampling at the density of the Li et al. data (which is much lower than in the planar sampling by Ling et al.) introduces a radial weak inward bias under volume growth.

The inward bias also affects the distribution under surface growth - a similar effect as in sampling from the cross *Appendix 2—figure 3*. This makes it harder to distinguish surface and volume growth in simulations mimicking the sampling in Li et al. in *Appendix 2—figure 6*.

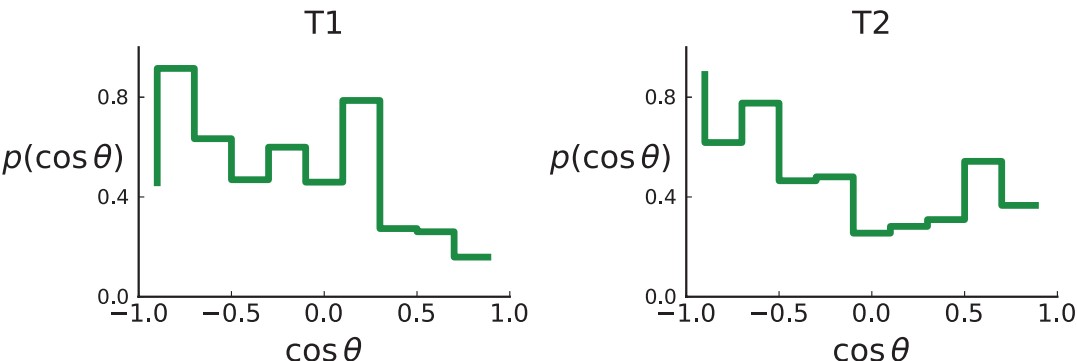

**Appendix 2—figure 5.** Distributions of parent-offspring direction cosines $\cos(\theta)$ for the Li et al. 3D sequencing data. Direction angles $\theta$ are computed from high-spatial-resolution (genotyped) samples of the tumours T1 (left) and T2 (right), after having applied the clustering scheme of 11 to obtain clones from the list of sample genotypes.

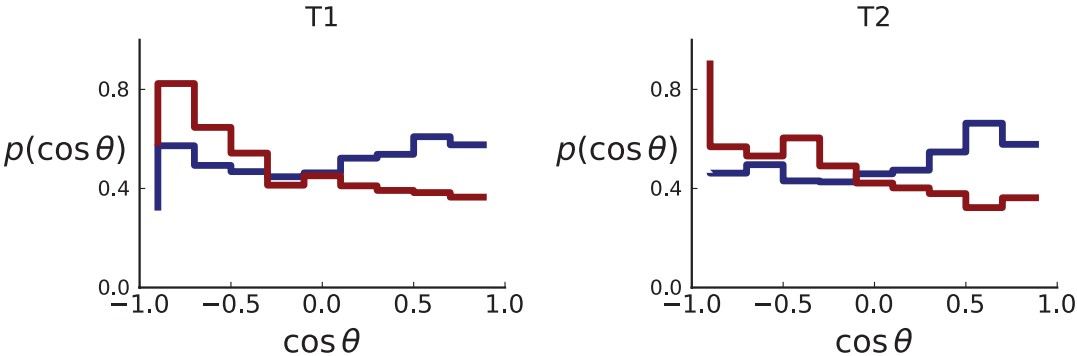

**Appendix 2—figure 6.** Distributions of parent-offspring direction cosines $\cos(\theta)$ for sampling mimicking the spatial sampling of *Li et al., 2022* of 3D simulations under volume (red) and surface (blue) growth. Direction cosines are taken from samples in layers of 3D tumour hemispheres following the sample coordinates of Li et al. for tumour T1 (left) and T2 (right) for direct comparison with *Appendix 2—figure 5*. Simulation parameters are as in *Appendix 2—figure 2*, but no cross-section is taken. Volume growth shows a small downward trend, corresponding to a small radially inward bias. This may be due to the geometric effect discussed in Appendix 2.1 (Step 2).

## Appendix 3

### Mutation density on rings

Surface growth and volume growth can also be distinguished by looking at the accumulation of mutations with increasing distance to the tumour centre of mass, without the need to explicitly infer parent-offspring relationships. Under volume growth, mutations are distributed uniformly in space, whereas under surface growth cells at the surface accumulate mutations at a constant rate as the population grows radially. The latter results in an increasing mutation density (per volume or per area) as a function of distance from the centre of mass.

In this section, we look at the mutation density as a function of distance. We plot the number of mutations found within a ring of radius $r$ and width $w$ against $r$ and normalize by the number of cells or samples in the ring. To this end, we

1. set the origin to the tumour centre of mass
2. select the ring width $w$ at least larger than the distance to nearest neighbours
3. for each ring with radius $r_i$
   a. take all samples within $r_i$ and $r_i + w$
   b. count the mutations in each sample and sum up all counts
   c. divide the total count by the number of samples in the ring to obtain the density

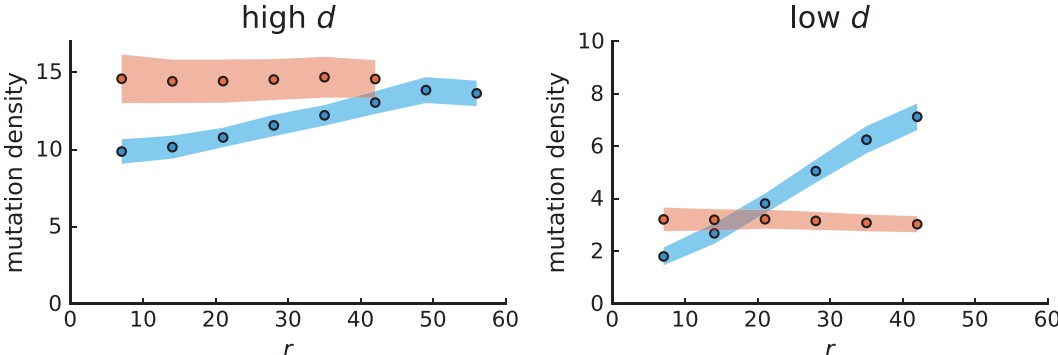

**Appendix 3—figure 1.** Mutation density curves for sampling in 3D simulations. We show the mean mutation density within a ring of radius $r$ and width $w = 7$ (see text). Each curve is averaged over 20 simulations, coloured ribbons indicate the standard deviations from the mean. For each simulation under volume growth (red, $\rho = \infty$) and surface growth (blue, $\rho = 6.0$), we take a cross-section and apply a spatial sampling with 285 samples (high spatial resolution) as described in the text. The right hand side shows simulations with zero death rate $d = 0.0$. On the left, (relative) death rates are high, $d = 0.8$ for volume growth and $d = 0.4$ for surface growth (see also *Appendix 2—figure 3*).

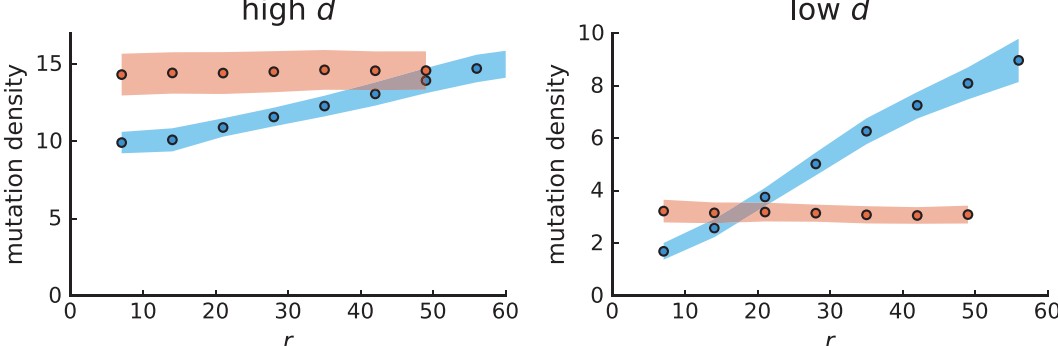

**Appendix 3—figure 2.** Mutation density curves for single cells in 3D simulations. Same as in *Appendix 3—figure 1* but for single cells under volume (red) and surface growth (blue) at high (left) and low (right) death rates.

To mimic the sampling scheme in **Ling et al., 2015**, we take a thin planar cross-section through the centre of mass as described in Appendix 1. Given the sample number of circa $n = 285$ samples used throughout, the nearest neighbour distance is always below 7 cell diameters, which is why we choose the ring width to be 7 cell diameters for both single cells and samples.

We model stochastic effects in sequencing by setting a low coverage of 5 total reads per mutant site per sample and drawing alternate and reference reads from to a binomial distribution with the expected number of mutant reads given by the mutation frequency.

We then measure mutation densities, either for the full population of single cells in the plane or under the sampling scheme.

The mutation density versus ring radius $r$ is flat for volume growth (see **Appendix 3—figure 1**) because all cells divide at the same rate regardless of their radial position. The mutation density therefore assumes a value given by the number of mutations a cell is expected to accumulate until the given tumour size $N$: After $T$ generations (cell divisions) each cell has accumulated $m \sim \mathrm{Poisson}(T\mu)$ mutations. Notably, a cell that is born after $T_0$ generations inherits its parents mutations and ends up with $m \sim \mathrm{Poisson}(T_0\mu) + \mathrm{Poisson}((T - T_0)\mu) = \mathrm{Poisson}(T\mu)$ mutations. Consider a ring of radius $r$ with $n$ cells. Due to the high dispersion of cells under volume growth, the number of divisions $T_i$ is independent for different cells $i$ in the ring at radius and the number of mutations is again Poisson-distributed $m_r = \sum_i m_i \sim \mathrm{Poisson}(\sum_i T_i \mu)$. The expected mutation density $\langle \rho_r \rangle = \langle m_r/n \rangle = \log N/(1 - d/b)\mu$ and standard deviation $\mathrm{std}(\rho_r) = \sqrt{\langle \rho_r \rangle/N_r}$ describe the mutation density curves under volume growth.

For surface growth, the mutation density increases as a straight line, see **Appendix 3—figure 1**. At zero death rate, the mutation density is compatible with a linear dependence on the radial distance. However, at a finite death rates there is a clear vertical offset, with a finite mutation density at zero distance. This is because at a finite death rate, cells die uniformly over the tumour volume, and this allows for the division of other cells in their vicinity. Hence, the higher the rate of cell death, the more cells divide also in the bulk of the tumour; the effect of spatial constraints thus diminishes with the death rate. As a result, for a model of surface growth based on local tumour cell densities, the distinction between the surface and volume growth disappears at high rates of cell death. Importantly, this is not a failure of a metric designed to distinguish between different growth modes, but both modes becoming asymptotically the same at high death rates, with cells dividing uniformly throughout the tumour volume. Specifically, the results shown in **Figure 2B** of the main text and **Appendix 2—figure 5** of a flat distribution of direction angles (or direction cosines in the 3D case) show that there is no enhanced cell growth near the edge of the tumour. However, it is possible that surface growth would manifest itself in a tumour that grows more slowly.

## 3.1 A model of surface growth with explicit spatial dependence

To probe the distinction of surface growth and volume growth at different rates of cell death, we briefly discuss an alternative model of surface growth. Instead of spatial constraints leading to a density-dependent growth rate (see Appendix 1), we make the rate of cell division explicitly depend on the spatial position of each cell. For concreteness, we consider a division rate that is zero in the tumour centre and increases with the radial distance $r$ from the tumour centre to $b = 1$. at its edge,

$$b(r) = \frac{b}{1 + \exp\left(-\dfrac{r - (R - w/2)}{w/s}\right)} \; . \tag{2}$$

$R$ denotes the radius of the tumour, which changes during the simulation, while the parameters edge division rate $b = 1$, edge width $w = 15$ and the profile steepness $s = 10$ are initially set. Under this model, cells in the bulk cease to divide as the edge proceeds to grow outwards. Consequently, the mutation density in the bulk does not increase over time. **Appendix 3—figure 3** shows that under the model where the cell division rate depends on the local density, the surface growth mode has a flat mutation density curve in the high death rate limit because the high turnover allows cells in the tumour bulk to divide at the rate of cell death. In contrast, the mutation density profiles of simulations with explicit spatial dependence of the division rate $b$ on the radial position cells collapse to a line with positive slope when rescaled by the effective mutation rate $\mu/(1 - d/b)$. The dependence of the scale on the turnover rate $d/b$ is simply given by the number of generations $T$ to reach size $N$ with $T = (1 - d/b) \log(N)$.

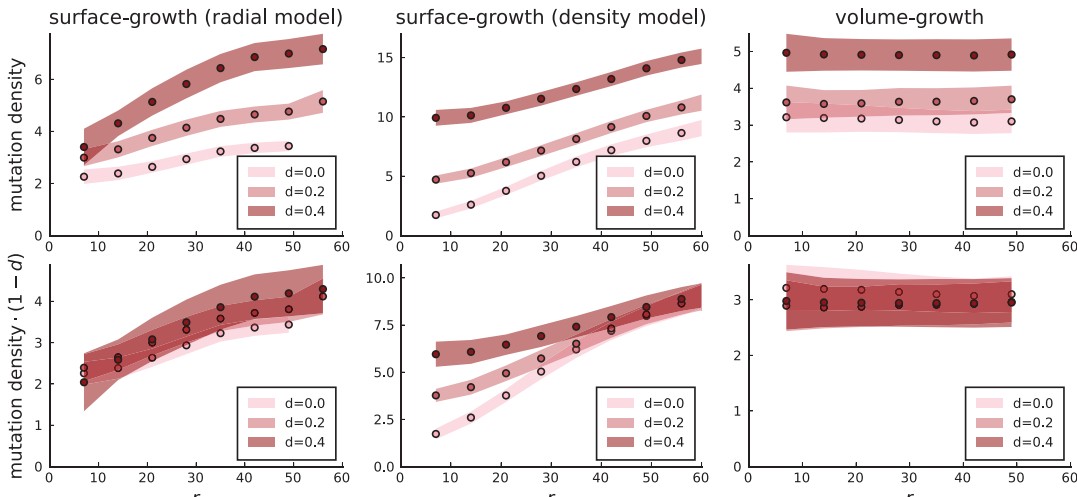

**Appendix 3—figure 3.** Mutation density profiles under different growth models. The mutation density is shown as a function of the ring radius for the model of position-dependent surface growth (with a division rate *Equation 2* nearly zero below the surface), for the density-dependent surface growth model, and under volume growth. Rescaling mutation densities by $(1 - d/b)$ collapses the radial model and the volume growth model to a single curve. Settings for 3D simulations under all models are as in *Appendix 3—figure 2*: $N = 40000$, $b = 1.$, $\mu = 0.3$.

On the other hand, when cells at the center cannot divide even at low density, any positive cell death rate $d > 0.$ leads to a regions without cells (mimicking a necrotic core), which gets larger with higher turnover rate $d/b$.

Simulations of this model of explicit spatial dependence produce the same signature of surface growth in their distribution of directional angles as found in Appendix 2 for the model of surface growth.

### 3.2 Mutation density in the Ling et al. data

*Appendix 3—figure 4* shows the mutation density versus the distance from the tumour centre for the 285 samples from Ling et al. which have been genotyped. The resulting curve is compatible with a flat line, corresponding to a growth mode where new mutations arise uniformly throughout the volume.

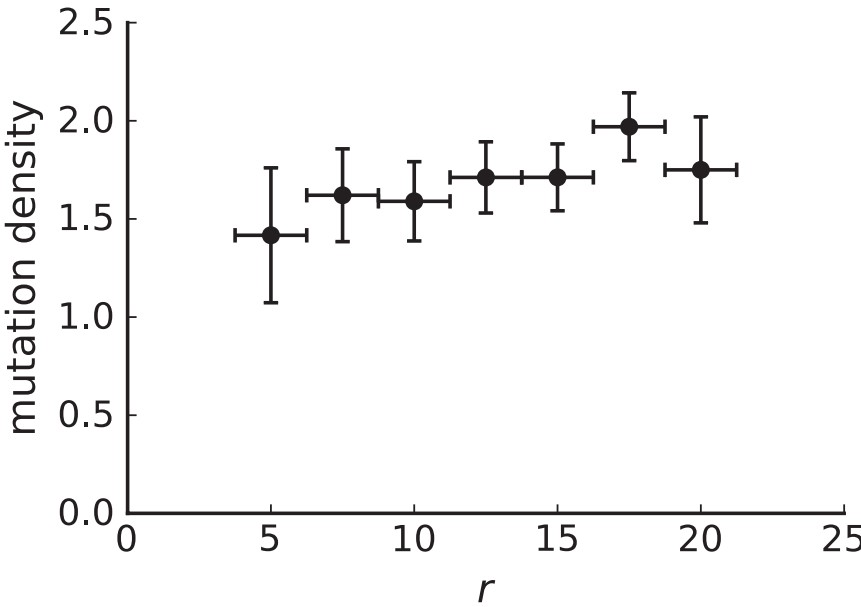

**Appendix 3—figure 4.** Mutation density curve of genotyped samples. The 285 samples from Ling et al. that have been genotyped have a mean nearest neighbour distance of about 1.5mm. Choosing a ring width of 2.5mm allows us to define 8 rings around the samples' centre of mass. The horizontal error bars indicate that samples can fall anywhere in this 2.5mm window. Vertical error bars were estimated using the standard deviation of the mutation density under volume growth derived in the text $\text{std}(\rho_r) = \sqrt{\langle \rho_r \rangle / N_r}$, where we use the measured value of $\rho_r$ in place of $\langle \rho_r \rangle$ to estimate the error. We found in numerical simulations that this yields a lower bound on the error in 3D simulations with sampling. The curve can be considered flat if the change in value is of the same scale as the error. Our null model is volume growth, therefore, a lower-bound of the error estimate favors rejecting flatness of the curve (the signature of volume growth), thus making the estimate conservative. The flat shape of the mutation density curve is again consistent with volume growth rather than surface growth. Rings at radius 50 (centre) and 450 (edge) contain only 2 samples each and were thus dropped.

Throughout this analysis, we focused on the mutation *density*, rather than the number of mutations found in a ring. The reason is that the area of a ring of fixed width increases linearly with the radius $r$ of the ring (for a small width). For this reason, under volume growth, the *number of mutations* in a ring increases linearly with its radius.

## 3.3 Mutation density on shells in 3D

Extending the measurement of the radial change in mutation density to samples taken in a 3D spherical tumour requires only a minor adjustment to the definition as presented in the beginning of Appendix 3. Instead of a ring of radius $r$ we now consider a spherical shell of width $w$ containing the samples with a radial distance within $r$ and $r + w$ from the tumour centre. The mutation density is the ratio between total count of mutations summed across samples and the number of samples in a given shell.

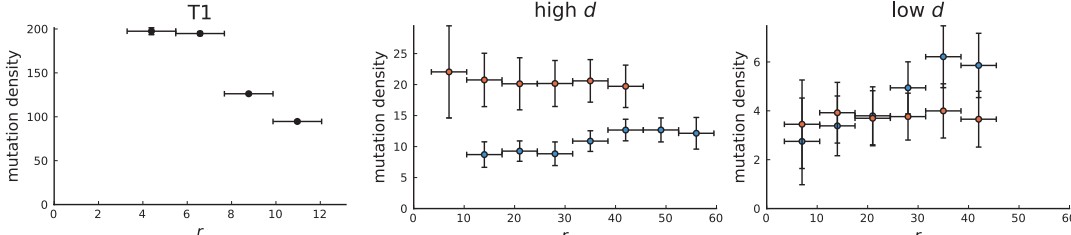

**Appendix 3—figure 5.** Mutation density curve of tumour T1 (left) and T1-like sampling in simulations (centre and right). The plots show mutation density for spherical shells at different distances $r$ from the tumour center. The left plot is based on the 153 genotyped samples from 9 slices of T1 that report the presence or absence of mutations found by whole-genome sequencing of 16 samples. The other two plots show mutation density as a function of shell radius $r$ in simulations of 3D spherical tumours under T1-like sampling. Within each simulation 16 samples are taken from 3 slices for sequencing and 153 samples from 9 slices to check for the presence of the detected mutations. The positions of slices and samples are specified by T1. Parameters of the 3D simulations are as in *Appendix 3—figure 1*.

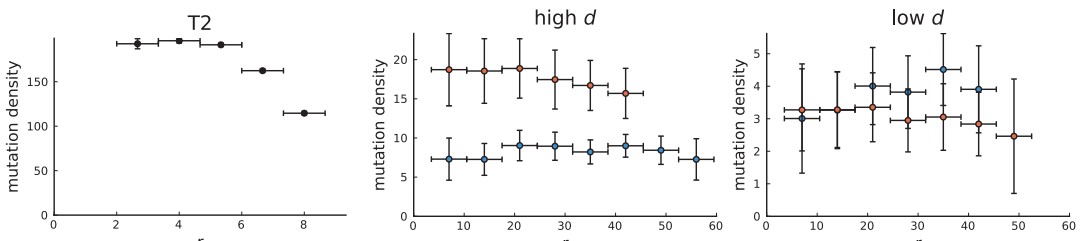

**Appendix 3—figure 6.** Mutation density curve of tumour T2 (left) and T2-like sampling in simulations (centre and right). The plots show mutation density for spherical shells at different distances $r$ from the tumour center. The left plot is based on the 151 genotyped samples from 6 slices of T2 that report the presence or absence of mutations found by whole-genome sequencing of 9 samples. The two plots for simulations (middle and right) are analogous to *Appendix 3—figure 5* but with T2-like sampling.

*Appendix 3—figure 2* shows the mutation density on rings as a function of the radius for single cells of planar cuts through simulated 3D spherical tumours. The corresponding curves for single cells on spherical shells at different radii of the simulated tumours are virtually identical to the mutation density in rings of the cross section. As expected, the density of mutations is independent of the radius for volume growth and increases with radius under surface growth.

However, imitating the sampling scheme of Li et al. in simulations, as explained in Appendix 2.2, shows a poorer distinction between the two modes of growth, see *Appendix 3—figure 5* and *Appendix 3—figure 6*. The radial increase in mutation density under surface growth is dampened in particular at the tumour surface. Under volume growth the curve is still flat (within the error margins). The mutation density calculated for tumours T1 and T2 has a considerably larger decrease near the surface. The slight decrease at high $r$ seen both in the empirical data and the simulations might be caused by the uneven sampling for whole-genome sequencing: WGS samples are taken from few slices mostly inside the tumour and might miss mutations that appear near the surface.

We note here that these results appear to contradict findings in *Li et al., 2022* concerning the distribution of mutations across the tumour. Specifically, Li et al. state that 'peripheral regions not only accumulated more mutations, but also contained more changes in genes related to cell proliferation and cell cycle function' and 'Phylogenetic trees show that branch lengths vary greatly with the long-branched subclones tending to occur in peripheral regions'. We find that the classification of samples into 'centre' and 'periphery' by Li et al. does not coincide with the distance of samples from the centre of the tumour, see *Appendix 3—figure 7*. For tumour T1, there are several samples labeled as 'centre' are further from the centre as some samples classified as 'periphery', for example L1. Furthermore, the samples labeled 'centre' have mutations that belong to a single clade (see Fig. 5 in *Lewinsohn et al., 2023*), which suggest that the classification scheme mixes spatial and genomic information.

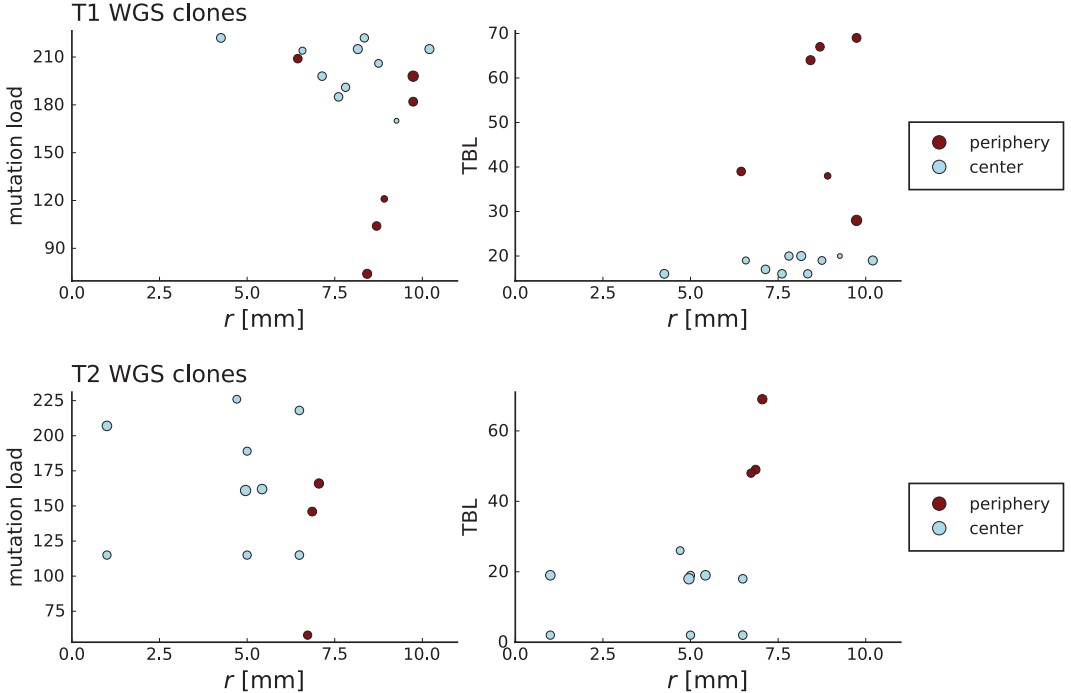

**Appendix 3—figure 7.** Clone sizes and terminal branch lengths (TBL) for genotyped mutations in Li et al. WGS samples. The plots on the left (right) show the number of mutations (terminal branch length) of clones in the WGS samples of tumour T1 (top) and T2 (bottom) against the radial position relative to the tumour center of mass in mm on the x-axis. The size of markers indicates abundance of the clone within the sample and colors are determined by the classification of samples by Li et al. into "periphery" and "center". Samples marked as peripheral have higher terminal branch lengths but in the case of T1 are not located closer to the tumour boundary than those marked as central.

## 3.4 SDevo analysis on simulated volume growth

In this section, we compare our finding of volume growth in the *Li et al., 2022* to the recent result of *Lewinsohn et al., 2023*. In a phylogenetic analysis of the *Li et al., 2022*, Lewinsohn et al. found a significantly elevated rate of cell birth (relative to death) near the edge of the tumour, compared to the tumour's core. This is at variance with our result on the same data, which showed no signal of the outward growth associated with surface growth. We show that the result of *Lewinsohn et al., 2023* is compatible with volume growth: we demonstrate that the phylogenetic analysis of *Lewinsohn et al., 2023* produces a signal of surface growth also on artificial data from a volume growth model, when samples are placed precisely as in the empirical data by Li et al. and analysed as in *Lewinsohn et al., 2023*.

We grew a population from a single cell to size 40,000 in 3D at a relative rate of cell death $q = 0$ and on average $\mu = 10$ mutations per generation. The growth mode was volume growth as described in Appendix 1. Repeating the analysis at $q = .2, .4, .6, .8$ gave similar results. Samples of typically 30 cells were taken from the same positions they were taken from in tumours T1 and T2 of *Li et al., 2022*, respectively. Samples were classified following the classification used in Extended Data Fig. 7 of *Lewinsohn et al., 2023*. In this scheme, the samples with distance from the centre higher than 0.9 of the tumour's radius are classified as 'edge samples', those with distance less than this threshold are classified as 'core samples'. This classification differs from that that used in the main text of Lewinsohn et al., which uses the classification introduced in *Li et al., 2022* briefly discussed in this Appendix 3.3. As the latter classification is not based on distance alone, we will not pursue it here.

We then applied the algorithm SDevo developed in *Lewinsohn et al., 2023* with the parameters provided in xml-template-files by Lewinsohn et al. on the Github page https://github.com/blab/spatial-tumor-phylodynamics (*Lewinsohn, 2023*).

20 shows the resulting distribution of inferred birth rates in the edge and core parts of the tumours across 30 different simulated tumours grown under volume growth. The mean ratio of the inferred birthrates in the edge versus centre samples was 1.47 (95% interval [0.03, 49.7]), even though for volume growth this ratio should be one, or close to one. Specifically, this ratio is higher, and hence the spurious signal for surface growth is stronger, than what Lewinsohn et al. found in the empirical data: Extended Data Fig. 7 of *Lewinsohn et al., 2023* reports the estimated birth rate ratio (edge/core) of 1.15 for tumour T1.

For samples placed as in tumour T2, we found a mean growth rate ratio of 6.98 (95% interval [0.07,33.12]), compared to 3.89 reported for tumour T2 in Extended Data Fig. 7 of *Lewinsohn et al., 2023*. From *Appendix 3—figure 8* we conclude that in many cases of artificial tumours grown under volume growth, SDevo detects a higher birth rate in samples near the edge of the tumour, and therefore does not reliably differentiate between surface and volume growth.

A potential cause for this result is the sampling probabilities. The inference of birth-death model parameters from a phylogenetic tree generally depends on estimates of the probability that an individual from a (possibly large) population was sampled (*Stadler, 2009*). Correspondingly, SDevo requires estimates of the sampling probabilities in the core/edge regions of the tumour as input parameters. From the xml files referenced above we found these parameters were set to 0.2 and 0.1 for the core and edge, respectively. We estimated the sampling probabilities by measuring tumour volume in units of closely packed spheres of the size of samples. Specifically, we calculated the number of samples in a particular part of the tumour (core or edge) divided by the number of spheres of the size of samples that fit into the corresponding volume (edge or core region of a particular tumour) under close packing. This gave $f_{\text{core}} = .00006$ and $f_{\text{edge}} = .00012$ for T1 and $f_{\text{core}} = .00004$ and $f_{\text{edge}} = .00024$ for T2, much lower than the sampling probabilities used in *Lewinsohn et al., 2023*. The absolute sampling probabilities (not just the relative one between edge and core) can in principle affect the inference results, because the inferred death rate depends non-homogeneously on the sampling probability (*Stadler, 2009*), and birth and death rate are inferred relative to one another. However, there is a difference even in the relative sampling probability of core and edge regions between our estimates and the parameters used in *Lewinsohn et al., 2023*; the ratio of sampling probabilities is 2 (core relative to edge) in *Lewinsohn et al., 2023* and about 1/2 in our estimate for T1 and 1/6 for T2.

To test this hypothesis by running SDevo with the sampling probabilities as estimated here, however, this not not lead to qualitatively different results. Further work would be needed to determine the capabilities and the limitations of the SDevo algorithm (*Lewinsohn et al., 2023*).

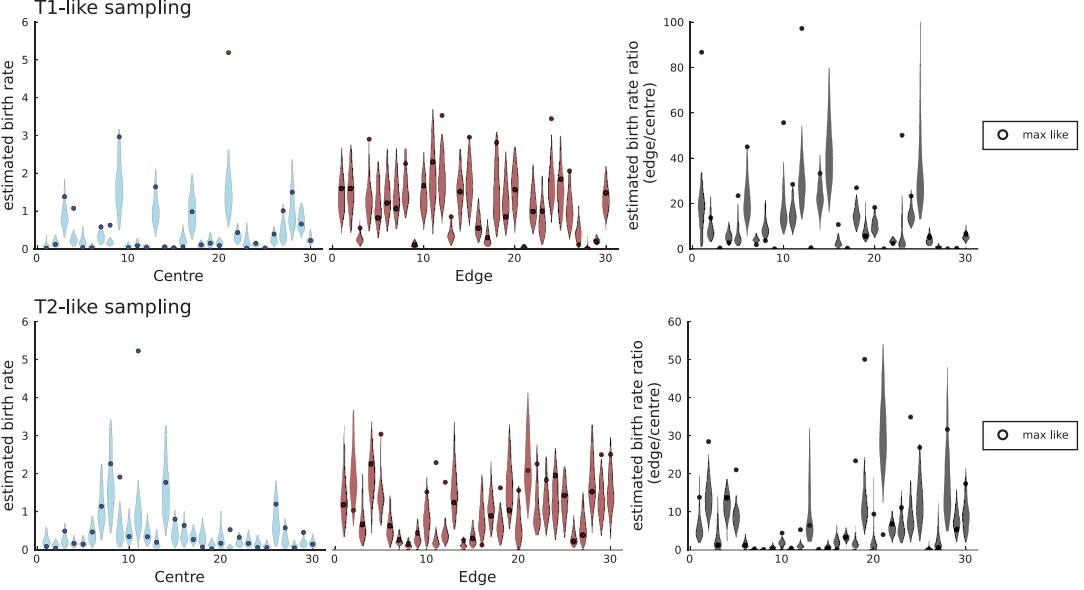

**Appendix 3—figure 8.** Birth rates for periphery and center inferred with SDevo from simulations of volume growth with spatial sampling. We run 30 simulations in 3D to a population size $N = 40000$ at rates $b = 1.$, $d = 0.$, $\mu = 10.$ and take punch samples for deep sequencing at positions specified in *Li et al., 2022* for tumours T1 and
*Appendix 3—figure 8 continued on next page*

*Appendix 3—figure 8 continued*

T2. We only call mutations exceeding a 0.3 cellular fraction threshold within a sample as specified in ***Lewinsohn et al., 2023*** and use the template xml file provided by Lewinsohn et al. to generate inputs for the SDevo algorithm. Each simulation shows a violin plot of inferred birth rates (95% interval of last 500 sampled states of the MCMC-chain in SDevo) for samples labeled as 'centre', samples labeled as 'edge', and the ratios of the two rates, as in Figure 5 of ***Lewinsohn et al., 2023***. A circle marker for each violin plot indicates the values at the maximum likelihood, which often falls outside the 95% interval of the last 500 sampled states.

## Appendix 4

### Spatial dispersion of cells

We quantify how clones are dispersed throughout the tumour, using the WE-sequencing data of *Ling et al., 2015* and *Li et al., 2022*. Rather than looking at sum of spatial distances between cells with a particular mutation, we define a dispersion parameter as these sum of distances relative to those in a dense packing of cells. The dispersion parameter is one for a (hypothetical) dense and spherical arrangement of cells carrying a particular mutation, and larger than one if these cells are dispersed within the tumour (with cells not carrying that mutation between them). For each mutation $m$ we compute the dispersion parameter $\sigma_m$ as follows:

### 4.1 Algorithmic description

1. Obtain all mutant clades and compute whole-tumour frequencies
2. Discard mutations that occur in a single sample or fall below a resolution threshold $f_{\text{res}}$
3. Estimate the sample density $\rho_s$ as the number of samples devided by the sampled surface area
4. For each mutant clade $m$ do
   a. obtain the frequencies (cancer cell fractions) $f_i^m$ of mutation $m$ in all samples $i$ of the clade
   b. compute the dense packing radius $r_m = \sqrt{\sum_i f_m^i/(\pi \rho_s)}$
   c. compute the dense packing dispersion

$$\sigma_m^{\text{dense}} = 128/(45\pi)r_m \tag{3}$$

   d. compute pairwise distances $d_{ij} = |\vec{p}_i - \vec{p}_j|$ and weights $w_{ij}^m = f_i^m f_j^m$
   e. compute the relative dispersion $\sigma_m = (\sum_{j,i<j} w_{ij}^m d_{ij} / \sum_{j,i<j} w_{ij}^m) / \sigma_m^{\text{dense}}$
5. plot the (unweighted) histogram of the values $\sigma_m$

To compute the mean distance of points on a disk, we consider two points on the disk with distance $r$ (in units ofthe disk radius $R$) at an angle θ. We note that the probability density $\delta(r)$ is proportional to the area of the disk of radius $R$ with points that have a distance $r$ to another point on the disk at angle θ. This area is given by the intersectof the disk to the disk shifted by r in direction θ. This intersect is segmented by the shift-axis and the orthogonalaxis into four identical circle segments of area $4\int_{r/2}^1 \sqrt{1 - x^2}dx$. Considering distance vectors for all θ, the densityfunction $\delta(r)$ is proportional to an additional factor $r$ for the circumference of r-vectors. Normalizing by integration from $r = 0$ to 2 yields

$$\delta(r) = \frac{r}{\pi}\left(\left(4\arctan\left(\frac{\sqrt{4-r^2}}{r}\right)\right) - r\sqrt{4-r^2}\right)$$

The mean value of $r$ can then be computed as

$$\langle r \rangle = \int_0^2 r\delta(r)dr = \frac{128}{45\pi}.$$

Frequencies of mutations within single samples and within the whole tumour are crucial to all quantitative analysis in this work. What we call mutation frequencies are cancer cell fractions (CCF) of mutations in sequencing data which estimate the fraction of cells carrying that mutation. Given a sample with variant allele frequency $f_{\text{vaf}}$ (VAF), sample purity $\rho$ and ploidy $p$ we get the mutation's multiplicity $m$ by rounding $u = f_{\text{vaf}} \cdot (p \cdot \rho + 2 \cdot (1 - \rho))/\rho$ to the nearest (nonzero) integer and the cancer cell fraction $f = u/m$. The whole-tumour frequency of a mutation is its tumour wide cancer cell fraction, computed as the average over samples and weighted by the sample purity (assuming equal sample sizes) $\bar{f} = \sum_i f_i \cdot \rho_i/\sum_i \rho_i$ (*Dentro et al., 2017*).

To put the algorithm in words, in (1-2) we consider clades of mutations above a whole-tumour frequency cutoff $f_{\text{res}}$. For a given clade defined by mutation $m$, we measure the spatial distance $d_{ij}$, in step 4d between pairs of samples i,j that contain $m$ and assign a weight that estimates the number of single cell pairs that carry $m$ between both samples. The absolute dispersion of mutation $m$ is defined as the weighted average over these pairwise distances. But different mutations have different whole-

tumour frequencies and would naturally take up areas of different sizes. We therefore measure dispersion relative to the value of dispersion a clade would have if all cells were densely packed in a disk, as given by step 4e. The dispersion for dense packing can be computed as the mean distance between all points on a disk of radius $r_m$ by equation (*Equation 3*) in step 4c.

As in the algorithm for distinguishing surface from volume growth (see Appendix 2), every mutation contributes with equal weight to the computed distribution, here the distribution of dispersion $\sigma_m$. Summary statistics of the dispersion parameter $\sigma_m$ such as the mean over mutations, or its distribution over mutations show the extent of dispersion and cell migration within the population. To compare dispersion distributions observed in empirical data to those seen in spatial simulations under comparable sampling in the surface and volume growth regimes, we use the (approximate) two-sample Kolmogorov-Smirnov test between the experimental and the simulated distribution. The Kolmogorov-Smirnov (KS) statistic is the maximum distance between the two cumulative distribution functions (cdf) $F$, $D_{n,m} = \sup_\sigma |F_{1,n} - F_{2,m}|$ with the number samples $n,m$ of the two empirical distributions respectively. $D$ is a measure of how distant the two distributions are from one another and can be used two rank pairs of distributions by similarity. When normalized by the factor $\sqrt{nm/(n+m)}$ and given a rejection level $\alpha$ (here we use the default $\alpha = 0.05$), a p-value can be computed from the statistic $D_{n,m}$ which states how likely it is that the empirical distributions are sampled from the same underlying distribution - that is, a low p-value indicates that the two underlying distributions are significantly different.

The choice of mean pairwise distance as a measure of dispersion is conceptually simple, but other choices are also possible. Similar measures such as the radius of gyration (root mean square radial displacement) $\sigma_m^{\text{gyr}} = \sum_i f_i^m |\vec{p}_i - \langle\vec{p}\rangle_m|^2 / \sum_i f_i^m$ (around the clade center of mass $\langle\vec{p}\rangle_m = \sum_i f_i^m \vec{p}_i / \sum_i f_i^m$) yield the same qualitative results and their dispersion under dense packing can readily be calculated as an expectation value over the disk.

## 4.2 Processing of whole-exome sequencing data

Fastq files of the sequences were downloaded from https://bigd.big.ac.cn/search/?dbId=gsa&q=PRJCA000091&page=2; *Ling et al., 2015*. Fastq files were processed using our in-house mapping pipeline based on GATK4 (version 4.1.0.0 from DockerHUB). We used GRCh37.p13 as the reference genome and gencode.v19 as the annotation file. Singletons, and reads with less than 70 bp were discarded. Additionally, we excluded known variants in the human population (taken from https://data.broadinstitute.org and https://ftp.ncbi.nih.gov/). We used the interval bed file S02972011 from SureSelect all human exon version 3. Variant calling and initial filtering was performed with Mutect2 (GATK), using the joint variant caller mode and applying the optional filter.

We focus on the high-depth WES data instead of the set of genotyped samples for two reasons: Firstly, because dispersion, unlike the directional growth bias and the turnover, is based on individual mutations alone, we do not need to infer clones or parent-offspring relationships and can effectively leverage the sample frequencies from whole-exome sequencing. Secondly, the distribution of dispersion values turns out to be very noisy for the set of 35 mutations probed by genotyping in the high-spatial-resolution data. However, we have measured the dispersion parameter for this small set of mutations both in the WES samples and the genotyped samples and found that their respective dispersion distribution shows the same large tail as the dispersion distribution of the 217 subclonal SNVs detected by Mutect (see main text, *Figure 3*).

To compute the dispersion parameter for the WE-sequencing data of Ling et al., we call mutations using Mutect 2 which jointly calls on all 23 samples. We obtain 238 subclonal SNVs by imposing the following filters: a given SNV, (1) passes Mutect's filters, (2) has no supporting ALT reads in the normal sample, (3) has 5 or more supporting reads in total, (4) has a total of 150 or more reads (coverage), and (5) has a whole-tumour frequency (CCF) $\bar{f}_m$ between 0.025 and 0.3 (subclonal). The criteria (2–4) are adopted from the variant calling by Ling et al. (2 and 3) eliminate false positives, (4) assures accuracy in the estimation of mutant frequencies while at the same time also reducing the effective genome size (see main text, 'Cell turnover'). Setting a lower bound on the number of supporting alternate reads, filter (3) determines the resolution of mutant frequencies.

Regarding the cutoffs on whole-tumour frequencies (5), the CCF-sprectrum consists of two clearly distinct clusters: Clonal mutations forming a large gaussian peak near $\sqrt{r_1^2 + r_2^2 - 2r_1r_2\cos(\theta)}$ and subclonal neutral mutations at low frequencies following a power-law decay, with very few mutations falling around $\bar{f}_m = 0.5$ inbetween the two profiles. We found that imposing an upper cutoff at $\bar{f}_m = 0.3$ fully removes clonal mutations while recovering most subclonal mutations. The distribution of $\sigma$ turns out

to depend on the minimum whole-tumour frequency of mutations, therefore, we also need to set a cutoff on $\bar{f}_m$ at the lower end of the SFS CCF-spectrum in order to compare with simulations. This dependence is linked to the size of the tumour when specific mutations arose, see Appendix 7. The distribution of σ turns out to depend on the minimum whole-tumour frequency of mutations, therefore, we need to set a cutoff on $\bar{f}_m$ at the lower end of the SFS in order to compare with simulations. This dependence is linked to the size of the tumour when specific mutations arose, see Appendix 7.

We choose a minimum whole-tumour frequency of $f_{\mathrm{res}} = 1/40$, because this is the frequency in the SFS below which the number of detected mutations drops rapidly, as can be seen from the saturation in the cumulative SFS (see *Appendix 5—figure 2*). The cumulative SFS counts the number of mutations with frequency larger or equal to $f$ as a function of $1/f$. It is monotonically increasing and would extend to the smallest mutation frequency but effectively flattens below the sequencing resolution $f_{\mathrm{res}}$. Beyond $f_{\mathrm{res}} = 1/40$ the sequencing disproportionately misses low frequency mutations which would be relevant to the comparison with simulations at cutoffs lower than $f_{\mathrm{res}}$ for the dispersion parameter.

### 4.3 Dispersion in two dimensions

We compare the dispersion parameter in the data of Ling et al. to simulations of populations growth to 40,000 cells under volume growth ($\rho = \infty$) and surface growth ($\rho = 6$) which we sample to mimic the spatial sampling of *Ling et al., 2015* as described in Appendix 1.2. For each simulated 3D tumour, we take a planar cross-section through the centre of mass and cover the plane with a triangular lattice that fits 23 samples of 20 cells each at its nodes. To mimic stochastic effects in sequencing, in each sample and for each mutant we draw a binomially distributed number of alternate reads given the true frequency and a read depth of 20 reads. These choices follow from the sequencing depth in the WES data of Ling et al., which has a mean coverage of 20 reads per sample. In the simulations, samples consist of 20 cells, which allows for a sequencing depth that is sufficiently high to resolve whole-tumour mutation frequencies even below 1/40. In fact, given the limited sequencing depth we found that increasing the sample size has no effect on the distribution of dispersion parameter.

We compute whole-tumour frequencies $\bar{f}_m$, retain mutants with $\bar{f}_m > 1/40$ - the cutoff used for mutations in the WES data of Ling et al. and determine their dispersion distribution as described above.

The normalized Kolmogorov-Smirnov statistic and log-p-value between the data and simulations in 3D with sampling (*Appendix 4—figure 3*) are 5.742 and -65.3 for surface growth and 1.13 and -1.9 for volume growth, confirming that the volume growth regime fits the observed distribution of dispersion parameters much better than the surface growth regime. (A perfect fit, leading to a KS statistics of zero, is not expected, for instance because of fluctuations due to the finite number of mutations.).

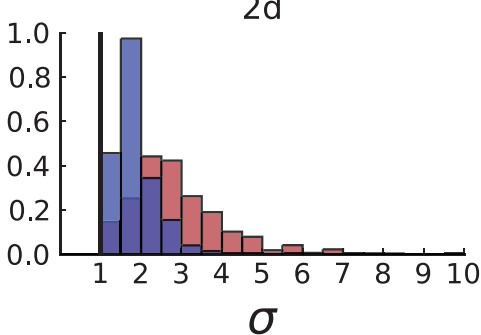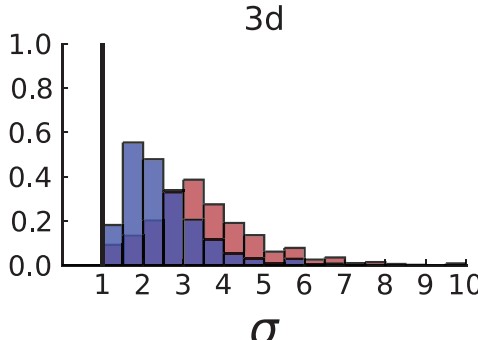

**Appendix 4—figure 1.** Dispersion distributions in simulations including spatial sampling — 2D (left) and 3D (right). Histograms of the dispersion parameters under spatial sampling with 23 samples with a sequencing model mimicking the whole-exome sequencing of Ling et al. Samples are taken from populations grown to 10,000 cells in 2D (left) and cross-sections of 3D simulations of populations grown up to 40,000 cells (right). As in Appendix 2, we use the cell division rate $b = 1$, mutation rate $\mu = 0.3$, as well as cell death rates $d = 0.8$ and $\rho_c = 6$ for volume growth (red) and $d = 0.4$ and $\rho_c = \infty$ for surface growth (blue), respectively (see also *Appendix 2—figure 3*).

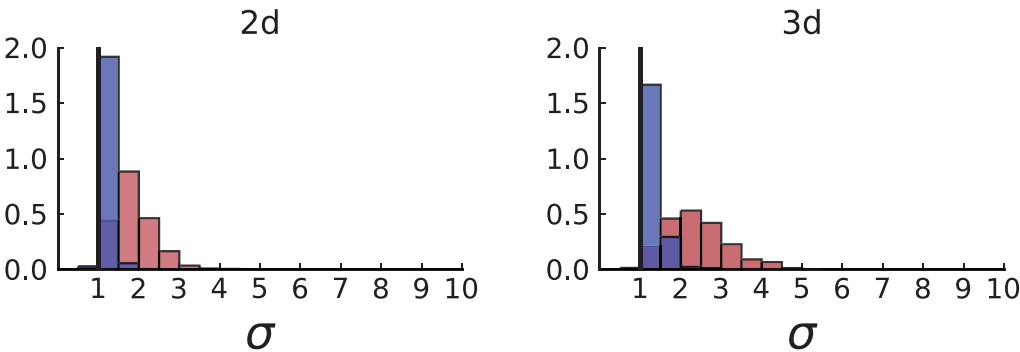

**Appendix 4—figure 2.** Dispersion distributions in simulations with sampling of single cells — 2D (left) and 3D (right). Same as in *Appendix 2—figure 2* but for single cells under volume (red) and surface growth (blue) at the same parameters $b, d, \mu$.

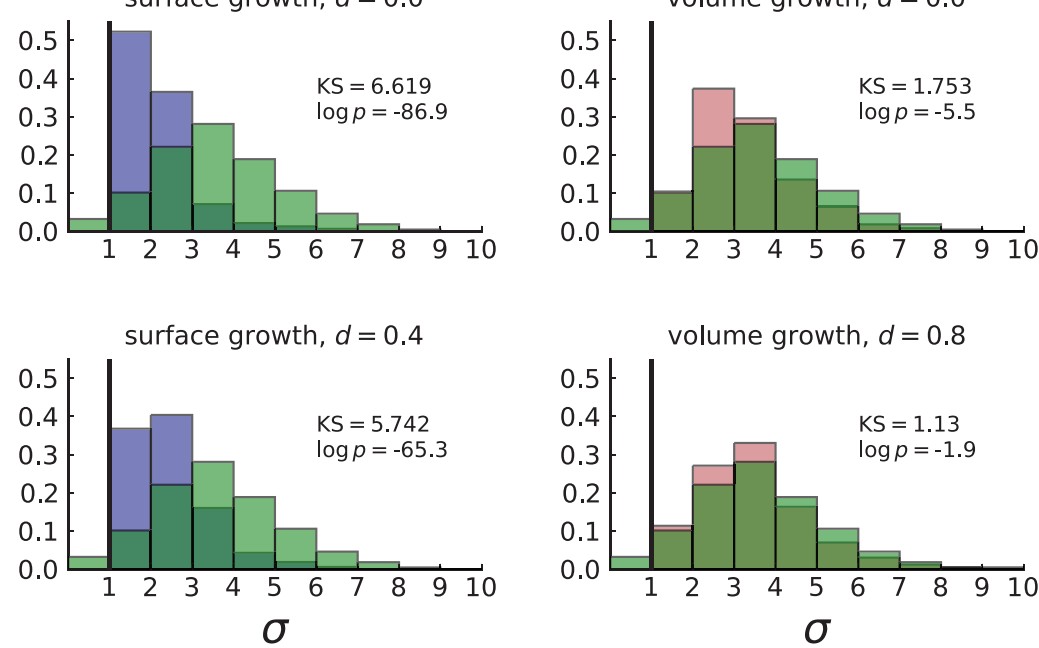

**Appendix 4—figure 3.** Normalized Kolmogorov-Smirnov distance between dispersion distributions of Ling et al. data and simulations. Each frame compares the histogram of the dispersion values σ of subclonal mutations in the WES data (in green) to that of simulations in 3D with spatial sampling under volume (red) and surface (blue) growth at low and high turnover rates $d$. Simulation settings are as in *Appendix 4—figure 1*, right. The KS statistic serves as a measure of distance between the two distributions with smaller values indicating that the data is more likely to match the model. The dispersion observed in the data fits with simulations of volume growth at a high rate of turnover (**KS = 1.13**, bottom right).

The same methods apply for uniformly sampled single cells ($f_m^i = 1$), see *Appendix 4—figure 2*, and for the sampling scheme used for *Appendix 4—figure 1*. However, simulations show that the absolute scale of dispersion $\sigma_m$ strongly depends on sampling density, namely, the value of mean dispersion (in the volume growth limit) increases as the sampling density gets reduced. It is therefore import to reproduce the sampling settings when comparing simulations to sequencing data.

## 4.4 Dispersion in three dimensions

Just like the measure of the direction of mutants in Appendix 2.2, the dispersion can also be computed for samples taken in three dimensions. Only the mean distances of cells in a densely packed sphere $\sigma_m^{\text{dense}}$ is different from that of the two-dimensional disk used in steps 4(b) and 4(c): Given a 3D sample density $\rho_s$, a mutant occurring with frequencies $f_m^i$ in samples $\{s_i\}$ could be packed into a sphere of radius $r_m = \left( \sum_i f_m^i / (4/3\pi) \right)^{1/3}$. The mean distance of points within this sphere is $\sigma_m^{\text{dense}} = 36/35 r_m$. To see this, consider two points x and y within the sphere of radius R on shells of radius $r_1 < R$ and $r_2 < R$ and with an angle $\theta = \angle(x, y)$. The distance between the two points is $\sqrt{r_1^2 + r_2^2 - 2r_1 r_2 cos(\theta)}$, so the average distance for points within the sphere becomes,

$$\langle |\mathbf{x} - \mathbf{y}| \rangle = \int_0^R dr_1' f_{r_1}(r_1') \int_0^R dr_2' f_{r_2}(r_2') \int_0^\pi d\theta' f_\theta(\theta') \sqrt{r_1'^2 + r_2'^2 - 2r_1' r_2' cos(\theta')} \,,$$

where the density functions $f_{r_1}(r') = 3r'^2/R^3$, $f_{r_2}(r') = 3r'^2/R^3$ are proportional to the surface area of the sphere with radius $r'$ and the density function $f_\theta(\theta') = \sin(\theta)/2$ is proportional to the circumference of the circle defined as the rim of the cone with side length $\max(r_1, r_2)$ and opening angle $\theta'$. The inner integral yields

$$\int_0^\pi d\theta' \, \sin(\theta')/2 \sqrt{r_1^2 + r_2^2 - 2r_1 r_2 cos(\theta)} = \frac{1}{6r_1 r_2}((r_1 + r_2)^3 - |r_1 - r_2|^3),$$

and the outer integrals can be computed by distinguishing the two cases $r_1 < r_2$ and $r_2 < r_1$ resulting in

$$\langle |\mathbf{x} - \mathbf{y}| \rangle = \frac{36}{35}R.$$

Based on the corresponding definition of the dispersion parameter, we measure the dispersion of mutants in the whole-genome sequencing (WGS) data of Li et al. The data consists of 16 samples from tumour T1 and 9 samples from tumour T2. The cumulative SFS reaches a plateau at frequency $f_{\text{res}} = 1/30$ for T1 and 1/18 for T2, which is the respective whole-tumour frequency resolution, as discussed for the Ling et al in the previous Appendix 4.2. The frequency resolution is lower than in Ling et al., in agreement with the expected linear scaling between the inverse frequency resolution and the number of samples $n$ which can be written as $(nf_{\text{res}})_{\text{Li}} = (nf_{\text{res}})_{\text{Ling}}$ given that both studies sequenced at almost the same read depth per sample.

Whereas in the case of the tumour data of Ling et al., samples were taken nearly spatially uniformly for subsequent deep sequencing analysis, in the two tumours T1 and T2 only few of the slices were used for WGS. To account for this uneven sampling we use the same sample positions in simulated 3D tumours: For each simulation we take slices at specified heights and sample each slice at the positions of the WGS samples from tumours T1 and T2. The resulting distributions of dispersion values for mutants for T1 and the corresponding numerical simulations are shown in *Appendix 4—figure 4*. The dispersion values observed for tumour T1 are high compared surface growth simulations (as in the 2D data by Ling at al.), and the KS statistics is not compatible with surface growth, see *Appendix 4—figure 4*.

In the case of T2, there are insufficient mutations that appear in a sufficiently large number of samples. Out of 150 mutations in T2, only 2 appear in more than 4 samples. (Out of 308 mutations in T1, 58 appear in more than 4 samples.) The dispersion parameter becomes unreliable when the number of samples involved is low; a few samples from a spherical distribution of mutants typically yield artefactual values of σ higher than one, due to the sparse and uneven distribution of samples in space.

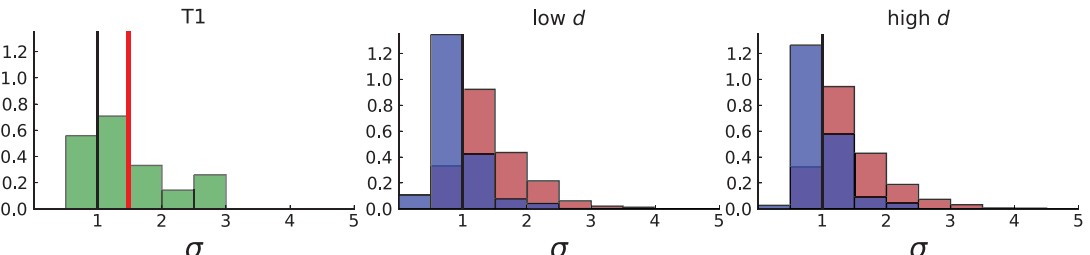

**Appendix 4—figure 4.** Dispersion distribution of tumour T1 and T1-like sampling in simulations. The left plot shows dispersion values of 308 mutations with frequencies larger than the frequency resolution $(f_\text{res})_\text{T1} = 1/30$ based on the 16 WGS samples of T1. The plots on the right show dispersion in simulations of 3D spherical tumours under T1-like sampling, meaning that, from each simulation 16 samples are taken from 3 slices, where the positions of slices and samples are specified by T1. Parameters of the 3D simulations are as in *Appendix 4—figure 1*. The normalized KS statistic and log-p-value between the data and simulations are 7.272 / 7.267 and -105.1 / -104.9 for surface growth (low/high $d$) and 1.652 / 2.189 and -4.8 / -8.9 for volume growth (low/high $d$).

# Appendix 5

## Cell turnover

We infer the cell death rate and mutation rate by tracking the dynamics of clades and clones. A clone describes the set of cells with a particular genotype. In the absence of back mutations, a clade describes the set of cells carrying a particular mutation. (With back mutations, a clade is the set of descendants of a particular mutant cell, including the original mutant itself.) Under (volume) exponential growth, a clade is expected to grow exponentially at the growth rate $\lambda = b - d$ and maintain on average a constant frequency within the tumour (*Durrett, 2013*). A clone grows in a similar way, but at a rate which is reduced to mutations which generate new clones.

However, clone and clade sizes are also subject to fluctuations due to genetic drift: Clones and clades can become extinct even in growing populations, and the rate at which that happens depends on the rate of cell death (relative to the rate of birth) and the mutation rate. We use the fraction of clones with extinct parental clones and the fraction of offspring clades which coincide with a given ancestral clade to infer both the mutation rate and the relative rate of cell death.

Any mutation we encounter when tracing back the lineage of an offspring clade constitutes an ancestral clade. A clade coincides with an ancestral clade if all clones within the ancestral clade are also part of the offspring clade (a clone carrying mutations $m_i$ is part of every clade $i$ defined by mutation $m_i$). Under an infinite site model, this situation refers to a pair of coinciding mutations $m_1, m_2$, i.e. any cell carrying $m_1$ also carries $m_2$ and vice versa - $m_1$ and $m_2$ are the branching points within the lineage from which ancestor and offspring clade originated, respectively. A clone with extinct parental clone necessarily implies a clade that coincides with its most recent ancestral clade. Clades however can coincide with several ancestral clades if these underwent successive sweeps and ended in the fixation of the offspring clade. So more generally, for $n - 1$ ancestral clades coinciding with the offspring lineage there is a set $m_{i=1}^n$ of mutations that is common to all cells carrying any of the mutations $m_i$.

We begin with a small but illustrative example of clone and clade turnover in Appendix 5.1 before introducing the turnover based inference scheme in Appendix 5.2. We then test this scheme on simulated data (also under spatial sampling) in Appendix 5.3. For a derivation of analytical expressions and a discussion of the inference model used in this section, see Appendix 5.4 below and (*Angaji et al., 2021*).

## 5.1 Clone and clade turnover example

We consider a population in which mutations 1, 2, 3, 4, 5 arose in that order (the order is given only to illustrate the example, see *Appendix 5—figure 1*). In this population, clones carrying mutations $\{\}, \{1\}, \{1, 2, 3, 4\}, \{1, 2, 3, 4, 5\}$ survived. Clone $\{1\}$ has an extant parental genotype, namely $\{\}$. None of the extant genotypes carry exactly three mutations out of $\{1, 2, 3, 4\}$. The parental clone of this genotype thus became extinct. Finally genotype $\{1, 2, 3, 4, 5\}$ has an extant parent in $\{1, 2, 3, 4\}$.

Hence 1 out of 3 clones have extinct parents, resulting in a clone turnover of $W_c = 1/3$. In order to compute the clade turnover we go over all mutations 1, 2, 3, 4, 5, which each define a clade. The only ancestral clade of mutant 1 is the original clade which gave birth to all clades. The clone $\{\}$ survived, therefore clade 1 does not coincide with its ancestor. Mutation 1 hence contributes one to the denominator and zero to the numerator of the clade turnover. Mutation 2 does not coincide with either of its two ancestral clades (origin and 1) and thus adds two to the denominator and zero to the numerator.

The clade of mutation 3 is necessarily also distinct from the ancestral clades of mutation 2 - however, its clade has replaced that of 2 and therefore adds three to the denominator and one to the numerator. Note that the clone $\{1, 2, 3\}$ does not need to be extant; as long as clade 3, which includes its subclades 4 and 5, replaces clade 2 it adds to the turnover. The clade of 4 in turn coincides with both 2 and 3, resulting in a contribution of four to the denominator and two to the numerator. Lastly, clade 5 has the original clade and mutants 1, 2, 3, 4 as ancestral clades and coincides with neither of them, adding five to the denominator and zero to the numerator. The clade turnover is hence $W_l = 3/15$.

In this example we assumed that we know the order in which the mutations occurred, but in fact the clade turnover can be computed without knowledge of the phylogenetic tree - we refer to *Angaji et al., 2021* for the tree-free algorithm.

## 5.2 Algorithmic description

We consider a population of cells dividing at rate $b$ and dying at rate $d$. The population starts from a single cell and the dynamics lasts for a time period $T$. With probability $\mu$ a mutation occurs in one of the daughter cells at division. The rates of birth and death are the same for all cells, meaning that the population is under volume growth (Appendix 1, $\rho_c = \infty$).

Extinctions are not *directly* metric from a single snapshot. Clones and clades that go extinct only leave evidence that they existed through their mutant offspring. We calculated the probability for a clone to have its parental clone become extinct (*Angaji et al., 2021*),

$$\text{clone turnover}: \quad W_o(d/b, \mu, bT) = \frac{\mu(d/b + (\mu/2)^2)}{(1 - \frac{\mu}{2})^2(1 - d/b - \frac{\mu}{2})} \frac{1 - e^{-2(1 - d/b - \mu/2)bT}}{1 - e^{-2\frac{\mu}{2}bT}} \,, \tag{4}$$

the probability for a clade to replace an ancestral clade,

$$\text{clade turnover}: \quad W_a(d/b, N) = \frac{d/b}{2 \log N}(1 - \log(N)^{-2}) \,, \tag{5}$$

see Appendix 5.4 for a brief derivation. The clone turnover $W_o$ depends on the mutation rate, as mutations lead to the establishment of new clones. The clade turnover, however does not depend on the mutation rate. From the empirically measured turnover of clones and clades it is thus possible to infer the mutation rate and the rate of cell death (relative to the growth rate). The model assumes exponential growth and neutral mutations in a infinite site scenario. We infer clones by a simple clustering of coinciding mutations (see Appendix 8) as was done in Appendix 2. Again we find no significant changes when using LICHeE (*Popic et al., 2015*) instead.

The computation proceeds as follows:

1. estimate the population size as $N = n^{3/2} \frac{2}{\pi^{1/2}3^{1/4}} \frac{1}{2}$ given the number of samples in the plane $n$
2. clade turnover:
   a. remove mutations that occur in less than 2 samples from clones
   b. compute the fraction $\hat{W}_a$ of pairs where offspring clade and ancestor clade coincide
   c. Solve $\hat{W}_a = W_a(d/b, N(1 - d/b))$ for $d/b$
3. clone turnover:
   a. remove clones that have mutations that occur in less than 2 samples
   b. compute the fraction $\hat{W}_o$ of clones with missing parental clone
   c. Solve $\hat{W}_o = W_o(d/b, \mu, \log(N(1 - d/b))/(1 - d/b))$ for $\mu$
4. optional: subsample a fraction $0 < L < 1$ of mutations and repeat from (4) to infer $d/b, \mu$ and compare the results for different values of $L$

**Step 1** Both clade and clone turnover depend on the effective population size $N$, which depends on the resolution frequency as well as relative cell death rate $d/b$ through *Equation 28*, as discussed in detail in Appendix 7. To put the points from Appendix 7 briefly: $N$ depends on the inverse frequency resolution $f_{\text{res}}$ which in turn relates to a set frequency cutoff within the sampled plane through *Equation 27*. The dependence of $N$ on $d/b$ is due to the stochastic extinction of clades from fluctuations in population size. Extinction is frequent at high rates $d/b$ and increases the expected population size of a surviving population by a factor $1/(1 - d/b)$. This $d/b$ dependence of $N$ is therefore included in the inference steps 2c and 3c.

**Steps 2a and 3a**. We set the frequency cutoff to $2/n$, where $n$ is the number of samples, by demanding that a mutation typically occurs in 2 or more of the $n$ genotyped samples. This cutoff is required to avoid cases where clones that are not sampled contribute erroneously to the turnover. The effective population size $N$ then is the size of the tumour at which, under neutral evolution, mutations occur which can just be recovered in the genotyped samples. Mutations that are found in only one sample thereby typically occurred later in time, after the population reached size $N$, and do not contribute to the observable measures of turnover. These mutations are removed from all clade genomes (2a) and their clones are removed from the set of clones. Plugging the frequency cutoff $2/n$ into *Equation 27* gives the expression for $N$ in step 1.

**Steps 2b and 3b**. Here we calculate clade and clone turnover. See Appendix 5.1 for an example of how to determine clade- and clone-turnover from the set of surviving clones. We again refer to *Angaji et al., 2021* for more detailed instructions.

**Steps 2c and 3c**. The clade and clone turnover calculated from the data are compared to the analytical expressions *Equation 5* and *Equation 4* to infer rates of mutation and cell death. First, $d/b$ is inferred from the clade turnover and then used in 3c to obtain the mutation rate $\mu$.

As an additional consistency check, the inferred rates can be used to compare a predicted cumulative SFS to the cumulative SFS obtained from the sequencing data, see also Appemdix 4.2. Under neutral evolution and volume growth the cumulative SFS is linear with slope $\mu/(1 - d/b)$ (*Williams et al., 2016*). *Appendix 5—figure 2* shows a good match between the slope given by the inferred rates $d/b$, $\mu$ and the cumulative SFS of the WES whole-tumour frequencies of Ling et al.

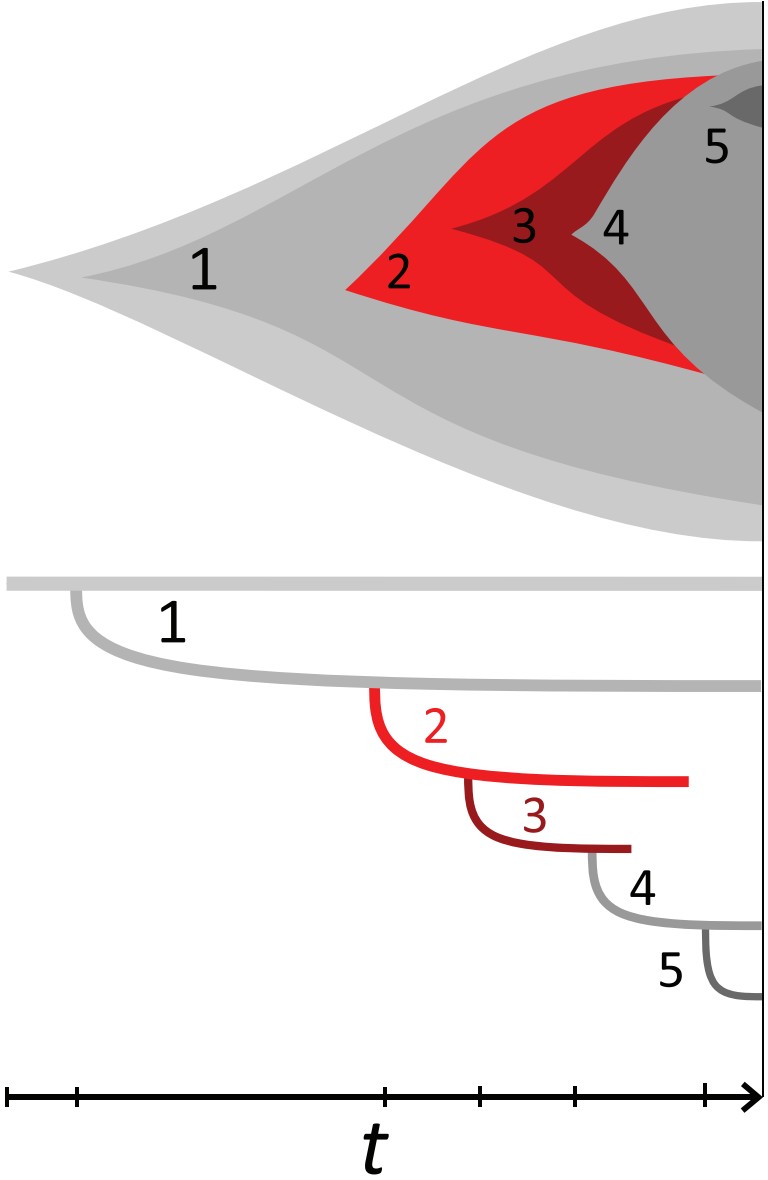

**Appendix 5—figure 1.** Example phylogeny and turnover. This illustration shows a growing population of cells that accumulates 5 mutations, on the top showing the muller plot of the 5 clones' sizes and below the corresponding phylogeny against time t. The growth rate is positive but under stochastic dynamics clones and clades may become extinct. The clones carrying mutations {1, 2} and {1, 2, 3} went extinct and their offspring clones/clades add to the clone/clade turnover, see text. Specifically,the clone {1, 2, 3, 4} lost its parental clone,similarly the clade defined by mutation 3 coincides with clade 2 and clade 4 in turn replaced both clades 2 and 3.

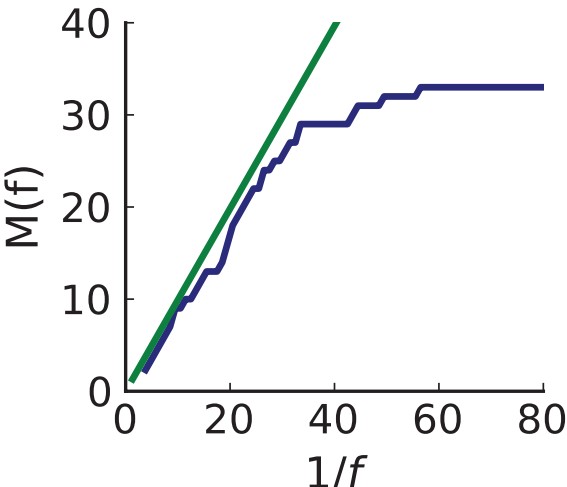

**Appendix 5—figure 2.** Cumulative SFS of WES data compared to prediction using inferred rates $d/b$ and $\mu$. The number of mutations with frequency larger than $f$ in the entire tumour against $1/f$ in genomically-resolved sequencing data (solid blue line). $M(f)$ flattens below the allele frequency corresponding to the sequencing resolution, $f_{res} = 1/40$. The green line indicates the expected curve under neutral evolution for the inferred rates of mutation $\mu$ and cell death $d$ (**Durrett, 2013**) and provides a consistency check on our inference. Its slope is the scaled mutation rate $\mu/(1 - d/b)$.

In Appendix 7 we test how the choice of the population size $N$ affects the inference of mutation rate and the rate of cell death. We mimick the procedure in Ling et al., where first 23 samples are taken for WE sequencing, and a small set of mutations is then probed by genotyping in a larger set of 285 samples (see **Appendix 5—figure 3**). We find that (1) the number of samples $n$ required for estimating $N$ is given by the size of the large set of genotyped samples which determines the clonal composition, and (2) setting the lower cutoff of 2 supporting genotyped samples on mutations avoids biasing the turnover towards high death rates. In conclusion, the spatially-resolved set of the Ling et al. data reconstructs the clonal composition of the tumour, determines the effective population size $N$, and can be used to determine the rate of cell turnover $d/b$ and the mutation rate across the target genome. The preceding step of sampling at a lower spatial resolution for whole-exome sequencing, on the other hand, controls the size of the mutational target by filtering sites according to their coverage. We use the number of sites that pass these filters to calculate the mutation rate per nucleotide (see main text, 'Cell turnover').

## 5.3 Testing the inference based on clade/clone turnover

Our inference scheme for cell turnover based on stochastic clone extinction does not use spatial information. However, to test how the inference works under spatial sampling, we use data from spatial simulations under volume growth. Neutral evolution of the different clones is assumed, compatible with the original findings of **Ling et al., 2015** and the corresponding (cumulative) site frequency spectrum shown in **Appendix 5—figure 2**.

We test the inference as specified in the steps above both in single cells sampled uniformly and using samples mimicking the sampling scheme in **Ling et al., 2015** as described in Appendix 1. **Appendix 5—figure 3** compares inferred and underlying relative death rates and mutation rates in numerical simulations of volume growth ($\rho = \infty$).

In **Appendix 5—figure 4**, we perform the same comparison, but focus on the regime of a high relative death rate. Specifically, we set the model parameters (relative death rate and mutation rate) close to the values inferred from the data of **Ling et al., 2015** and ask how well the inference scheme can recover those parameters. The average inferred rates match the underlying rates very well, but we also find large fluctuations in individual datasets. The uncertainty in the inferred mutation rate is connected to the high rate of cell death; a small change in $d/b$, say from 0.9 to 0.99 changes the number of generations needed for the cell population to grow by a factor of 10, and in turn changes the inferred mutation rate per generation by the same factor.

Employing LICHeE clustering of clones in simulations yields inferred rates $\mu$ and $d/b$ very similar to those obtained with our simple clustering scheme. As for the analysis of the direction of mutants in Appendix 2, we also compare our inference results for the simple clustering of the Ling et al. data to the results yielded by the LICHeE clustering tool. LICHeE infers a split of clones that is almost identical to our clustering scheme, presumably due to the large number of samples and the low number of mutations. Consequently, turnover inference based on the set of clones by LICHeE results in a very similar, high turnover rate and the median mutation rate falls well into the error margin of inferred mutation rates for the simple clustering.

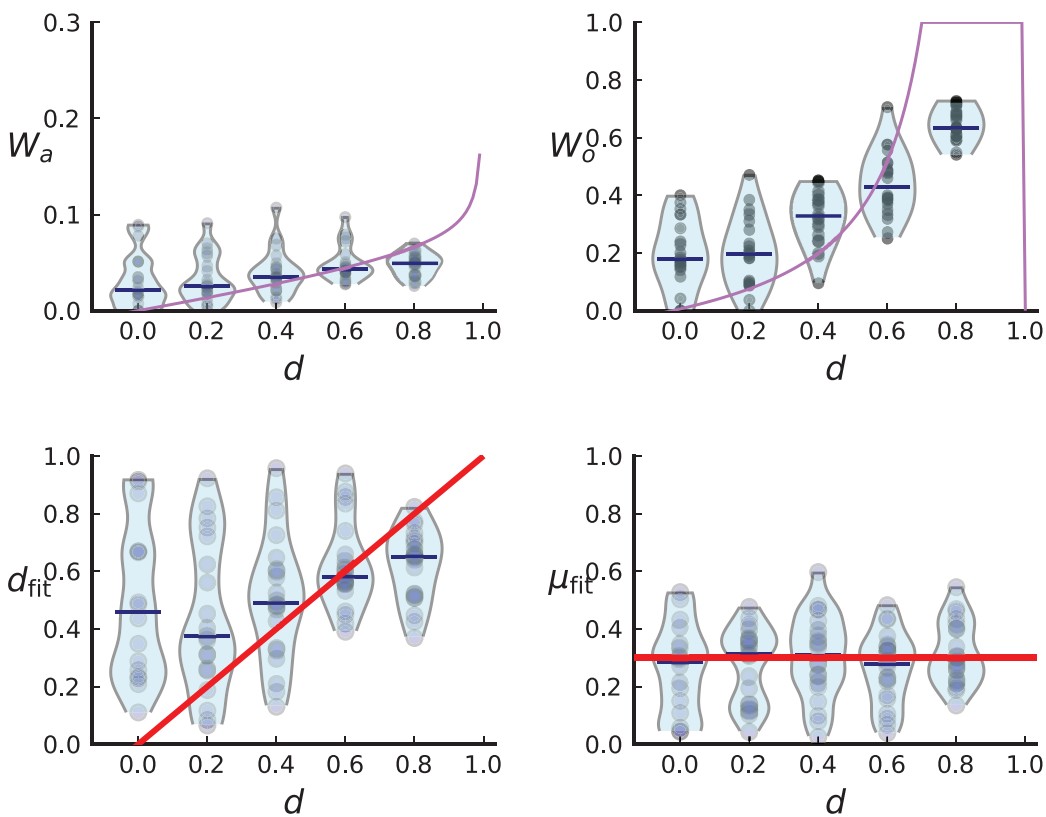

**Appendix 5—figure 3.** Turnover inference for simulated sampling from cross-sections in 3D over a range of death rates. For each choice of death rate $d$ we simulate 20 populations under volume growth in 3D to 40,000 cells each and take a cross-section. Mutations detected in a first set of 3 samples are then probed for in a set of 285 samples and used to infer $d$ and $\mu$.

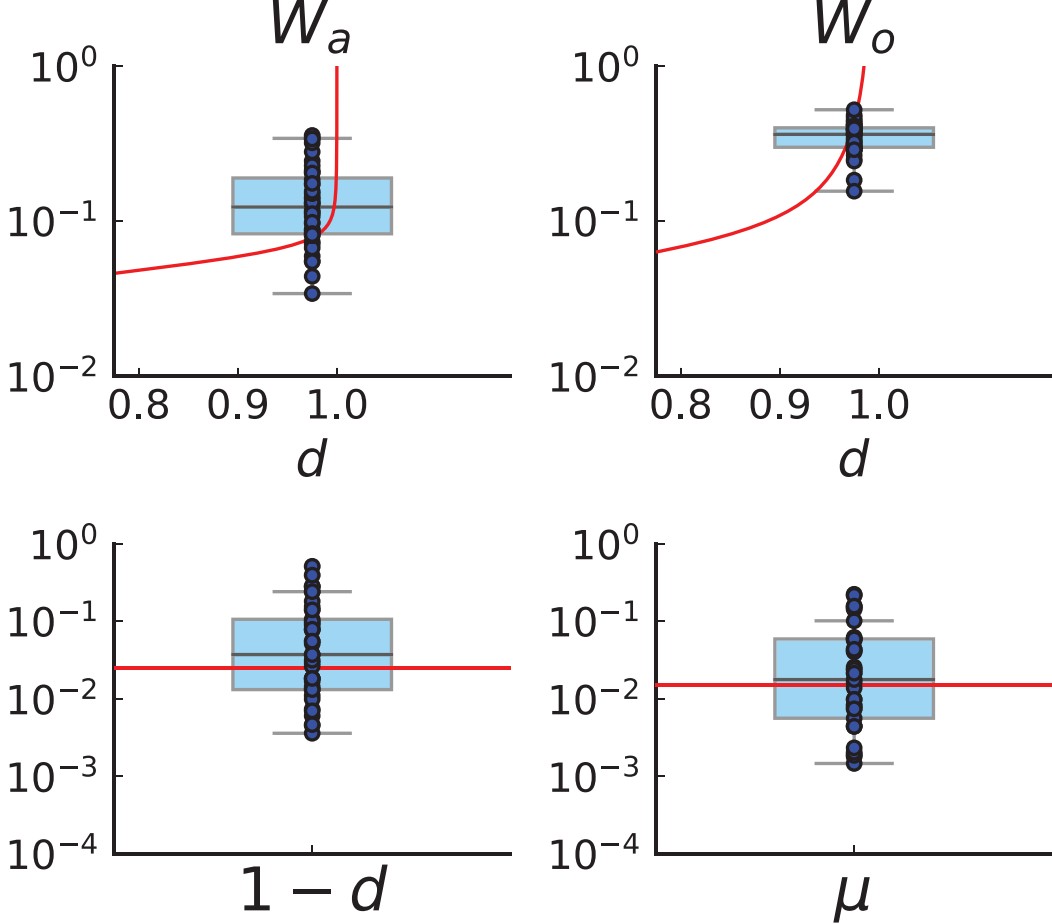

**Appendix 5—figure 4.** Turnover inference for simulated sampling in 3D at the rates $d/b$ and $\mu$ compatible with those inferred from the Ling et al. data. We run 40 simulations in 3D to 40,000 cells at the rates inferred for the Ling et al. tumour, $d = 0.975$ and $\mu = 0.015$. Spatial sampling is performed as explained in Appendix 1.2 and *Appendix 5—figure 3*. We plot the measured clade and clone turnover, $W_a$ (top left) and $W_o$ (top right), with the theoretically expected value as a function of d in red, and the inferred rates, $d$ (bottom left) and $\mu$ (bottom right), with the true value of the respective rate in red. The median of inferred rates is $d = 0.949$ and $\mu = 0.024$, but there are substantial fluctuations around these median rates.

We also apply the turnover-based inference to the tumours T1 and T2 analyzed by Li et al. and test on simulations with analogous spatial sampling schemes. The results are shown in *Appendix 5—figure 5*. Clones of T1 and T2 were determined using the LICHeE tool for clone inference in multi-region sequencing data, see Appendix 8.

The turnover theory described in Appendix 5.4 assumes at most one mutation per cell division. The number of mutations validated by Li et al. is very large compared to Ling et al. (906 mutations for T1 and 565 mutations for T2), resulting in a genomic mutation rate that exceeds this limit. This situation can be easily dealt with by taking subsets of the validated mutations of relative size $L$, effectively shrinking the genome size by factor $L$. However, we find that the inferred mutation rates do not scale linear with $L$ - despite showing a downward trend with decreasing $L$ as required by the dependence of the clone turnover on the mutation rate. Only the inferred death rate $d$ is constant as expected.

To investigate, we use numerical simulations and again imitate the sampling of the two tumours in our 3D spatial simulations for different mutation rates, see *Appendix 5—figure 6*. The inference scheme fails to infer the true rates in simulations when applying the sampling procedure of T1 and T2, whereas the planar sampling scheme of Ling et al. accurately infers $\mu$ (*Appendix 5—figure 3*) and maintains the proper scaling in simulations when reducing the mutation rate by subsampling

of mutations. We conclude that the samples in 3D data of Li et al. are not sufficiently dense to characterize the loss of clones with an accuracy sufficient for inference.

We suspect that the method of turnover-based inference breaks down in this case due to the low resolution of the clonal composition, which was particularly noticed for T2 in the previous sections. For both tumours, samples are taken from several slices of the upper hemispheres. The sampling covers the hemispheres rather uniformly, therefore, the estimate of the effective number of samples in the whole sphere $\tilde{n}$ is simply $2n$ for $n$ samples in the hemisphere. Using the same minimal frequency $f_{\min} = 2/n$ as before, the effective population size in step (1I) of the turnover computation becomes $N = 2n\frac{1}{2} \approx 160$ for both tumours. This shows that taking a number of samples from the sphere that is comparable to that in the cross section implies a much lower sampling density. The low density of samples, however, in turn makes it harder to accurately infer clones their ancestry needed for inference.

An issue that goes beyond the scope of this work is the turnover inference if one or more lineages are under selection. Under selection the turnover theory no longer applies to the full population but needs to be modified to apply to individual lineages with different birth and death rates. It would be straightforward to include the effects of selection by defining such clade-specific turnover metrics, but this is outside the scope of this paper. We note that both T1 and T2 have an SFS that deviates to some degree from neutrality exhibiting potential clusters of mutations, which may be due to selection (*Li et al., 2022*).

## 5.4 Brief derivation of clade and clone turnover

The probability of eventual extinction can be obtained by first-step analysis, see *Angaji et al., 2021* for details. Consider a single cell whose descendants form a clone or clade. The dynamics of clones and clades can be described by the same master equation by defining the effective rates of cell birth and death $\alpha$ and $\beta$. Clades gain cells at rate $\alpha = b$ and lose cells at rate $\beta = d$. But clones only gain a cell if neither parent nor offspring mutates at division, $\alpha = b \cdot (1 - (\mu/2)^2)$, and additionally lose a cell if both mutate $\beta = d + b \cdot (\mu/2)^2$. The probability that the originating cell and all offspring eventually die is denoted by $q$. After a single small time-step $\Delta t$, the cell either died with probability $p_0 = \beta \Delta t$, neither died nor divided $p_1 = 1 - (\alpha + \beta)\Delta t$ or divided $p_2 = \alpha \Delta t$. In each case the remaining cells each independently produce lineages which eventually go extinct with probability $q$. This leaves us with the following self-consistent equation for $q$

$$q = p_0 + p_1 q + p_2 q^2 \tag{6}$$

which can also be rewritten as $q = \frac{\beta}{\alpha+\beta} + \frac{\alpha}{\alpha+\beta}q^2$ in accordance with a time discrete branching process with probability of cell birth $\frac{\alpha}{\alpha+\beta}$ and death $\frac{\beta}{\alpha+\beta}$ (*Durrett, 2015*, chapter 3).

This equation is readily solved by $q = \min(1, \frac{\beta}{\alpha})$, which is

$$\text{for clades} \quad q = \frac{d}{b}, \quad \text{and for clones} \quad q = \frac{d + b(\frac{\mu}{2})^2}{b(1 - \frac{\mu}{2})^2}. \tag{7}$$

Given the probability $p_n(t)$ of a clone/clade having size $n$ at time $t$, the probability that this clone/clade will eventually die out is

$$\sum_{n=0}^{\infty} q^n p_n(t) = z(q, t), \tag{8}$$

which is also the definition of the generating function $z$ of the time-dependent probability mass function $p_n(t)$ at $q$. Instead of finding a solution for $p_n(t)$, we can obtain moments of the size distribution by taking derivatives of the generating function.

We begin by formulating the master equation governing the size of exponentially growing clones and clades. Consider a population that starts from a single cell and grows for a time $T$. Clones appear at different times and have different running times during which they can produce offspring. The probability that a cell was born at time $t$ is proportional to the population size $N(t) = e^{\lambda t}$. The normalising factor is $(e^{\lambda T} - 1)/\lambda$, yielding for large final times $T$ an exponential distribution $\lambda e^{\lambda(t-T)}$ for the running times $\tau \equiv T - t$.

$$\partial_t p_n(t) = \alpha(n-1)p_{n-1}(t) - (\alpha+\beta)np_n(t) + \beta(n+1)p_{n+1}(t) - \lambda p_n(t) . \tag{9}$$

In addition to the first terms describing a standard birth-death process, the master equation also contains a term $-\lambda p_n(t)$ describing the exponential decay of available running time. The clone's founding cell is here born at time $t = 0$. Using the definition *Equation 8*, we get a linear partial differential equation for the generating function

$$\partial_t z(q,t) = (\alpha q^2 - (\alpha+\beta)q + \beta)\partial_q z(q,t) - \lambda z(q,t) \tag{10}$$

which for the boundary condition $z(q,0) = q$ is solved by

$$z(q,t) = e^{-\lambda t}\frac{(1-q)\beta - e^{-(\alpha-\beta)t}(\beta - \alpha q)}{(1-q)\alpha - e^{-(\alpha-\beta)t}(\beta - \alpha q)} . \tag{11}$$

We defined the clone turnover as the probability for a clone to have its parental clone become extinct, and the clade turnover as the probability for a clade to replace an ancestral clade. In both cases, the offspring appears on a background of size $n$ with a probability proportional to $np_n(t)$, while the background becomes extinct with probability $q^n$. Averaging over both the time since the occurrence of the parental clone $t$ and parent size at birth $n$ defines the turnover parameter

$$W = \frac{\int_0^T dt \sum_{n=0}^\infty np_n(t)(\alpha/\beta)^n}{\int_0^T dt \sum_{n=0}^\infty np_n(t)} = \frac{q\int_0^T dt\partial_q|_{q=\beta/\alpha}z(q,t)}{\int_0^T dt\partial_q|_{q=1}z(q,t)} \tag{12}$$

$$= \frac{\beta}{\alpha}\frac{\lambda - (\alpha-\beta)}{\lambda + (\alpha-\beta)} \cdot \frac{1 - e^{-(\alpha-\beta+\lambda)T}}{1 - e^{(\alpha-\beta-\lambda)T}} . \tag{13}$$

Inserting the respective effective rates $\alpha$, $\beta$ for clones and clades finally yields the clone turnover *Equation 4* and clade turnover *Equation 5*. For the detailed derivation, we refer to *Angaji et al., 2021*.

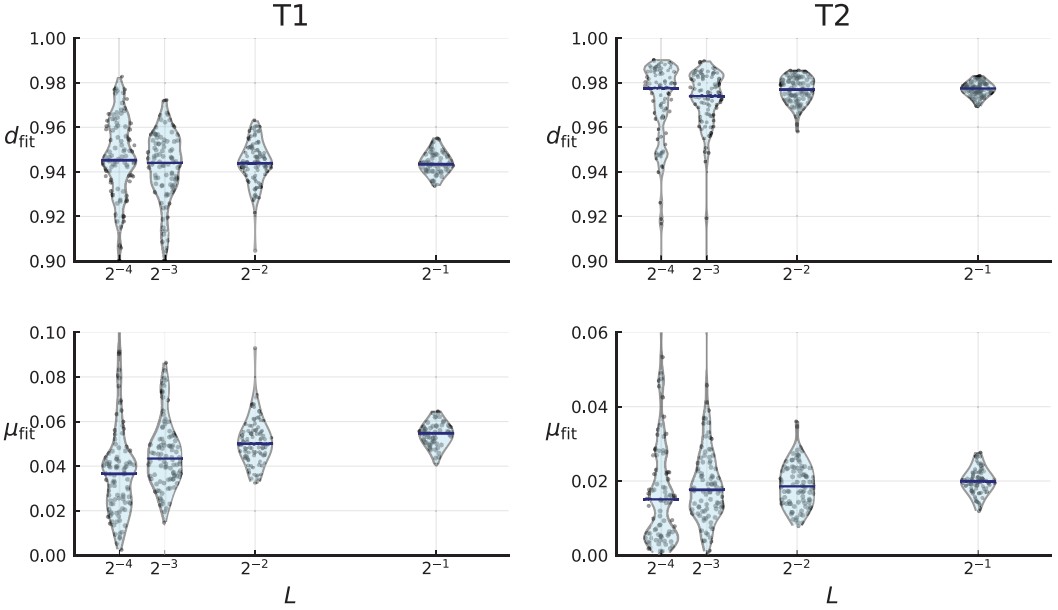

**Appendix 5—figure 5.** Cell death and mutation rates inferred from the clade and clone turnover measured on the clones of T1 (left) and T2 (right). For a given $L$ we repeatedly subsample a fraction $L$ of the mutations, calculate clade and clone turnover and infer $d$ and $\mu$ for each subsampled set of clones (violinplots, individual samples are shown as blue dots). Blue bars indicate the median. Clones of T1 and T2 were determined using the LICHeE tool for clone inference in multi-region sequencing data, see Appendix 8.

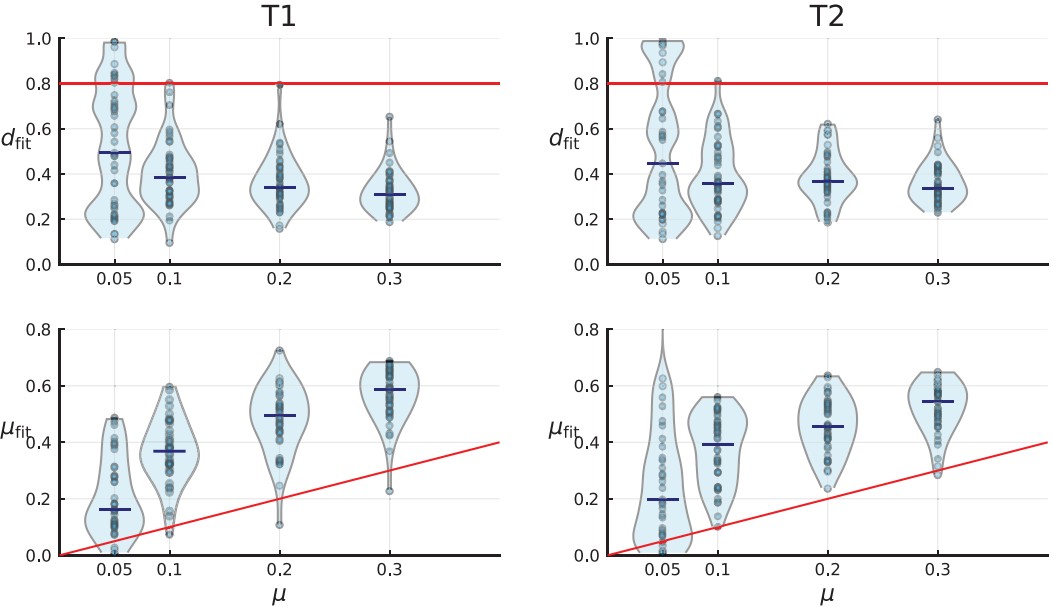

**Appendix 5—figure 6.** Turnover-based inference of cell death and mutation rates under 3D sampling of spatial simulations. 3D simulations are artificially sampled in a way that mimics spatial sampling, sequencing and genotyping in the tumours T1 (left) and T2 (right) by Li et al., as in **Appendix 2—figure 6**, explained in Appendix 2.2. Red lines indicate the true simulation rates. Under this sampling scheme, the inference of mutation and death rate fails, see text.

## Appendix 6

### Inference from mutational distances

In this section, we investigate how to jointly infer the mutation rate per generation and the rate of cell death (relative to that of cell birth) from samples from a growing population using the mutational distances. Our approach is based on combining exact results of *Stadler, 2009* on the distribution of the times since the last common ancestor in a population growing under birth-death dynamics and by *Cheek and Johnston, 2023* on the statistics of the number of cell divisions in a sampled lineage. We compare the resulting statistics of the pairwise mutational distances with the distribution of mutational distances used recently Werner and collaborators (*Werner et al., 2020*) to jointly infer the mutation rate and the relative rate of cell death. Our treatment is a modification of the approach of Werner et al., and below we discuss the need for this modification.

The inference of the mutation rate per generation and the rate at which individuals die (relative to the rate of birth) is a long-standing problem. In population genetics, the times of speciation along a phylogenetic tree and the number of mutations along the tree have been used to perform this inference, using approaches based either on coalescence theory or on birth-death models (*Slatkin and Hudson, 1991*; *Stadler et al., 2015*; *Volz and Frost, 2014*). In these approaches, the accuracy of the inferred parameters (mutations per generations, rates of a birth/death model) depend on how well the times of speciation can be reconstructed. An overview and unified perspective can be found in *MacPherson et al., 2021*. In the present case, due to the low number of mutations along phylogenetic branches the speciation times cannot be determined accurately.

### 6.1 The distribution of mutational distances

Picking a pair of samples randomly and uniformly, the members of that pair differ by $m_1$ and $m_2$ mutations, respectively, from their last common ancestor. The mutations of the last common ancestor (relative to the normal tissue) are taken to be the intersection of the mutations in the two samples (this assumes no back-mutations). The distribution of the numbers of mutations $m_1$ and $m_2$ (mutational distances) across pairs of samples is the metric we will use for inference. A related approach has been taken by *Ling et al., 2015*, who use both coalescent theory and birth-death models to calculate the fraction of pairs of samples that are genetically identical. The full spectrum of mutational distances has recently been used by *Werner et al., 2020* on multi-region tumour sequencing data.

The starting point is a population of cells (individuals) grown from a single cell. These cells independently divide at a rate $b$ and die at a rate $d$ (a birth-death process). Each cell carries a genome of length $L$ and during division, each genomic site of the two daughter cells is mutated with probability $\mu/L$. (In *Werner et al., 2020*, $\mu$ denotes the mutation probability per site.) The population grows to a final size $N$, and the pairwise mutational distances between extant cells are measured. The aim is to infer the mutation rate and the rate of cell death relative to the rate of birth from the distribution of mutational distances.

Starting point is the probability distribution of the times of speciation under a birth death model. A result by theorem 4.1 in *Stadler, 2009* gives the distribution of the time since the last common ancestor in pairs of extant cells picked uniformly from a population that has reached size $N$ as

$$
\rho_{b,d,N}(t) \quad = \frac{2be^{(b-d)t}}{N-1}\left(\frac{1}{\alpha+1}\right)^{N+1} \\
\left[(\alpha+1)^{N+1} - \binom{N+1}{2}\alpha^2 - (N+1)\alpha - 1\right] ,
\tag{14}
$$

where $\alpha = \frac{b-d}{b}\frac{e^{-(b-d)t}}{1-e^{-(b-d)t}}$. We rescale time to $\tau = bt$ obtaining the probability distribution for the rescaled time

$$
\rho_{q,N}(\tau) \quad = \frac{2e^{(1-q)\tau}}{N-1}\left(\frac{1}{\alpha+1}\right)^{N+1} \\
\left[(\alpha+1)^{N+1} - \binom{N+1}{2}\alpha^2 - (N+1)\alpha - 1\right] ,
\tag{15}
$$

with $q = d/b$ the death rate relative to the birth rate and $\alpha = (1-q)\frac{e^{-(1-q)\tau}}{1-e^{-(1-q)\tau}}$.

We now consider a pair of extant cells with time $\tau$ since their last common ancestor. The two cells and their ancestor define two lineages, with $i_1$ and $i_2$ cell divisions after speciation, respectively, since the last common ancestor. *Cheek and Johnston, 2023* have calculated the distribution of the number of cell divisions of a lineage of age $\tau$ conditioned on survival. The distribution is affected by lineages that have undergone many cell divisions being statistically overrepresented in the set of extant cells. This is an example of the inspection paradox well known in probability (*Stein and Dattero, 1985*), where for instance determining the average size of a family by polling individuals for the number of their siblings results in a biased estimate. The resulting statistics for the number of divisions $i$ found by *Cheek and Johnston, 2023* is a mixture of Poisson distributions and in the present notation is

$$p_\tau(i) = 2^i e^{-d\tau} \frac{b - de^{-(b-d)\tau}}{b(e^{(b-d)\tau} - 1)} \sum_{j=0}^{i} \frac{1}{j!} \left( \frac{be^{(b-d)\tau} - d}{(b-d)e^{b\tau}} \left( \log\left( \frac{(b-d)e^{b\tau}}{be^{(b-d)\tau} - d} \right) \right)^j - e^{-b\tau}(b\tau)^j \right) \quad (16)$$

This expression follows *Equation 20* in the archived version (https://arxiv.org/pdf/2205.13875.pdf) of *Cheek and Johnston, 2023* the corresponding *Equation 21* differs by an erroneous factor of $(-1)^j$. The partial sum of the exponential series can be written in terms of the incomplete Gamma-function.

Finally, we look at the distribution of the number of mutations in a lineage given the number of cell divisions $i$ after speciation. Since mutations occur independently at different sites and at different divisions along a lineage, and the per-site mutation probability is small, the number $m$ of mutations after $i+1$ divisions (now including the speciation event itself) is Poisson distributed with mean $(i+1)\mu$, giving

$$P_{(i+1)\mu}(m) = \exp(-(i+1)\mu)((i+1)\mu)^m/m! . \quad (17)$$

We combine these results to estimate the distribution of the mutational distances $m_1$ and $m_2$ from the last common ancestor of a pair of cells randomly selected from extant cells in a population of size $N$, obtaining

$$P(m_1, m_2) = \int_0^\infty d\tau \rho_{q,N}(\tau) \sum_{i_1=0}^{\infty} p_\tau(i_1) P_{(i_1+1)\mu}(m_1) \sum_{i_2=0}^{\infty} p_\tau(i_2) P_{(i_2+1)\mu}(m_2) . \quad (18)$$

Similarly, the distribution of individual mutational distances (between the ancestor of a pair of cells and an element of that pair) is given by

$$P(m) = \int_0^\infty d\tau \rho_{q,N}(\tau) \sum_{i=0}^{\infty} p_\tau(i) P_{(i+1)\mu}(m) \quad (19)$$

These expressions neglect that choosing pairs of samples does not generate the typical number of cell divisions given the age of a lineage, but the error turns out to be small. *Appendix 6—figure 1A* shows the distribution of mutational distances for a single population ($\mu = 1, q = 3/4, N = 300$) compared to the result *Equation 19* (green line).

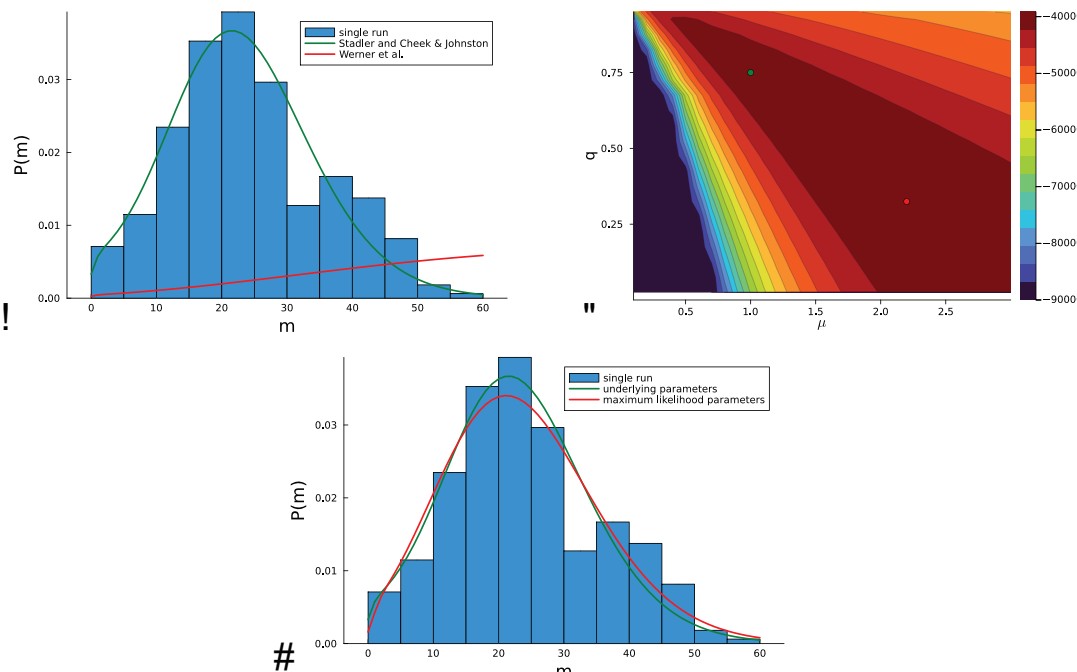

**Appendix 6—figure 1.** Inference on artificial data using the likelihood *Equation 18* based on mutational distances. For a concrete example, in a single run, a population is grown at rate of birth $b = 1$ and rate of death $d = 3/4$ with a genomic mutation rate $\mu = 1$ from a single cell until the population size has reached $N = 300$ cells. (**A**) The histogram of the mutational distances $m$ from the single population is shown in blue. The green solid line shows the corresponding distribution *Equation 19*, the red solid line indicates the distribution of mutational distances derived in equation (7) in *Werner et al., 2020*, see text. (**B**) The log-likelihood landscape defined by *Equation 18* with the mutation rate $\mu$ on the $x$-axis and the relative death rate $q = d/b$ on the $y$-axis. The underlying parameters $\mu = 1$ and $q = 0.75$ are indicated with a green dot, the maximum of the likelihood landscape is indicated by the red dot. The likelihood landscape shows a pronounced ridge of high likelihoods. The ridge-like shape of the likelihood landscape means that there are many different parameter values compatible with the data that yield nearly the same distribution of mutational distances *Equation 18*. (**C**) The last point is illustrated by the distribution of the mutational distances: As in (**A**), the histogram of mutational distances $m$ is shown in blue, and the green solid line again shows the distribution *Equation 19* at the underlying parameters $\mu = 1$ and $q = 0.75$. The red line shows the the distribution *Equation 19* at the maximum-likelihood parameters (near $\mu = 2.2$ and $q = 0.33$).

## 6.2 Likelihood-based inference from pairwise mutational distances

*Equation 18* serves as the likelihood function for the parameter inference. Enumerating all pairs of extant cells gives a list of mutational distances between each element of the pair and the last common ancestor, $\{m_1^\nu, m_2^\nu\}$, where the index $\nu$ runs over all pairs of extant cells. Maximizing the log-likelihood defined by *Equation 18*

$$\sum_\nu \log\left(P(m_1^\nu, m_2^\nu)\right) \tag{20}$$

with respect to the free parameters $q$ and $\mu$ for a given population size $N$ yields the maximum-likelihood estimates of the relative death rate $q = d/b$ and the genomic mutation rate $\mu$ per cell division and cell. The computation of double sums in the likelihood *Equation 18* can be avoided by precomputing the sum over $i_1$ for different values of $\tau$ and $m_1$ and using the same lookup-table for the sum over $i_2$.

The likelihood *Equation 18* differs in three points from the likelihood used in *Werner et al., 2020*: (i) We use a birth-death model, whereas Werner et al. use an approximation based on coalescence theory (*Slatkin and Hudson, 1991*). This point should be inconsequential, the correct use of either model should yield a correct inference (see below). (ii) We use the exact result by *Cheek and Johnston, 2023* for the distribution of the number of divisions *Equation 16* which accounts

for the skewed statistics due to the inspection paradox. Werner et al. use a heuristic which does not account for this effect. (iii) The distribution *Equation 18* describes correlations between the mutational distance $m_1$ and $m_2$, which enter the log-likelihood *Equation 20*. The method in *Werner et al., 2020* does not exploit these correlations.

Regarding point (i), neither the distribution of pairwise speciation times nor the coalescence probabilities per time can be evaluated without knowing the final population size $N$, see for instance (*Stadler, 2009* and *Slatkin and Hudson, 1991*; *Equation 5*). Yet, the expression derived in *Werner et al., 2020* (equation (6) in *Werner et al., 2020*) for the coalescence probability does not depend on the final population size. The resulting distribution of mutational distances (equation (7) in *Werner et al., 2020*) is shown in *Appendix 6—figure 1A* (red line) and disagrees strongly both with our result *Appendix 6—figure 1* (green line) and numerical simulations.

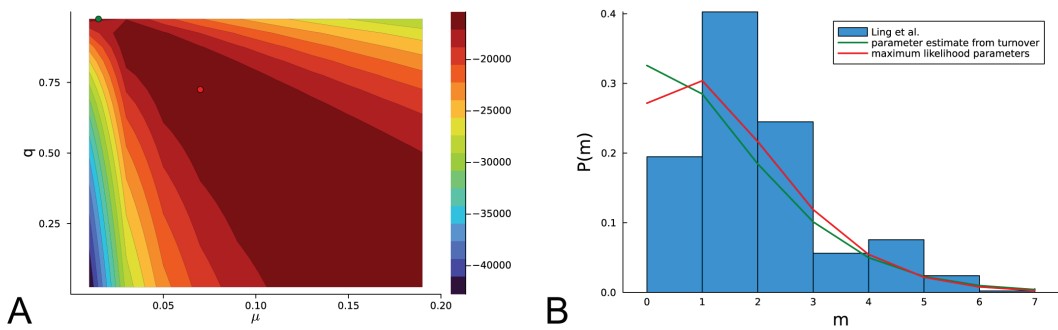

**Appendix 6—figure 2.** Inference using the likelihood *Equation 18* on the the empirical data of Ling et al. (**A**) The log-likelihood landscape defined by *Equation 18* with the mutation rate $\mu$ on the $x$-axis and the relative death rate on the $y$-axis, on the basis of mutational distances between samples in the data of *Ling et al., 2015*. The maximum of the likelihood landscape is indicated by the red dot ($\mu = 0.07$ and $q = 0.73$), the parameters previously inferred using the turnover of Appendix 5 ($q = 0.975, \mu = 0.015$) are indicated with a green dot (top edge left). As in the artificial data of in *Appendix 6—figure 1*, the likelihood landscape shows a pronounced ridge of high likelihood. (**B**) The histogram of the mutational distances $m$ from samples of Ling et al. is shown in blue. The green line shows the distribution *Equation 19* at the parameters inferred using the turnover of Appendix 5. The red line shows the the distribution *Equation 19* at the maximum-likelihood parameters.

Still, the likelihood of *Werner et al., 2020* can infer the model parameters correctly in the regime of large mutation rates, where there are clear oscillations in the distribution of mutational distances of period $\mu$: The number of cell divisions between two cells is always integer, so the average number of mutations is an integer multiple of $\mu$. In the regime $\mu \gg 1$, one can resolve individual cell divisions in the spectrum of mutational distances, which cause distinguishable peaks at $\mu, 2\mu, 3\mu, \ldots$ (see artifical data in Fig. 2A in *Werner et al., 2020*). In this regime, the likelihood is dominated by parameters yielding the correct oscillation period. In the presence of such oscillations, the model parameters can easily be determined without using a likelihood by reading off the mutation rate from the oscillation period, and the relative death rate from a plot of the cumulative frequency of mutations versus the inverse frequency as in *Williams et al., 2016*. However, outside this regime, we found the method (*Werner et al., 2020*) returns incorrect parameter estimates. For whole-exome data, such as the Ling et al. data set and many of the data sets analyzed in *Werner et al., 2020*, $\mu$ need not be much larger than one (in particular in WES data from tumours lacking the hypermutator phenotype), making a correct treatment of either coalescence probability or the distribution of speciation times necessary.

*Appendix 6—figure 1B* shows the log-likelihood *Equation 20* landscape for the same population of 300 cells used in *Appendix 6—figure 1A*. A pronounced ridge-like shape is seen with a broad plateau of high likelihoods. This ridge includes the underlying parameters ($\mu = 1, q = 3/4$), although the maximum occurs elsewhere near $\mu = 2.2$ and $q = 0.3$. In general, broad maxima of the likelihood make the precise position of the likelihood maximum strongly depend on fluctuations in the data. *Appendix 6—figure 1C* shows how the distribution of mutational distances *Equation 19* at the maximum likelihood parameters (red line) and at the underlying parameters (green line). Both of these distributions fit the distribution of mutational distances quite well, although the fitting

parameters are rather different. This is a property of the summary statistics used here, in line with the ridge of high likelihood values seen in *Appendix 6—figure 1B*.

The same picture is also seen in the empirical data of *Ling et al., 2015*. *Appendix 6—figure 2A* shows a pronounced ridge of high likelihood. *Appendix 6—figure 2B* shows the corresponding distribution of mutational distances (blue histogram) as well as the distribution *Equation 19* both for the parameters inferred using the genetic turnover (green line, see Appendix 5) and for the parameters maximising the likelihood *Equation 20* based on the mutational distances (red line). Again both parameter set lead to a reasonable fit with the distribution of mutational distances, indicating that mutational distances are not a suitable summary statistics for inference (unless the expected number of mutations per cell division is substantially larger than one, see above).

## Appendix 7

### Estimating the population size $N$

The probability of coalescence or the time since the last common ancestor of two cells both depend on the population size. As a result, the distribution of the mutational distances *Equation 19* and hence the likelihood function *Equation 18* depend on the population size. Also, to infer the relative growth rate $q = d/b$ from the turnover parameter as in Appendix 5 we need to know the population size. Similarly, when telling surface growth from volume growth in Appendix 2, or tracking the spread of mutations in Appendix 4, we use mutations which entered the population at a specific time when the tumour had a certain size. Therefore our measures describe the growth dynamics at that particular time and tumour size, depending on the input data and mutation filters. In this section we discuss out the effective population sizes $N$ for different sets of mutations.

Due to the finite sequencing resolution this effective population size is not simply the final number of cells in the tumour population. Rare variants below the frequency resolution $f_{res}$ are underrepresented or missing due to a finite sequencing coverage and stochastic errors of sequencing. Only mutations with sufficiently high frequency can be identified by sequencing, and those arose early in the evolution of the tumour when the tumour had a small size (*Williams et al., 2016*). The statistics of mutations found in a large tumour with a finite sequencing depth corresponds to a smaller tumour where all mutations and frequencies are known. Here, we address the question of what the effective population size of that smaller tumour is, given a particular frequency resolution $f_{res}$ and test our answer in numerical simulations.

Under deterministic exponential growth - that is if one neglects fluctuations in clone sizes - the analysis is simple: each generation brings forth new mutations that have an initial frequency $f = 1/N$, where $N$ is the total population size at birth of the first mutant (*Williams et al., 2016*). Under neutral evolution, new clones match the total population growth rate $\lambda = b - d$ (cell birth rate $b$ and death rate $d$) and thus maintain their frequency over time. Consequently, keeping only mutations down to a cutoff frequency $f_{res}$ effectively casts the system into the early stages of tumour growth. Specifically, one observes the clonal makeup of the population at the time when mutations with frequency $f_{res}$ have just appeared. Correspondingly, one would set the effective population size to

$$N = 1/f_{res} ,  \tag{21}$$

see (*Williams et al., 2016*).

We now discuss a correction to this deterministic result due to stochastic fluctuations in the clone sizes. In an extreme case, such fluctuations can drive a clone to extinction. For $t \to \infty$ the scaled population size $e^{-\lambda t}N(t)$ (of a clone or an entire population) has been shown by *Durrett, 2015* (chapter 3 theorem 1) to have a limiting distribution given by

$$\frac{d}{b}\delta_{N,0} + \frac{\lambda}{b}\left(1 - \frac{d}{b}\right)e^{-(1-d/b)e^{-\lambda t}N} . \tag{22}$$

Here, $\frac{d}{b}$ is the extinction probability, such that when conditioning on survival

$$(e^{-\lambda t}N(t)|N(t) > 0 \,\forall\, t) \overset{t\to\infty}{\longrightarrow} (1 - \frac{d}{b})e^{-(1-d/b)e^{-\lambda t}N} \tag{23}$$

the expected population size becomes $\langle N(t) \rangle = \frac{1}{1-d/b}e^{\lambda t}$ for sufficiently large times $t$. The exponential term in this expression is the result due to determininistic growth, the prefactor is a correction due to fluctuations which can drive a clone to extinction. The contribution is large if the death rate $d$ is close to the rate of birth $b$, so extinctions events are frequent.

Whereas $e^{-\lambda t}N(t)$ is an exponentially distributed random variable for an ensemble of populations, but fixed for a given population, the mutant clones have an analogous relation between expectation values of the population size at birth of the mutant, $N$, and the mutation frequency $f_{res}$

$$N = \frac{1}{1 - d/b}\frac{1}{f_{res}} . \tag{24}$$

This gives a correction to the deterministic estimate $N = \frac{1}{f_{res}}$ due to stochastic extinction events.

We test this relationship numerically in two ways. *Appendix 7—figure 1* shows the population size at birth of a mutation against (one over) its final frequency and compares the numerical results to the deterministic estimate *Equation 21* and the stochastic correction *Equation 24*.

*Appendix 7—figure 2* probes how well the effective population size is described by *Equation 24* when low-frequency mutations up to a frequency threshold $f_{res}$ are disregarded. We grow a population to size $N = 10000$ and disregard all mutations with final frequencies less than $f_{res} = 0.01$. We then compare the resulting mutational distances (mean and standard deviation over each population) with those of populations grown to different final sizes *without* removing low-frequency mutants. Under the deterministic model (*Equation 21*), we expect the mutational distances to match those in a population of size $N = 100$. Instead, we find that the mutational distances match those in a population of a size given by the stochastic correction (*Equation 24*), which increases with the relative rate of cell death $q = d/b$.

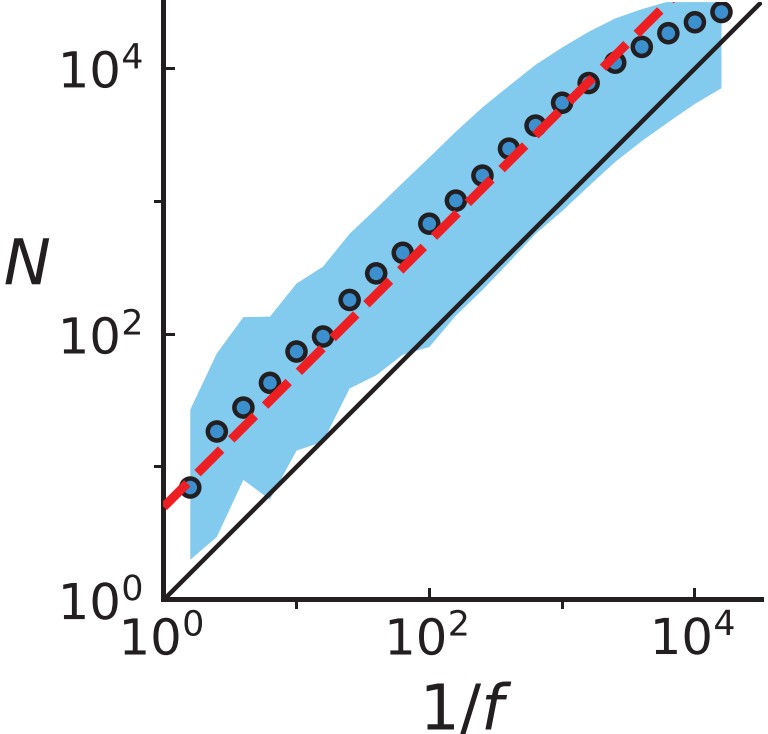

**Appendix 7—figure 1.** Population size $N$ at birth of a mutation with final frequency $f$ Circular markers show the mean population size $N$ for a given frequency bin. The blue ribbon covers the 99th percentile. The red line shows the relation *Equation 24* with the correction for fluctuating clone sizes, the black line shows the deterministic result *Equation 21*. Mutations were collected from 20 3D spatial simulations up to a population size 40,000 with birth rate $b = 1$., death rate $d = 0.8$, and mutation rate $\mu = 0.3$.

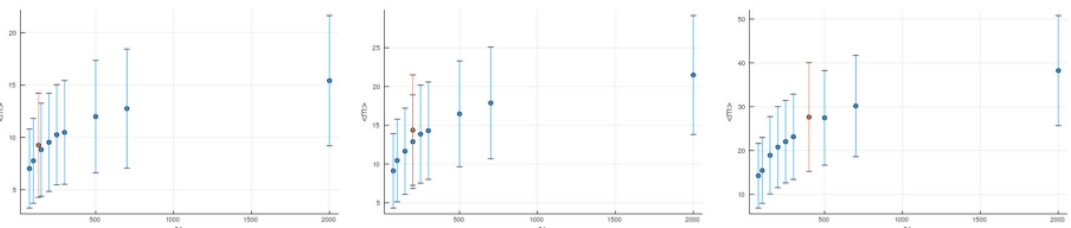

**Appendix 7—figure 2.** The effect of disregarding low-frequency mutations. The mean mutational distances and their standard deviations (blue markers and blue error bars, respectively) are shown for populations grown to different sizes $N$ (x axis) at different relative death rates $q = 0.25, 0.5, 0.75$ (left to right). In each of these panels, the orange markers show the mean and standard deviation of the mutational distance for a population *Appendix 7—figure 2 continued on next page*

*Appendix 7—figure 2 continued*

of size $N = 10000$ whose low-frequency mutants with frequency $f < 0.01$ have been discarded. For the effective population size we use *Equation 24*. Rather than staying constant as they would under *Equation 21*, the corresponding mean mutational distances increase with the relative death rate $q$, and this increase is compatible with *Equation 24*, as seen from the orange markers following the blue ones. All quantities are averaged over 100 runs. Note that error bars give the standard deviations of the mutational distances, not standard errors.

## 7.1 Frequency thresholds in the empirical data

It remains to set the minimal frequency $f_{res}$ in the data of *Ling et al., 2015* and *Li et al., 2022*. $f_{res}$ depends not only on the sequencing depth per sample but also the spatial sampling density. We discuss the dependency of the frequency threshold $f_{res}$ on the spatial sampling for the case of sampling within one cross section of a 3D spherical tumour, as in the Ling et al. data, or several slices of a hemisphere, as in the two tumours from Li et al. In both cases, the idea is that only a fraction of the whole tumour is being sampled. Therefore, the rarest mutations found in the sample actually occur at a lower frequency within the whole 3D tumour and happened to fall within the sampled fraction of the tumour. We set a threshold on the frequency of mutations $f_m$ within the set of sequenced samples that is larger or equal to the frequency resolution for the given sequencing depth.

The minimal resolved whole tumour frequency $f_{res}$ is the smallest frequency with respect to the whole tumour of any mutation that occurs at a frequency greater than $f_m$ in the sampled plane. It necessarily describes a mutation that only occurs in our set of samples and has frequency $f_m$ therein. From our choice of $f_{res}$ on, we proceed to estimate the effective population size $N$ for each of the three tumours.

### Sampling from a tumour cross-section (Ling et al.)

*Ling et al., 2015* take samples from a cross section of the 3D tumour and detect mutations with a range of mutation frequencies within this set of samples. To find which is the minimal mutation frequency with respect to the whole tumour that still passes a threshold $f_m$ on the sampled slice, we ask how many samples under the given regular spacing would fit into a spherical tumour of the same diameter.

Given the spatially even and dense sampling we assume a triangular lattice and take the mean nearest neighbour distance between samples as the lattice constant $r$. We compute the mean spherical tumour radius $R$ using the dense packing ratio $\frac{\pi}{2\sqrt{3}}$ for the plane.

$$R = \sqrt{nr^2 / \frac{\pi}{2\sqrt{3}}}$$

(25)

Then $\tilde{n}$ is the number of samples that fill the spherical tumour of radius $R$ given the dense packing ratio $\frac{\pi}{3\sqrt{2}}$ for the 3D lattice:

$$\tilde{n} = (R/r)^3 \frac{\pi}{3\sqrt{2}}$$

(26)

$$n^{3/2} \frac{2}{\sqrt{\pi\sqrt{3}}}$$

(27)

Finally, given a frequency threshold $f_m$ within the sampled plane for $n$ samples, we can estimate the minimal, resolved frequency in the whole tumour to be $f_{res} = f_m n/\tilde{n}$ because the total read-depth at a given site on the genome scales linearly with the number of samples from $n$ to $\tilde{n}$.

Essentially, if our lower bound on mutation frequency in the sampled plane is $f_m$ we include mutations down to the frequency $f_{res}$ with respect to the 3D tumour regardless of fluctuations in read counts. Increasing the sample size $n$ necessarily improves the frequency resolution as $f_m^{-1} \propto n$ under uniform sampling but the minimal, resolved frequency when sampling only the cross-section of a spherical tumour scales as $f_{res} \propto n^{3/2}$. Thus, the effective population size *Equation 24* becomes

$$N_{eff} = \frac{1}{\frac{n}{\tilde{n}} f_m} \frac{1}{1 - d/b}$$

(28)

$$= n^{3/2} \frac{2}{\sqrt{\pi \sqrt{3}}} \frac{1}{nf_m} \frac{1}{1 - d/b}. \tag{29}$$

In the rest of this section as well as in other sections we will simply refer to the effective population size $N_{\text{eff}}$ as $N$.

Based on this estimate and the correction for high cell turnover from *Equation 24*, we use the effective population size *Equation 28* both for the inference of the relative death rate from the turnover, and for the analysis based on the mutational distances.

Because the frequency resolution and hence the population size estimate depends on the density of sampling we actually have two different effective population sizes: In the set of samples characterized by genotyping (high resolution, used to distinguish between volume and surface growth and for turnover inference) we consider mutations that occur in at least two of these samples. Such mutations can be assigned to clones, whereas it is impossible to confidently tell the relation of a mutation that occurs in a single sample to other mutations in its sample, see Appendix 8. Choosing the frequency cutoff such that mutations occur in at least two genotyped samples means that both metrics (directed growth and turnover) describe the dynamics of the set of clones up to the tumour size where a clone fails to colonize a spatially separated sample and reach detectable size therein until surgical resection. This is where the sampling density comes into play, since increasing the number of samples makes it more likely to sample a region where the clone has already reached sufficient size. We obtain the corresponding effective population size $N$ using *Equation 28*: $n = 285$ samples underwent genotyping, giving an estimate of $\tilde{n} = 4000$ in 3D, the resolution is set to $n \cdot f_m = 2$ as specified in 8 because mutations must be recovered in two samples, and finally our turnover-based inference results in a turnover rate $d/b = 0.975$, giving $N = \tilde{n} \cdot \frac{1}{nf_m} \frac{1}{1-d/b} \approx 82500$.

In contrast, we can only measure the spatial dispersion of mutations that grew to detectable frequency in at least two whole-exome sequenced samples (deeply sequence samples and used to obtain cancer cell fractions and the site frequency spectrum). These samples resolve mutations up to a minimum frequency of $f_{\text{res}} = 1/40$ (see Appendix 4), meaning that those mutations arose when the size of the tumour was only around $N \approx 3300$ in size, where $N$ is again the effective population size *Equation 28*. Further details beyond this early stage of tumour development remain unresolved. Future data with a deeper sequencing and a higher spatial resolution will allow probing tumour evolution beyond this early stage.

The larger set of 285 genotyped samples targets the sites of mutations detected by whole-exome sequencing in the smaller set consisting of 23 samples. One might therefore ask whether the estimate on effective population size $N$ depends on the smaller, WES-sequenced set of samples which defines the selection of mutations or the larger genotyped set from which clones are inferred, see also Appendix 5. We explored this question in simulations over a range of turnover rates $d/b$ from 0 to 0.8 by applying a sampling procedure similar to that of Ling et al. (see *Appendix 5—figure 3*), varying the size of the smaller (sequencing) and the larger (genotyping) set of samples, and observing how the measured clade and clone turnover $W_a$ and $W_o$ compares to the theoretical prediction which depends on $N$. How closely the measured turnover fits the theory at a given $N$ reflects in the quality of inference of $d/b$ and $\mu$. We find that, (1) the measured turnover fits the theory closest if $N$ is computed from the (larger) sample size of the genotyping set, (2) the measured turnover does not vary up to sampling noise when taking differently sized mutation-calling sets, and (3) the measured turnover decreases and increases for larger and smaller sample size of the genotyping set, respectively, as is theoretically expected for larger and smaller $N$, respectively.

## Sampling from a tumour hemisphere (Li et al.)

Similarly, we estimate the effective population size for the tumours T1 and T2 from *Li et al., 2022*. In contrast to *Ling et al., 2015*, samples are taken from several slices of the upper hemispheres of both tumours. Therefore, the estimate of the effective number of samples in the whole sphere $\tilde{n}$ is simply $2n$ for $n$ samples in the hemisphere.

Including the $d$-dependent correction in the estimate of effective population size $N$ for the conditioning on survival of tumours *Equation 24*, given the estimated turnover rates $d/b = 0.944$ and 0.976, yields $N_{\text{T1}} = 3000$ and $N_{\text{T2}} = 6600$. Despite the high uncertainty in $d/b$, the estimated

population size is lower than what we found for the clonal data of Ling et al. by more than an order of magnitude due to the lower density of samples in the data by Li et al.

# Appendix 8

## Inference of clones

Our methods are based on the spatial distribution of genetic mutations in a mixed population of cells with different genotypes. It is straightforward to evaluate and interpret them for single-cell data where each sample represent a true genotype from the population. However, samples obtained by needle biopsies contain thousands of cells (about 200 in Ling et al. and Li et al. or ~ 3200 in the Li et al. WGS samples) and generally contain a mixture of clones. This mixing of clones is particularly prevalent under volume growth, as can be seen by the dispersion of mutations, see Appendix 4. Genomic data from a large number of samples is well suited for the reconstruction of clones because the different samples already reveal to some extent the variety in genotypes. The clonal composition can be further resolved by clustering mutations by their sample frequencies on the level of single samples as well as the whole tumour (if mutation frequencies are available). For instance, two mutations might have very similar whole-tumour frequencies but do not coincide across samples (i.e. some samples have mutation A but not B) or have different frequencies in single samples and therefore do not belong to the same clone.

Clustering by frequency works best given a large number of mutations. Mutations belonging to one clone occur at equal frequency up to sampling noise and hence form a frequency cluster. The more mutations a clone has, the more confidently a clone can be identified, opposed to the random similarity in frequency between unrelated mutations. A larger number of samples on the other hand imposes stricter constraints on the clustering of mutations. Mutations must have very similar frequency in all samples to be part of the same clone, which might be hard to fulfill in the presence of sequencing noise.

We use two methods for the reconstruction of clones: As a first approach we use a simple clustering scheme where mutations that coincide across samples are clustered into clones. A simple example to illustrate this scheme mixes genotypes labeled by letters A, B, C, D into samples. The samples {AC},{AB},{BD} decompose into {[AC]}, {[A][B]}, {[BD]}, and as a rule, an ancestral clone will not be added to the sample, so {[AC]} does not become {[A],[AC]}. Despite its simplicity this scheme, when applied to artificially sampled simulations, is sufficient to recover the distributions of the direction angle (*Appendix 2—figures 2 and 3*) and the rates from the turnover based inference already measured using simulated single cell data of the same simulations (see *Appendix 5—figure 3*).

We further compare this simple clone reconstruction scheme to the LICHeE tool (*Popic et al., 2015*) designed specifically for multi-sample clone inference. LICHeE can leverage mutation frequencies of multiple samples to simultaneously cluster clones and determine their lineages and has been shown by the authors to better resolve tumour phylogenies when compared to tools designed for single bulk-samples (*Popic et al., 2015*).

In the case of the data by Ling et al., LICHeE detects 27 clones and their phylogenetic tree from the presence profiles in the large set of 285 genotyped samples. Out of these clones, 22 agree with clones from our simple clustering, with the remaining 5 only differing each by one mutation which are ambiguous in their assignment to a clone and were removed from the set of mutations. However, both clustered sets of clones produce very similar results in the discussion of directed growth, see Appendix 2, and in the turnover based inference of mutation and cell death rates, see Appendix 5.

Li et al., on the other hand, probe for the presence of 906 and 565 SNVs using genotyping in the tumours T1 and T2, respectively. Given the large number of mutations per sample, it is more likely than in the data by Ling et al. that the genotyping misses some mutations in a given sample or falsely reports them as present. Our simple clone reconstruction is sensitive to both false negatives and positives and very likely produces false clones here. The LICHeE tool, on the other hand, cannot resolve the constraint network defined by the large sets of genotyped sample profiles.

Instead, we apply LICHeE to the set of WGS samples, where it first clusters mutations by their measured variant allele frequencies before solving the (smaller) constraint network imposed by the WGS samples. LICHeE infers 25 and 14 clones for T1 and T2, respectively. We then proceed to recover the inferred clones in the genotyped samples by splitting the sample phenotypes into their constituent clones. This is useful for the analysis of the growth mode in Appendix 2.2, which relies on the higher spatial resolution of the genotyped samples. Testing this approach of splitting a sample into the different clones on the WGS samples from which LICHeE inferred the clones in the first place exactly recovers the split proposed by LICHeE.

**Appendix 8—table 1.** Clones and their phylogeny from the Ling et al data.

Each row shows one branch in the phylogenic tree inferred by LICHeE for the Ling et al. data. Columns labelled 'parent' and 'offspring' show mutations of the parental and offspring clone, respectively. The mutations along the branch are the private mutations the offspring has gained, the difference between the two sets of mutations. 'root' indicates that the clone has no ancestor in the given set of clones other than the common ancestor to all clones which has the clonal mutations.

| Parent | Offspring |
| --- | --- |
| [30] | [30, 33, 34] |
| root | [30] |
| root | [22] |
| root | [6] |
| [23, 24] | [23, 24, 25] |
| [10, 11] | [10, 11, 12, 13, 14] |
| [23, 24] | [23, 24, 26] |
| [6, 7] | [6, 7, 8] |
| root | [3] |
| root | [15, 16] |
| [15, 16] | [15, 16, 17] |
| [6] | [6, 7] |
| [6, 7] | [6, 7, 9] |
| [15, 16, 17] | [15, 16, 17, 18] |
| [3] | [3, 4] |
| [30] | [30, 31] |
| [23, 24, 27] | [23, 24, 27, 28] |
| root | [10] |
| [30] | [30, 35] |
| [23, 24] | [23, 24, 29] |
| [23, 24] | [23, 24, 27] |
| root | [23, 24] |
| [3] | [3, 5] |
| [30, 31] | [30, 31, 32] |
| root | [1, 2] |
| [10] | [10, 11] |
| [15, 16] | [15, 16, 19, 20, 21] |

**Appendix 8—table 2.** LICHeE-derived branches for tumours T1 and T2 of the Li et al data.
Each row shows one branch in the phylogenic tree inferred by LICHeE for the T1 tumour of the Li
et al. data. Clones have too many private mutations to list and are therefore only numbered Clone
numbers are given in columns 'parent' and 'offspring' along with the number of mutations along a
branch ('mutations'). 'root' indicates that the clone has no ancestor in the given set of clones other
than the common ancestor to all clones which has the clonal mutations.

| T1 | | | T2 | | |
|---|---|---|---|---|---|
| parent | offspring | mutations | parent | offspring | mutations |
| 15 | 1 | 32 | 4 | 1 | 2 |
| 15 | 2 | 25 | 3 | 2 | 14 |
| 13 | 3 | 8 | 8 | 3 | 6 |
| 8 | 4 | 7 | root | 4 | 20 |
| root | 5 | 68 | 11 | 5 | 4 |
| 20 | 6 | 6 | 7 | 6 | 16 |
| 19 | 7 | 30 | 1 | 7 | 27 |
| 9 | 8 | 10 | 11 | 8 | 17 |
| 5 | 9 | 56 | 8 | 9 | 7 |
| 13 | 10 | 3 | root | 10 | 38 |
| 20 | 11 | 6 | 4 | 11 | 56 |
| 16 | 12 | 19 | 7 | 12 | 10 |
| 21 | 13 | 10 | 3 | 13 | 2 |
| 9 | 14 | 14 | root | 14 | 39 |
| 7 | 15 | 57 | | | |
| 5 | 16 | 35 | | | |
| 12 | 17 | 12 | | | |
| 21 | 18 | 11 | | | |
| root | 19 | 46 | | | |
| 8 | 20 | 14 | | | |
| 22 | 21 | 9 | | | |
| 12 | 22 | 18 | | | |
| root | 23 | 2 | | | |
| root | 24 | 55 | | | |
| 22 | 25 | 10 | | | |

# Appendix 9

## Subsampling of mutations and mutational signatures

In order to estimate the statistical robustness of our estimate of the per-generation mutation rate and the relative rate of cell death, we perform the inference described in Appendix 5 also on subsets of the mutations. (A standard bootstrapping, where mutations are resampled without replacement is not feasible here, as it would produce duplicates of samples with zero mutational distance.) *Appendix 9—figure 1* shows the inference results for the Ling et al. data for different relative sizes of the subset $L$ ranging from 0.5 to 0.9. No systematic drift with the sampling fraction $L$ is seen; correspondingly *Figure 4C* in the main text shows the results from the different fractions combined.

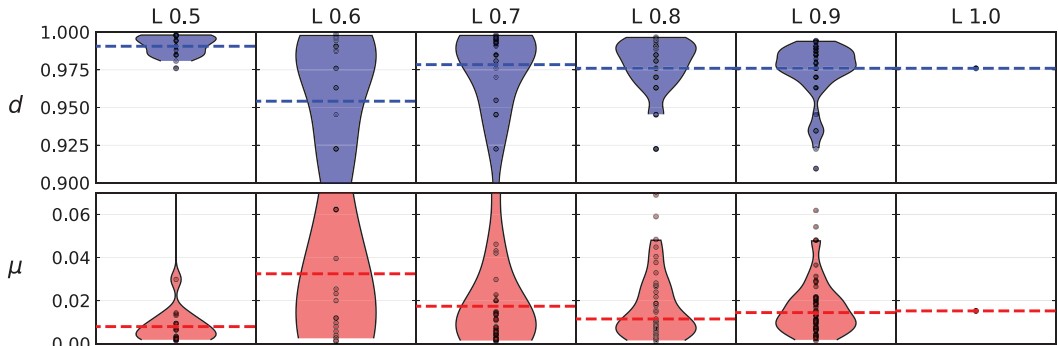

**Appendix 9—figure 1.** Violin plots of model parameters (number of mutations per generation and relative death rate) inferred from different fractions $L$ of the mutations present in the Ling et al data.

An unrelated point linked to subsampling of mutations concerns our observation of a particular mutational signature (SBS22) in the clonal mutations of Ling et al., but not the subclonal mutations. This is compatible with exposure to the mutagen causing this particular signature during tumourigenesis, and the absence of the mutagen during the later stages of tumour growth. Here, we ask how likely is an alternative scenario, namely that the signature is present also in subclonal mutations with the same statistical weight as seen in the clonal mutations, but is not detected due to the finite number of mutations (sampling noise).

We decomposed the mutational profiles of the tumours from the two studies using our in-house method SigNet (*Serrano Colome et al., 2023*). SigNet is based on an artificial neural network and predicts the weights of the COSMIC v3 mutational signature catalog (*Alexandrov et al., 2020*) in individual samples. If the sample profile does not correspond to the training data, it is decomposed with non-negative least squares (NNLS). SBS1, SBS5 and SBS40 are associated with endogenous mutational processes. Since the samples that contain SBS25 were decomposed with a non-negative least squares approach (as they were classified as out-of-distribution by SigNet), it is likely that SBS25 is an erroneous assignment derived from SBS5 and SBS22 mutations (*Alexandrov et al., 2020*). The high number of signatures classified as 'Others' in samples for which the NNLS approach was used is compatible with the low accuracy of this approach when the number of mutations in the sample is small.

To address the effect of subsampling on the mutational signatures, we run the in-house SigNet algorithm (*Serrano Colome et al., 2023*) on $10^4$ artificially generated instances with different weights of SBS22. Weights of the other signatures contributing to each simulated sample were maintained at the same relative proportions as found in the decomposition of the subclonal mutations in the whole-exome data of Ling et al. for all weights of SBS22. Each instance contained 66 mutations, the number of subclonal exomic mutations in the Ling et al. tumour.

The weight of SBS22 when pooling clonal and subclonal mutations was 0.78. The probability that although subclonal mutations have SBS22 at this particular weight, but the signature is not detected on the basis of only 66 mutations is estimated to be less than $10^{-4}$ (the event did not occur in $10^4$ runs of the simulation). This provides an estimate of how likely the alternative scenario of undetected SBS22 is (p-value), leading us to reject the alternative scenario.

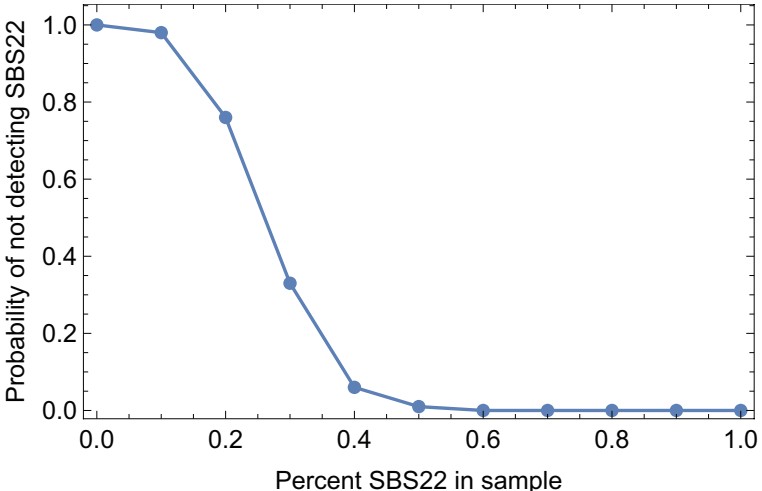

**Appendix 9—figure 2.** Sampling mutations with different weights of a particular signature (SBS22). Weights of SBS22 are shown on the x-axis, the probability of not detecting SBS22 given 66 mutations is shown on the y axis and rapidly decays with the underlying weight of SBS22, see text.

