## [Editor Report · eLife Assessment]

The article uses a cell-based model to investigate how mutations and cells spread throughout a tumour. The paper uses published data and the proposed model to understand how growth and death mechanisms lead to the observed data. This work provides an **important** insight into the early stages of tumour development. From the work provided here, the results are **convincing**, using a thorough analysis.

---

## [Referee Report · Reviewer #1 (Public review)]

Summary:

Arman Angaji and his team delved into the intricate world of tumor growth and evolution, utilizing a blend of computer simulations and real patient data from liver cancer.

Strengths:

Their analysis of how mutations and clones are distributed within tumors revealed an interesting finding: tumors don't just spread from their edges as previously believed. Instead, they expand both from within and the edges simultaneously, suggesting a unique growth mode. This mode naturally indicates that external forces may play a role in cancer cells dispersion within the tumor. Moreover, their research hints at an intriguing phenomenon - the high death rate of progenitor cells and extremely slow pace in growth in the initial phase of tumor expansion. Understanding this dynamic could significantly impact our comprehension of cancer development.

Weaknesses:

It's important to note, however, that this study relies on specific computer models, metrics derived from inferred clones, and a limited number of patient data. While the insights gained are promising, further investigation is essential to validate these findings. Nonetheless, this work opens up exciting avenues for comprehending the evolution of cancers.

Comments on revised submission:

The authors have effectively addressed my concerns. This revision is excellent.

---

## [Referee Report · Reviewer #2 (Public review)]

Summary:

The article uses a cell-based model to investigate how mutations and cells spread throughout a tumour. The paper uses published data and the proposed model to understand how growth and death mechanisms lead to the observed data. This work provides an insight into the early stages of tumour development. From the work provided here, the results are solid, showing a thorough analysis. The article is well written and presents a very suitable and rigorous analysis to describe the data. The authors did a particularly nice job of the discussion and decision of their "metrics of interest", though this is not the main aim of this work.

Strengths:

Due to the particularly nice and tractable cell-based model, the authors are able to perform a thorough analysis to compare the published data to that simulated with their model. They then used their computational model to investigate different growth mechanisms of volume growth and surface growth. With this approach, the authors are able to compare the metric of interest (here, the direction angle of a new mutant clone, the dispersion of mutants throughout the tumour) to quantify how the different growth models compare to the observed data. The authors have also used inference methods to identify model parameters based on the data observed. The authors performed a rigorous analysis and have chosen the metrics in an appropriate manner to compare the different growth mechanisms.

Context:

Improved mechanistic understanding into the early developmental stages of tumours will further assist in disease treatment and quantification. Understanding how readily and quickly a tumour is evolving is key to understanding how it will develop and progress. This work provides a solid example as to how this can be achieved with data alongside simulated models.

---

## [Author Response]

The following is the authors’ response to the original reviews.

**Public reports:**

In the public reports there is only one point we would like to discuss. It concerns our use of a computational model to analyse spatial tumour growth. Citing from the eLife assessment, which reflects several comments of the referees:

The paper uses published data and a proposed cell-based model to understand how growth and death mechanisms lead to the observed data. This work provides an important insight into the early stages of tumour development. From the work provided here, the results are solid, showing a thorough analysis. However, the work has not fully specified the model, which can lead to some questions around the model’s suitability.

The observables we use to determine the (i) growth mode and the (ii) dispersion of cells are modelindependent. The method to determine the (iii) rate of cell death does not use a spatial model. Throughout, our computational model of spatial growth is not used to analyze data. Instead, it is used to check that the observables we use can actually discriminate between different growth modes given the limitations of the data. We have expanded the description of the computational model in the revised version, and have released our code on Github. However, the conclusions we reach do not rely on a computational model. Instead, where we estimate parameters, we use population dynamics as described in section S5. The other observables are parameter free and model-independent. We view this as a strength of our approach.

**Recommendations for the authors:**

**Reviewer #1:**
(1.1) In Figure 1, the data presented by Ling et al. demonstrate a distinctive “comb” pattern. While this pattern diverges from the conventional observations associated with simulated surface growth, it also differs from the simulated volume growth pattern. Is this discrepancy attributable to insufficient data? Alternatively, could the emergence of such a comb-like structure be feasible in scenarios featuring multiple growth centers, wherein clones congregate into spatial clusters?

We are unsure what you are referring to. One possibility is you refer to the honey-comb structure formed by the samples of the Ling et al. data shown in Fig. 1A of the main text. This is an artefact arising from the cutting of the histological cut into four quadrants, see Fig. S1 in the SI of Ling et al. The perceived horizontal and vertical “white lines” in our Fig. 1A stems from the lack of samples near the edges of these quadrants. We have added this information to the figure caption.

An alternative is you are referring to the peaks in Fig 2A of the main text. The three of these peaks indeed stem from individual clones. We have placed additional figures in the SI (S2 B and S2 C) to disentangle the contribution from different clones. The peaks have a simple explanation: each clone contributes the same weight to the histogram. If a clone only has few offspring, this statistical weight is concentrated on a few angles only, see SI Figure S2 B.

(1.2) I am not sure why there are two sections about “Methods” in the main text: Line 50 as well as Line 293. Furthermore, the methods outlined in the main paper lack the essential details necessary for readers to navigate through the critical aspects of their analysis. While these details are provided in the Supplementary Information, they are not adequately referenced within the methods section of the main text. I would recommend that the authors revise the method sections of the main text to include pertinent descriptions of key concepts or innovations, while also directing readers to the corresponding supplementary method section for further elucidation.

We have merged the Section “Materials and Methods” at the end of the main text with the SI description of the data in SI 4.2 and placed a reference to this material in the main body.

(1.3) The impact of the particular push method (proposed in the model) on the resultant spatial arrangement of clones remains unclear. For instance, it’s conceivable that employing a different pushing method (for example, with more strict constraints on direction) could yield a varied pattern of spatial diversity. Furthermore, there is ambiguity regarding the criteria for determining the sequence of the queue housing overlapping cells.

Regarding the off-lattice dynamics we use, there are indeed many variants one could use. In nonexhaustive trials, we found that the details of the off-lattice dynamics did not affect the results. The reason may be that at each computational step, each cell only moves a very small amount, and differences in the dynamics tend to average out over time.

We deliberately do not give constraints on the direction. Such constraints emerge in lattice-based models (when preferred directions arise from the lattice symmetry), but these are artifacts of the lattice.

At cell division the offspring is placed in a random direction next to the parent regardless of whether this introduces an overlap. Cells then push each other along the axis connecting their two centers of mass – unlike in lattice based models a sequence of pushes does not propagate through the tumor straight away but sets off of a cascade of pushes. Equal pushing of two cells (i.e. two initial displacements as opposed to pushing one of the two) results in the same patterns of directed, low dispersion surface and undirected, high dispersion volume growth but is much harder computationally as it reintroduces overlaps that have been resolved in the previous step.

We have rewritten the description of the pushing queue in the SI Section 1. The choice of the pushing sequence is somewhat arbitrary but we found that it also has no noticable effect on the growth mode. Maybe putting it in contrast to depth-first approaches helps to illustrate this: We tried two queueing schemes for iterating through overlapping cells, width-first and depth-first. In both cases, we begin by scanning a given cell’s (the root’s) neighborhood for overlaps and shuffle the list of overlapping neighbours. In a width-first approach we then add this list to the queue. Subsequent iterations append their lists of overlapping cells to the queue, such that we always resolve overlaps within the neighborhood of the root first. A depth-first approach follows a sequence of pushes by immediately checking a pushed cell’s neighborhood for new overlaps and adding these to the front of the queue (which works more like a stack then). This can be efficiently implemented by recursion but has no noticeable performance advantage and results in the same patterns of directed, low dispersion surface and undirected, high dispersion volume growth. In our opinion the width-first approach of first resolving overlaps in the immediate neighborhood is more intuitive, which is why we adopted it for our simulation model.

(1.4) For the example presented in S5.1, how can the author identify from genomic data that mutation 3 does not replace its ancestral clade mutation 2? In other words, if mutation 2, 3 and 4 are linked meaning clone 4 survives but 2 and 3 dies, how does one know if clone 3 dies before clone 2? I understand that this is a conceptual example, but if one cannot identify this situation from the real data, how can the clade turnover be computed?

Thank you for this comment, which points to an error of ours in the turnover example of the SI: Clade 3 does in fact replace 2 and contributes to the turnover! (The algorithm correctly annotated clade 3 as orphaned and computes a turnover of 3/15 for this example). We have corrected this.

In this example, it does not matter for the clade turnover whether clone 3 dies before clone 2. As long as its ancestor (clone 2) becomes extinct it adds to the clade turnover. The term “replaces” applies to the clade of 3 which has a surviving subclone and thereby eventually replaces clade 2. The clade turnover its solely based on the presence of the mutations (which define their clade) and not on the individual clones.

(1.5) After reviewing reference 24 (Li et al.), I noticed that the assertions made therein contradict the findings presented in S3 (Mutation Density on Rings). Specifically, Li et al. state that “peripheral regions not only accumulated more mutations, but also contained more changes in genes related to cell proliferation and cell cycle function” (Page 6) and “Phylogenetic trees show that branch lengths vary greatly with the long-branched subclones tending to occur in peripheral regions” (Page 4). However, upon re-analysis of their data, the authors demonstrated a decrease in mutation density near the surface. It is crucial to comprehend the underlying cause of such a disparity.

The reason for this disparity is the way Li et al. labelled samples as belonging to peripheral or central regions of the tumour. We have added a new figure in the SI to show this: Fig. S14 shows the number of mutations found in samples of Li et al. against their distances from the centre, along with the classification of samples as center/periphery given in Li et al. In the case of tumor T1, the classification of a sample in reference Li et al. does not agree with the distance from the center: samples classified as core are often more distant from the center than those classified as peripheral. Furthermore, Lewinsohn et al. (see below) show in their Fig. 5 that samples classified as ‘center’ by Li all fall into a single clade, and we believe this affects all results derived from this classification. For this reason, we do not consider the classification in reference 24 (Li et al.) further. We now briefly discuss this in Section S3.3.

(1.6) The authors consider coinciding mutations to occur when offspring clades align with an ancestral clade. Nevertheless, since multiple mutations can arise simultaneously in a single generation (such as kataegis), it becomes essential to discern its impact on clade turnover and, consequently, the estimation of d/b.

The mutational signatures found here show no sign of kataegis. Also, the number of polymorphic sites in the whole-exome data is small and the mutations are uniformly spread across the exome. The point is well taken, however, the method requires single mutations per generation. In practice, this can be achieved by subsampling a random part of the genome or exome (see [45]). We tested this point by processing the data from only a fraction of the exome; this did not change the results. In particular, Figure S30 shows the turnover-based inference for different subsampling rates *L* of the Ling et al. data. Subsampling of sites reduces the exome-wide mutation rate, the inferred rate scales linearly with *L*, as expected.

(1.7) I could not understand Step 2 in Section S2.1, an illustration may be helpful.

We have added figure S2 explaining the directional angle algorithm to Section S2.1 in the supplementary information.

(1.8) Figure S2, does a large *rhoc* lead to volume growth rather than surface growth, not the other way around?

Thank you for catching this mix-up!

**Reviewer #2**
I do have a few minor comments/questions, but I am confident the authors will be able to address them appropriately.(2.1) Line 56: I am not sure what the units of “average read depth 74X” is in terms of SI units?

This number gives the number of sequence reads covering a particular nucleotide and is dimensionless. We have added this information.

(2.2) Lines 63 - 68: I am unsure what is meant by the terms “T1 of ca.” and “T2 of ca.”. Can these also be explained/defined please?

These refer to the approximate (circa) diameters of tumor 1 and tumor 2 in the data by Li et al. We have expanded the abbreviations.

(2.3) Line 69: I would like to see a more extensive description of the cell-based model here in the main text, such as how do the cells move. Moreover, do cells have a finite reach in space, do they have a volume/area?

We have expanded the model description in the main body of the paper and placed information there that previously was only in the SI.

(2.4) Line 76: You have said cells can “push” one another in your model. Do they also “pull” one another? Cell adhesion is know to contribute to tumour integrity - so this seems important for a model of this nature.

We have not implemented adhesive forces between pairs of cells so far. This would cause a higher pressure under cell growth (which can have important physiological consequences). However, the hard potential enforcing a distance between adjacent cells would still lead to cells pushing each other apart under population growth, so we expect to see the dispersion effect we discuss even when there is adhesion.

(2.5) Line 80-81: “due to lack of nutrient”. Is nutrient included in this work? It is my understanding it is not. No problem if so, it is just that this line makes it seem like it is and important. If it is not, the authors should mention this in the same sentence.

Thank you for pointing out this source of misunderstanding, your understanding is correct and we have modified the text to remove the ambiguity.

(2.6) Line 94-95: Since you are interested in tissue growth, recent work has indicated how the cell boundary (and therefore tissue boundary) description influences growth. Please also be sure to indicate this when you describe the model.

We presume you refer to the recent paper by Lewinsohn et al. (Nature Ecology and Evolution, 2023), which reports a phylogenetic analysis based on the Li et al. data. Lewinsohn et al. find that cells near the tumour boundary grow significantly faster than those in the tumour’s core. This is at variance with what we find; we were not aware of this paper at the time of submission. We now refer to this paper in the main text, and also have included a new section S3.4 in the SI accounting for this discrepancy. If you refer to a different paper, please let us know.

Briefly, we repeat the analysis of Lewinsohn et al., using their algorithm on artificial data generated by our model under volume growth. Samples were placed precisely like they were placed in the tumor analyzed by Li et al. We find that, even though the data was generated by volume growth, the algorithm of Lewinsohn et al. finds a signal of surface growth, in many cases even stronger compared to the signal which Lewinsohn et al. find in the empirical data. We have added subsection S3.4 with new figure S15 in the Supplementary Information.

(2.7) Line 107: “thus no evidence for enhanced cell growth near the edge of the tumour”. It is unclear to me how this tells us information relative to the tumour edge. It seems to me this is an artifact that at the edge of the tumour, there are less cells to compare with? Could you please expand on this a bit?

The direction angles tell us if new mutations arise predominantly radially outwards. With this observable, surface growth would lead to a non-uniform distribution of these angles even if we restrict the analysis to samples from the interior of the tumor (which, under surface growth, was once near the surface). So the effect is not linked to fewer cells for comparison. Also, we have checked the direction angles in simulations under different growth modes with the samples placed in the same way as in the data (see Figs. S3 and S4 right panels). We have expanded the text in the main text, section Results accordingly.

(2.8) I really enjoyed the clear explanation between lines 119 and 122 regarding cell dispersion!

Thank you!

(2.9) Figure 2B: Since you are looking at a periodic feature in theta, I would have expected the distribution to be periodic too, and therefore equal at theta=-180=180. Can you explain why it is different, please? Interestingly, you simulated data does seem to obey this!

The distribution of theta is periodic but the binning and midpoints of bins were chosen badly. We have replotted the diagram with bin boundaries that handle the edge-points -180/180 correctly. Thank you for pointing this out.

(2.10) Figure 3B: This plot does not have a title. Also, what do the red vertical lines in plots 3B, 3C and 3D indicate?

We have added the title. The red lines indicate the expectation values of the distributions.

(2.11) Figure 4: I am unsure how to read the plot in 4B. Also, what does the y-axis represent in 4C and 4D?

We have added explanations for 4B and have placed the labels for 4C and 4D in the correct position on the y-axes.

(2.12) Lines 194-199: you discuss your inferred parameters here, but you do not indicate how you inferred these parameters. May you please briefly mention how you inferred these, please?

These were inferred using the turnover method explained in the paragraph above, we have expanded the information. A full account is given in the SI Section S5.

(2.13) Line 258-260: “... mutagen (aristolochic acid) found in herbal traditional Chinese medicine and thought to cause liver cancer.” I do not see what this sentence adds to the work. Could you please be clearer with the claim you are making here?

Mutational signatures allow to infer underlying mutational processes. The strongest signature found in the data is associated with a mutagen that has in the past been used in traditional Chinese medicines. The patients from whom the tumours were biopsied were from China, so past exposure to this potent mutagen is possible. We are not making a big claim here, the mutational signature of aristolochic acid and its cancerogenic nature has been well studied and is referenced here. The result is interesting in our context because in one of the datasets (Li et al.) the signature is present in early (clonal) mutations but absent in later ones, allowing to make inferences from present data on the past. We have added the information that the patients were from China.

(2.14) In your Supplementary Information, S1, I believe your summation should not be over i, as you state in the following it is over cells within 7 cell radii. Please fix this by possibly defining a set which are those within 7 cell radii.

We have done this.